# MIRAI: EVALUATING LLM AGENTS FOR INTERNATIONAL EVENT FORECASTING

## ABSTRACT

We present MIRAI, a **benchmark** designed to systematically evaluate LLM agents as temporal forecasters to predict international events. Our benchmark features an agentic environment with tools for accessing an extensive database of historical, structured events and textual news articles. We refine the GDELT[1] event database with careful cleaning and parsing to curate a series of relational prediction tasks with varying forecasting horizons, assessing LLM agents' abilities from short-term to long-term forecasting. We further implement APIs to enable LLM agents to utilize different tools via a code-based interface. Notably, MIRAI features a dynamic data construction pipeline that supports periodically downloading recent news and events, and automatically generates the most recent test split. This allows us to evaluate any newly released model in a contamination-free manner as we can always construct a test split later than its knowledge cutoff date. MIRAI comprehensively evaluates the agents' capabilities in three dimensions: 1) autonomously source and integrate critical information from large global databases; 2) write codes with both domain-specific APIs and libraries for tool-use; and 3) jointly reason over historical knowledge from diverse formats and timespan to accurately predict future events. We establish a benchmark for evaluating LLM agents' international event forecasting abilities, advancing the development of more reliable models for analyzing international relations. [2]

## 1 INTRODUCTION

Accurate forecasting of international events is crucial (Brown & Lee, 2018), as understanding the evolution of geopolitical developments enables stakeholders to make well-informed decisions, mitigate risks, and seize opportunities in the interconnected world. Traditionally, researchers in international relations rely on domain expertise (Smith & Doe, 2020; Johnson & Roberts, 2019). They conduct detailed analyses of the complex interplay among nations, considering alliances, trade agreements, ideological affinities, and historical rivalries to forecast events such as conflicts, collaborations, or alliance shifts (Davis & Nguyen, 2017). With the rapid development of deep learning techniques, forecasting through data-driven neural networks becomes an attractive alternative. Despite their success, current methods rely on single types of information—either structured knowledge graphs (Mahdisoltani et al., 2015; Jin et al., 2020b; Li et al., 2021c) or textual datasets (Zou et al., 2022; Reddy et al., 2023). Knowledge graphs, although organized, can suffer from incompleteness (Huang et al., 2023; Galárraga et al., 2017) or bias (Huang et al., 2024), while textual analyses can lack necessary factual groundings of their reasoning to historical evidence, which compromises the interpretability and validation of their forecasts. These limitations raise concerns about AI forecasters' reliability, particularly for high-stake scenarios (Mclean et al., 2009).

Large Language Model (LLM) agents present a promising path to overcome these challenges (Sumers et al., 2024; Liu et al., 2023b; Weng, 2023; noa; Wang et al., 2023). These advanced AI systems exhibit the potential to mimic human experts by utilizing a diverse set of tools to automatically gather and process information from various sources, including text, knowledge graphs, and numerical data (Shen et al., 2023; Lu et al., 2023b; Zhuang et al., 2023b; Li et al., 2023b). Trained on extensive textual corpora, LLMs (Achiam et al., 2023; Anthropic, 2023; DeepMind, 2023; Touvron et al., 2023) are capable of grasping the subtleties of international relations, reasoning through complex relationships with linguistic explanations, and planning their tool usage effectively (Yuan et al.,

---

[1] GDELT: https://www.gdeltproject.org/

[2] We released our anonymous code repository for the data construction pipeline here, constructed benchmark here, and an interactive agent demo here.

Figure 1: **An example of forecasting the relations between Australia and China on 2023-11-18.** The agent fails to predict the change of relation and makes a wrong forecast.

2023; Liu et al., 2023a; Valmeekam et al., 2023; Ma et al., 2023b). Such capability opens up new possibilities for developing transparent and interpretable forecasting models that can be further scrutinized and refined.

Despite the immense potential of LLM agents for event forecasting, there is a lack of standardized benchmarks to assess their forecasting abilities in the realm of intricate international events. To address this gap, we introduce MIRAI (Multi-Information FoRecasting Agent Interface), the first forecasting benchmark designed as an agentic environment with rich structured and textual information. Leveraging the timely-updated Global Database of Events, Language, and Tone (GDELT) (Leetaru & Schrodt, 2013b), MIRAI adapts real-world international event data to create event-forecasting tasks in different test timeframes and across various time horizons. Our preparation script of MIRAI features a dynamic data construction pipeline that supports periodically downloading recent news and events, and automatically generate the most recent test split. This unique design allows us to consistently generate new **contamination-free** test sets, such that we can evaluate the forecasting capability of any recent LLM as long as its training data cutoff is before our split—a critical feature absent in previous forecasting benchmarks that only provided fixed datasets often predating most LLMs' knowledge cutoffs. Furthermore, MIRAI's agentic environment enables LLMs to interact with both relational and textual databases through APIs, facilitating autonomous information gathering, processing, and application in a contextually relevant manner.

We evaluate both open and closed source LLMs on MIRAI with ReAct-style (Yao et al., 2023b) agents using "Single Function" and "Code Block" action types. Our extensive experiments spanned multiple test splits (2023-11, 2023-12, 2024-01, 2024-02) and revealed: 1) Temporal forecasting tasks are challenging for LLM agents, with the highest-performing GPT-4o-mini agent using full suite of APIs achieves a 30.3 F1 score in second-level relation prediction tasks; while tasks involving long-term and fine-grained event forecasting are even more challenging; 2) The "Code Block" tool-use strategy, which allows more flexible interactions, demands robust code generation capabilities. E.g. GPT-4o-mini is able to better utilize and benefit from this strategy than other models we evaluate.

These findings emphasize the need for ongoing research into temporal reasoning and the effective use of tools by LLM agents. We expect that MIRAI could serve as a standard benchmark for evaluating LLMs in event forecasting. This would support the development of more precise and reliable models for political analysis, enhancing our understanding of global dynamics. To facilitate further research and development in this area, we release the code for our dataset construction pipeline and commit to updating our dataset split every month, ensuring that MIRAI remains a contamination-free and challenging benchmark for assessing LLM agent capabilities in international event forecasting.

In summary, our contributions are two-fold:

- We present MIRAI as **a comprehensive benchmark** uniquely combining three critical aspects for evaluating LLM agents' temporal forecasting capabilities:
  - An agentic environment with APIs for information integration, tool use, and reasoning.
  - Diverse data from structured events and textual news.
  - A dynamic pipeline ensuring contamination-free test splits beyond model knowledge cutoffs.
- Extensive experiments across models, relations, horizons, and temporal splits reveal key challenges in reasoning and tool use, offering insights and directions for advancing LLM forecasting agents.

## 2 THE MIRAI BENCHMARK

In this section, we introduce MIRAI benchmark from: the specifics of the data and task (Sec. 2.1), the implemented agents and environments (Sec. 2.2), and the database construction details (Sec. 2.3).

### 2.1 DATA AND TASK

We introduce MIRAI, a benchmark crafted for evaluating LLM agents for temporal forecasting in the realm of international events, with tool-use and complex reasoning. We consider *forecasting* as the

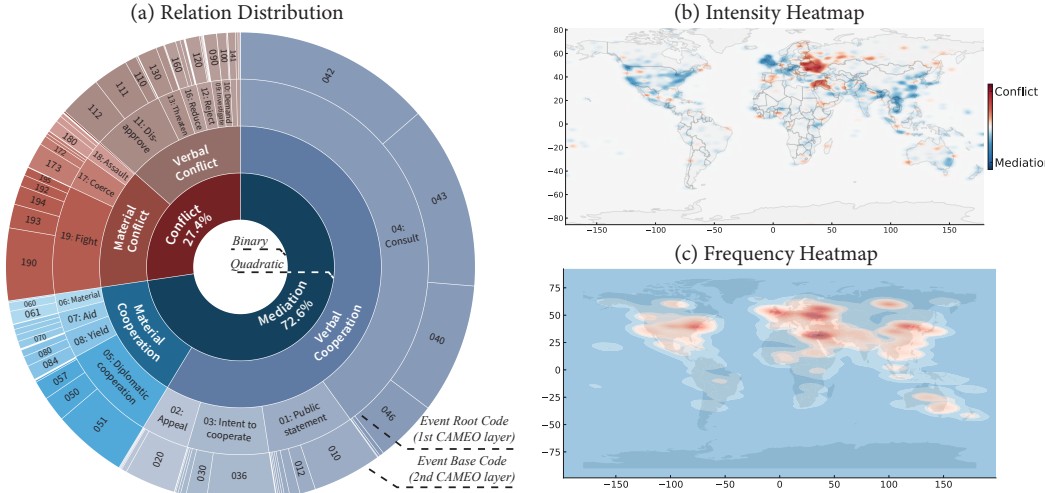

Figure 2: **MIRAI comprehensive global event coverage.** (a) Circular chart: Relation hierarchy and distribution in MIRAI. (b) Heatmap: Global events intensity, including areas of conflict (red) and mediation (blue). (c) Heatmap: Event frequency by region.

process of collecting essential historical data and performing temporal reasoning to anticipate the outcomes of future events.

**Structured and Textual Event Representations.** Formally, we represent an event as $e^t = (t, s, r, o)$ corresponding to $\boldsymbol{D}_e^t$, where $t$ is the timestamp[3], $s, o \in \mathcal{C}$ are respectively the subject and object countries[4] from the country pool $\mathcal{C}$, $r \in \mathcal{R}$ denotes the relation type defined by CAMEO ontology[5] (Boschee et al., 2015a), and $\boldsymbol{D}_e^t$ is the set of source news articles that mentioned event $e$ at timestamp $t$. Events at timestamp $t$ form a set $\boldsymbol{E}^t = \{e_1^t, ..., e_M^t\}$, where $M$ is the number of unique events at time $t$, with $\{\boldsymbol{E}^t\}_{t=1}^T$ being able to organize to temporal graphs where countries are nodes and relations are edges. Correspondingly, $\boldsymbol{D}^t = \{d_1^t, ..., d_N^t\}$ is the set of all news articles at $t$, where $N$ is the number of unique news articles at time $t$, with $\{\boldsymbol{D}^t\}_{t=1}^T$ as the full document collection. This dual representation of structured events and their associated textual information allows for a comprehensive analysis of international events, leveraging both the concise, categorized nature of the event tuples and the rich contextual details provided by the news articles.

**Hierarchical Event Categories.** As shown in Figure 2a, we incorporate two hierarchical relation levels from the CAMEO ontology to facilitate a detailed and comprehensive analysis of geopolitical dynamics. The first level includes 20 broad categories, represented by a two-digit code (e.g., "01: Public Statement" or "04: Consult"), which are subdivided into second-level categories identified by a three-digit code that corresponds to its parent category. For example, "03: Express intent to cooperate" is a first-level category that includes 10 different second-level relations such as "036: Express intent to meet". Subsequently, the quadruple "(2023-11-03, AUS, 036, CHN)" denotes that on 3 November 2023, the Australian leader announces a planned visit to China. These relations are also organized along two dimensions, from Verbal to Material and from Conflict to Cooperation, to form a quadratic categorization in the inner circle of Figure 2a.

**Event Forecasting Task.** The task of event forecasting $(t + l, s, ?, o)$ is to predict all the events between a pair of countries $s$ and $o$, happening $l$ days in the future from the current time $t$. Formally, given historical events $\boldsymbol{E}^{\leq t}$ and associated news articles $\boldsymbol{D}^{\leq t}$ up to the current time $t$, our goal is to forecast future relationships $\boldsymbol{E}_{s,o}^{t+l}$ between a pair of countries. This includes utilizing both structured event data and textual information, considering interactions not only between the target countries but also involving third parties, such as interactions with their mutual neighbors.

A forecasting task example in Figure 1 shows predicting Australia's actions towards China on 18 November 2023, based on information up to 17 November 2023. The query is formatted as "(2023-11-17 + [1 day], AUS, ?, CHN)", with a temporal distance of one day. Historical events show long-standing tensions between the two countries. Despite recent news of the Australian leader's visit to China and agreement to ease tensions, the overall relationship remains characterized by

[3]Each timestamp uniquely represents a day, formatted in "YYYY-MM-DD".

[4]In this paper, the term "country" includes all countries, dependent territories, special geographic areas, and their subdivisions. We use the standardized ISO-3166 Alpha-3 codes for country names, e.g. "AUS" for Australia.

[5]CAMEO, the Conflict and Mediation Event Observations, is a well-established ontology meticulously developed by domain experts over years, for categorizing international political events across multiple levels of granularity.

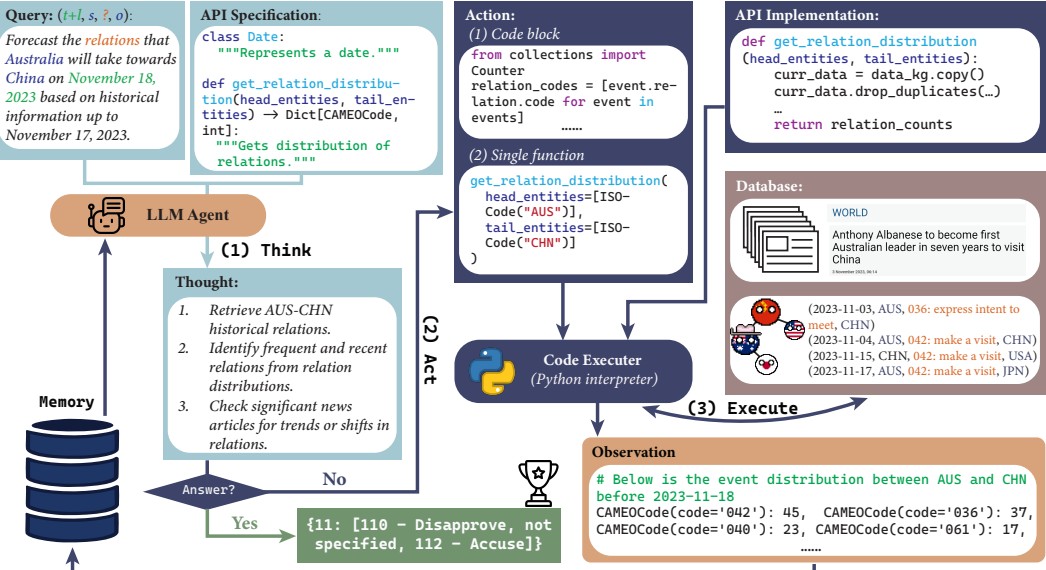

Figure 3: **Overview of the LLM agent's forecasting process using the ReAct strategy**. The framework consists of three main steps: (1) Think: Agent analyzes status and plans next action based on query and API specs. (2) Act: The agent generates a "Single Function" call or a "Code Block" to retrieve and analyze relevant data. (3) Execute: Python interpreter runs generated code, producing observations. These steps are repeated performed until the agent reaches final forecast.

`Disapprove` and `Accuse` actions on certain focuses. The agent, overly relying on short-term news, incorrectly predicts `Diplomatic cooperation`.

## 2.2 Agents and Environments

Similar to human political analysts, LLM agents must leverage a variety of information sources to make reliable predictions. We provide an environment with coding APIs to facilitate flexible access to various knowledge sources. Our LLM agent uses these APIs to interact with the environment for forecasting through the ReAct strategy (Yao et al., 2023a), characterized by the iterative process of *think*, *act*, and *observe*. This approach enables the agent to analyze the situation, retrieve data, and observe outcomes to make informed forecasts.

**APIs.** We provide the LLM agent with a comprehensive set of APIs to access a rich database of historical events and news articles. The API contains the essential *data classes* and *functions* designed for the various types of information within the database. *Data classes* cover unary types (e.g., date, country, relation) and composite types (e.g., date range, event, news articles). *Functions*, executable in Python, enable efficient database querying. The function design is characterized in two aspects:

- **Information types.** These functions cover diverse information types: country and relation mappings, hierarchies, and events and news articles statistics (counts, listings, and distributions). For example, `map_relation_description_to_cameo` takes a relation description and returns the five most likely relations with their CAMEO codes, names, and descriptions, providing precise relation information as needed.
- **Search conditions.** Additionally, the API functions for events and news articles offer optional parameters for tailored searches based on different criteria. For instance, `get_event` allows searches specifying conditions like `date_range`, `head_entities`, `tail_entities`, `relations`, and `text_description` to retrieve specific events from the database.

The data classes and functions provided in the API are shown in Appendix E.

**Interactions with Environment.** The environment is equipped with a Python code sandbox with full API and database access. MIRAI informs agents how to interact with the environment through API Specifications, including detailed Python docstrings for all data classes and functions that abstract implementation details. Agents use the ReAct strategy to iteratively alternate between *think*, *act*, and *observe* to gather information and forecast. The pipeline is illustrated in Figure 3.

1. **Think.** Specifically, the agent first *thinks* about the current situation. If confident based on the current information and analysis, it delivers a final forecast and stops. Otherwise, it plans further actions. In the first step shown in Figure 3, the agent plans to first retrieve recent events directly involving the two countries and then verify the details by checking related news articles.

2. **Act.** Next, the agent *acts* by generating executable codes to interact with the environment in two forms: "Single Function" for straightforward data retrieval or "Code Block" for complex operations including loops and conditionals. The agent can use API-defined functions and safe, well-established Python libraries such as `numpy`, `networkx`, and `scikit-learn`. Figure 3 illustrates that the agent can write either a "Code Block" with an imported library and an inline loop, or a "Single Function" call with targeted countries as an argument.

3. **Observe.** The environment executes the agent's code using a Python interpreter with full API and database access. All execution results are passed back to the agent as 'Observations'. For successful executions, the agent *observes* the corresponding output. For failures, it *observes* a specified error message along with the error type. In the figure, execution results are sent to the agent's memory for the next iteration.

These 'Thought', 'Action', and 'Observation' from the previous iterations are stored in the agent's memory base and used as the context for subsequent steps. Variables defined in previously generated code remain available for future actions. Examples of agents performing forecasting using "Code Block" and "Single Function" are shown in Appendix F. Prompts are shown in Appendix I.

### 2.3 DATASET CONSTRUCTION

Our database construction process involves three main steps: (1) raw data collection, (2) structured data cleaning, and (3) textual data processing. Throughout this process, we carefully consider critical aspects such as preventing test information leakage, standardizing information formats, ensuring data accuracy, verifying source reliability, and addressing ethical concerns.

**Raw Data Collection.** We construct the database based on the GDELT project[6], which captures global news media and extracts event information every 15 minutes, containing attributes about the event date, actor, action (relation), geography, and source news. The dataset used in this paper spans from January 1, 2023, upto February 29, 2024, providing a comprehensive base for our multi-split test design. Given GDELT's rapid updates and diverse sources, rigorous cleaning is essential.

**Structured Data Cleaning.** We refine the structured event data through four key steps: (1) *Temporal alignment*: Event dates are aligned with news publication dates to prevent information leakage. (2) *Event actor standardization*: Country codes are standardized to ISO-3166 norms, excluding missing or outdated codes to ensure consistency and eliminate ambiguity. (3) *Event scope delimitation*: We filter out all domestic events where the subject is identical to the object at the country level, focusing exclusively on international events to maintain relevance to our research objectives. (4) *Event relation standardization*: Event codes are standardized to the second level of the CAMEO ontology, balancing specificity with consistency and reliability. Third-level relations are omitted due to inconsistent hierarchical depth and increased risk of extraction errors at this granular level

**Textual Data Processing.** For associated news articles, we implement the following steps: (1) *Source reliability threshold*: We retain events with at least 50 daily news mentions. This reduces the influence of less reliable sources such as personal blogs and decreases erroneously extracted events. (2) *Textual context extraction*: News titles and content are sourced from the corresponding URLs of each event, retaining only textual information. (3) *Text cleaning*: We apply rigorous text cleaning procedures to reduce noise while enhancing the **reliability and ethical integrity** of the textual information, following the OBELICS protocol (Laurençon et al., 2023). This process operates at both paragraph and document levels, filtering low-value content with low word counts or high character/word repetition ratios and removing excessive special characters. Importantly, we employ a list of flagged words to identify and eliminate potentially sensitive or inappropriate content, aligning our data collection with ethical standards.

**Test Splits Construction.** We construct multiple test splits (2023-11, 2023-12, 2024-01, 2024-02), each covering a one-month period from November 2023 to February 2024. For each test split: (1) *Enhanced filtering*: We apply higher thresholds (100 daily mentions, 5 news articles) to ensure test data quality and reliability. (2) *Query formation*: We construct $(t, s, ?, o)$ queries, with answers listing relations between countries at time $t$. (3) *Balanced sampling*: For each month, we sample 100 queries to form balanced test splits, ensuring representation across dates, countries, and CAMEO code types. This multi-split design evaluates model performance across different time periods, assessing temporal robustness of forecasting capabilities. Using the same processing script, we can generate test split for any following month keep MIRAI a contamination-free and challenging benchmark.

---

[6]https://www.gdeltproject.org/

Table 1: Evaluation results with different agent tools and the tool-use strategies. The best-performing score is highlighted in **bold** and the second-best is underlined.

| Agent | Tool-Use | | Binary KL ($\Downarrow$) | Quad KL ($\Downarrow$) | First-level Relation (%) | | | Second-level Relation (%) | | |
|---|---|---|---|---|---|---|---|---|---|---|
| | Action Type | API | | | Pre. ($\Uparrow$) | Rec. ($\Uparrow$) | F1 ($\Uparrow$) | Pre. ($\Uparrow$) | Rec. ($\Uparrow$) | F1 ($\Uparrow$) |
| Direct IO | — | — | $6.5_{\pm1.7}$ | $15.9_{\pm1.5}$ | $27.6_{\pm8.1}$ | $19.7_{\pm5.9}$ | $18.8_{\pm6.9}$ | $6.6_{\pm1.5}$ | $5.1_{\pm0.4}$ | $3.5_{\pm0.8}$ |
| ZS-COT | — | — | $6.9_{\pm0.8}$ | $10.1_{\pm0.8}$ | $27.6_{\pm4.0}$ | $36.0_{\pm4.5}$ | $26.7_{\pm4.1}$ | $10.2_{\pm1.4}$ | $17.4_{\pm1.1}$ | $10.5_{\pm0.7}$ |
| ReAct | Single Function | *Event-Only* | $33.5_{\pm0.7}$ | $6.7_{\pm0.7}$ | $44.3_{\pm3.9}$ | $54.2_{\pm3.9}$ | $41.4_{\pm1.7}$ | $25.3_{\pm2.6}$ | $47.4_{\pm2.4}$ | $26.9_{\pm1.9}$ |
| ReAct | Single Function | *News-Only* | $6.1_{\pm1.0}$ | $12.8_{\pm0.6}$ | $27.8_{\pm3.1}$ | $25.9_{\pm2.9}$ | $21.8_{\pm2.3}$ | $6.3_{\pm2.2}$ | $9.0_{\pm2.0}$ | $5.4_{\pm1.3}$ |
| ReAct | Single Function | *All* | **$3.1_{\pm0.5}$** | **$5.9_{\pm1.0}$** | **$47.6_{\pm5.8}$** | **$58.3_{\pm2.6}$** | **$44.2_{\pm4.0}$** | **$28.7_{\pm3.9}$** | **$51.0_{\pm4.0}$** | **$29.6_{\pm3.7}$** |
| ReAct | Code Block | *All* | $5.1_{\pm0.9}$ | $8.9_{\pm0.5}$ | $27.1_{\pm4.0}$ | $38.6_{\pm2.5}$ | $25.9_{\pm2.2}$ | $11.6_{\pm2.4}$ | $26.3_{\pm2.0}$ | $12.6_{\pm1.7}$ |

**Analysis and Documentation.** The resulting dataset contains 1,296,991 GDELT event records, corresponding to 75,341 unique $(t, s, r, o)$ events and 401,013 unique news articles. Figures 2b and 2c illustrate the global distribution of our curated events, highlighting varying intensities of conflict and mediation across regions. We provide additional details of the benchmark data and human evaluation of the data quality in Appendix G. A standardized datasheet (Gebru et al., 2021) for MIRAI is in Appendix J, clearly and comprehensively documenting its motivation, composition, collection process, recommended uses, distribution, and maintenance.

**Evaluation metrics.** We instruct the agent to generate forecasts in a JSON dictionary, using two-digit first-level CAMEO codes as keys and lists of three-digit second-level codes as values. Evaluation involves calculating *precision*, *recall*, and *F1 score* between the predicted and ground-truth lists. Moreover, we map the predicted and ground truth relations to their respective binary and quadratic classes (as shown in Figure 2a). To measure prediction-ground truth alignment, we employ the *empirical Kullback-Leibler (KL) divergence*: $D_{\text{KL}}(P||Q) = \sum_i P(i) \log \left( P(i)/Q(i) \right)$, where $P$ and $Q$ represent the frequencies of ground-truth and predicted relations respectively. A lower KL divergence indicates a better alignment of the model's predictions with the ground-truth list.

## 3 EXPERIMENTS

In experiments, we compare forecasting performance across various agent methods with different tool-use (Sec. 3.1). Additionally, we evaluate the impact of different base models on forecasting (Sec. 3.2), and analyze contamination of the models' knowledge cutoff over test-time splits (Sec. 3.3). Finally, we conduct an in-depth analysis of key factors to agents' behavior (Sec. 3.4).

### 3.1 EVALUATE FORECASTING WITH DIFFERENT AGENT METHODS AND TOOLS

We investigate the effect of different tools (APIs) and agent tool-use strategies. We use GPT-3.5-Turbo (gpt, 2023) as the base model and evaluate on the 705 unsampled test queries in 2023-11. For all experiments, we set the model temperature to $0.4$ and run 5 times to calculate the mean and standard deviation. We provide the detailed prompts in Appendix I.

We consider two agent implementations without tool-use: **Direct IO** and **Zero-Shot Chain-of-Thought (ZS-CoT)** (Kojima et al., 2022; Wei et al., 2023). Direct IO let the LLM direcly provide answers using only its internal knowledge, serving as a baseline to reflect its internal world knowledge. ZS-CoT prompts the LLM for step-by-step reasoning before final prediction.

We implement **ReAct** (Yao et al., 2023b) for tool-use agents with two variants: 1) "Single Function" and 2) "Code Block" (detailed in Sec. 2.2). ReAct agents interact with our provided environments through an iterative process of thinking, acting, and observing. We thus further create API variants with access to 1) *News-Only* APIs, 2) *Event-Only* APIs, or 3) *All* API data classes and functions.

The experimental results in Table 1 reveal several key insights into agent performance:

**1) MIRAI presents a challenging task for LLM agents.** The best agent (ReAct with "Single Function" using all APIs) for second-level relation predictions achieves a precision of 28.7 and an F1 score of 29.6. These results underscore the complexity and difficulty of the temporal forecasting tasks in MIRAI and highlight the substantial room for improvement in LLM agents for event forecasting.

**2) Predicting fine-grained relations proves more difficult.** All models exhibit higher KL divergence for quadratic than binary classes, and lower F1 scores for second-level predictions compared to first-level ones. These findings confirm that predicting fine-grained relation types is more challenging.

**3) Diverse tool-use is critical for temporal forecasting.** ZS-CoT and Direct-IO, which rely solely on the internal world knowledge of LLMs for forecasting without tool-use, significantly underperform the ReAct agent with full API access to the database. This emphasizes the importance of basing forecasting and reasoning on retrieved historical data and knowledge. In terms of tool types, ReAct agents using *News-Only* APIs perform much worse than agents with *Event-Only* APIs. While news articles provide detailed context for events, they can also introduce noise and lead to issues such as

Table 2: Evaluation results on the 2024-02 test split using different base LLMs and action types. The best-performing score is highlighted in **bold** and the second-best is underlined. **Note that the traditional methods are task-specifically trained on training data up to 2023-06, while all LLM-based methods are zero-shot with their general pretrained knowledge. Human evaluation is conducted on a subset of 51 test events due to resource constraints and the time-intensive nature of expert evaluation.**

| Base LLM | Training Data Cutoff Date | Action Type | Binary KL ($\Downarrow$) | Quad KL ($\Downarrow$) | First-level Relation (%) | | | Second-level Relation (%) | | |
|---|---|---|---|---|---|---|---|---|---|---|
| | | | | | Pre. ($\Uparrow$) | Rec. ($\Uparrow$) | F1 ($\Uparrow$) | Pre. ($\Uparrow$) | Rec. ($\Uparrow$) | F1 ($\Uparrow$) |
| Mistral-7B-Instruct-v0.2 | 2023-12 | Single Func | $10.3_{\pm1.7}$ | $14.2_{\pm1.9}$ | $38.1_{\pm0.5}$ | $19.2_{\pm4.2}$ | $18.9_{\pm1.1}$ | $21.9_{\pm4.1}$ | $9.8_{\pm3.5}$ | $9.3_{\pm0.6}$ |
| | | Code Block | $9.1_{\pm2.3}$ | $14.3_{\pm1.6}$ | $31.3_{\pm5.1}$ | $12.5_{\pm2.6}$ | $15.1_{\pm3.6}$ | $13.1_{\pm1.2}$ | $9.4_{\pm1.6}$ | $8.4_{\pm1.9}$ |
| Llama-3-8B-Instruct | 2023-03 | Single Func | $9.0_{\pm2.4}$ | $14.1_{\pm1.6}$ | $39.8_{\pm1.6}$ | $15.6_{\pm1.2}$ | $18.6_{\pm0.2}$ | $15.8_{\pm0.5}$ | $11.8_{\pm0.6}$ | $10.3_{\pm0.1}$ |
| | | Code Block | $9.4_{\pm2.6}$ | $14.5_{\pm1.5}$ | $39.5_{\pm0.7}$ | $12.2_{\pm2.4}$ | $15.9_{\pm2.5}$ | $18.9_{\pm2.0}$ | $8.9_{\pm1.7}$ | $9.2_{\pm0.9}$ |
| Llama-3.1-8B-Instruct | 2023-12 | Single Func | $7.8_{\pm2.3}$ | $12.4_{\pm3.0}$ | $50.1_{\pm1.6}$ | $21.5_{\pm2.2}$ | $25.2_{\pm2.1}$ | $22.7_{\pm3.2}$ | $16.8_{\pm0.5}$ | $14.8_{\pm0.7}$ |
| | | Code Block | $9.1_{\pm2.3}$ | $14.4_{\pm1.5}$ | $35.7_{\pm1.3}$ | $15.9_{\pm0.0}$ | $18.3_{\pm0.1}$ | $14.7_{\pm0.1}$ | $12.8_{\pm1.0}$ | $10.0_{\pm0.2}$ |
| GPT-3.5-Turbo | 2021-09 | Single Func | $3.5_{\pm1.3}$ | **$7.5_{\pm2.7}$** | $55.7_{\pm5.8}$ | **$40.9_{\pm2.6}$** | $38.3_{\pm4.2}$ | $42.4_{\pm4.4}$ | **$34.3_{\pm5.0}$** | $28.3_{\pm3.9}$ |
| | | Code Block | $5.3_{\pm1.4}$ | $9.3_{\pm2.3}$ | $34.8_{\pm11.7}$ | $34.9_{\pm0.9}$ | $26.7_{\pm3.8}$ | $16.8_{\pm5.9}$ | $26.4_{\pm0.5}$ | $15.1_{\pm2.6}$ |
| GPT-4o-mini | 2023-10 | Single Func | $4.0_{\pm0.9}$ | $8.1_{\pm1.3}$ | **$61.3_{\pm11.4}$** | $34.7_{\pm1.9}$ | $39.0_{\pm6.0}$ | $40.0_{\pm5.5}$ | $32.6_{\pm1.6}$ | $29.7_{\pm3.8}$ |
| | | Code Block | **$3.2_{\pm0.8}$** | $7.7_{\pm1.9}$ | $59.8_{\pm5.1}$ | $37.1_{\pm0.4}$ | **$40.0_{\pm3.1}$** | **$46.5_{\pm2.1}$** | $29.7_{\pm0.6}$ | **$30.3_{\pm1.0}$** |
| *Traditional Forecasting Methods* | | **Task-specific** | | | | | | | | |
| REGCN (Li et al., 2021b) | 2023-10 | ✓ | 0.3 | 1.0 | 24.8 | 78.2 | 32.4 | 3.9 | 25.7 | 5.6 |
| ForecastQA (Jin et al., 2021a) | 2023-10 | ✓ | 9.7 | 13.8 | 55.0 | 16.2 | 22.1 | 40.0 | 8.8 | 12.0 |
| **Human evaluators** | — | — | 0.04 | 1.37 | 62.73 | 88.70 | 68.29 | 54.54 | 74.53 | 56.78 |

excessively long context, posing additional challenges for LLM agents. Moreover, the agents using both types of information achieve the optimal results.

## 3.2 EVALUATE FORECASTING WITH DIFFERENT BASE LLMS

We then investigate the role of the base LLMs in agent's performance. We evaluate both open-sourced LLMs, including Mistral-7B-Instruct-v0.2 (Jiang et al., 2023), Llama-3-8B-Instruct, and Llama-3.1-8B-Instruct (Dubey et al., 2024), as well as close-sourced LLMs including GPT-3.5-Turbo (gpt, 2023) and GPT-4o-mini (gpt, 2024). Comparisons are done on the **2024-02** test split that is after all models' training data cutoff date, which comprise 100 data-balanced queries. All models use ReAct framework with access to all APIs. The action types can be either "Single Function" or "Code Block" with a maximum tool call limit set to 20 steps. The same prompt is used across all models for fair comparison, as detailed in Appendix I. The experimental results are presented in Table 2, and we observe the following findings:

**1) Code Block benefits stronger LLMs but hurts weaker models:** The "Code Block" action type provides greater flexibility than the "Single Function". However, its benefits are not uniformly achieved across all models. Small open-sourced models and GPT-3.5-Turbo show reduced performance with Code Block, while GPT-4o-mini as a strong model gains improvements from it. This indicates that the ability to generate coherent and practical long Code Blocks is a distinguishing factor that determines LLMs as reliable forecasting agents.

**2) GPT-4o-mini outperforms other models:** GPT-4o-mini achieves the highest performance across many metrics in different levels. Notably, for second-level relation prediction, GPT-4o-mini achieves F1 scores of 29.7 and 30.3 using "Single Function" and "Code Block", surpassing all other models. Among the tested open-sourced smaller models, Llama-3.1-8B-Instruct leads the performance but still remains a significant performance gap to larger models. This indicates that MIRAI is hard enough, and can effectively distinguish different LLMs' reasoning capabilities.

**3) MIRAI naturally supports evaluating traditional forecasting methods:** We choose a Temporal Knowledge Graph (TKG) method REGCN (Li et al., 2021b) and a textual method ForecastQA (Jin et al., 2021a) as examples, trained both models on data up to 2023-06. These trained methods show strong results on first-level relation prediction, but fall short for fine-grained second-level relation prediction. We defer the implementation details and experimental discussions to Appendix D.3 and provide human forecasting performance as a reference in Appendix D.4.

We further conduct the following analysis to better understand agent capabilities:

**Can we make a small LM stronger via scaling inference-time compute?** It is evident that stronger LLMs have better agent performance; however, can we enhance a weaker LLM to achieve comparable forecasting performance using inference-time computation? To explore this, we take Mistral-7B-Instruct-v0.2 as the base LLM with ReAct using the "Single Function" strategy. For each query, we perform multiple sampling at a temperature of $0.4$. We then consider a variant of *self-consistency*, which only keeps entries appearing more than twice. We also calculate F1 (Max@K), which assesses the F1 score for each instance and keeps the maximum score across all rounds. As shown in Figure 4a, with more samples, the performance of Mistral-7B-Instruct-v0.2 significantly improves. Initially, a

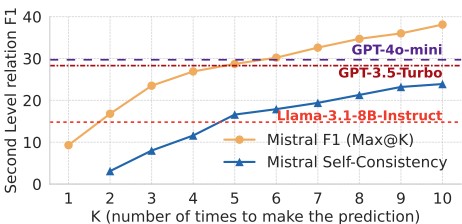
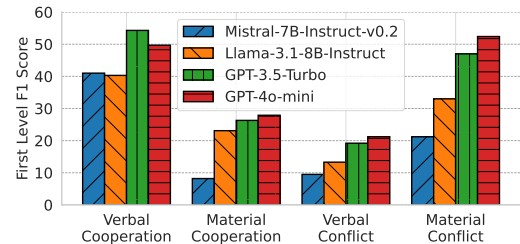

Figure 4: a) Self-consistency of Mistral-7B-Instruct-v0.2 model increases with more samples. b) F1 scores of different LLMs on relation prediction, categorized based on the quadratic classes.

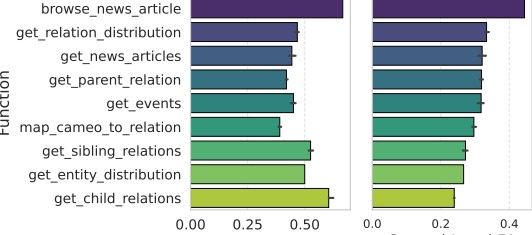
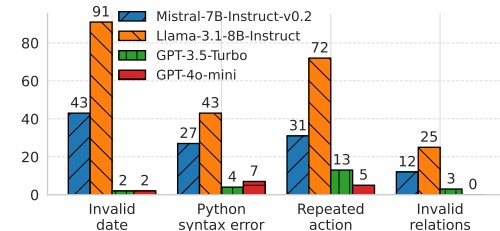

Figure 5: a) F1 Accuracy for each API function. b) Code execution error analysis for different LLMs.

single sample achieves an F1 score of 9.3, which is considerably lower than the scores achieved by larger models. However, as more solutions being sampled, the F1 (Max@K) and self-consistency of Mistral-7B-Instruct-v0.2 improve progressively. By the 10th samples, the Max@K reaches 38.1, even surpassing GPT-4o-mini. This result highlights the potential of inference-time search methods like self-consistency to push the boundaries of smaller language models in event forecasting.

**Code execution error analysis.** Our implemented agents interact with tools via code but often encounter execution error. We summarize the dominating error types for different LLMs in Figure 5b. We observe the invalid date as the most frequent error for smaller models, showing their difficulty in understanding time restrictions on the available historical data, which is set to before the query event date. Also, agents including larger models may propose repeated actions that have been conducted in its previous ReAct steps, failing to generate effective reasoning traces. We find Llama-3.1-8B-Instruct makes more execution errors than Mistral-7B-Instruct-v0.2 but achieves a better forecasting performance. We conduct further analysis on agents' final status in Appendix D.1 on their ReAct sequence length in Appendix D.2. Overall, GPT-4o-mini makes significantly fewer execution errors. This enhanced code generation capability contributes to its superior performance for event forecasting.

### 3.3 ANALYZE CONTAMINATION OF KNOWLEDGE-CUTOFF OVER TEST-TIME SPLITS

| Model | Training Data Cutoff Date | Test-Month Splits | | | |
|---|---|---|---|---|---|
| | | 2023-11 | 2023-12 | 2024-01 | 2024-02 |
| Llama-3-8B-Instruct | 2023-03 | $6.1_{\pm1.5}$ | $8.7_{\pm1.8}$ | $8.7_{\pm0.1}$ | $10.3_{\pm0.1}$ |
| Llama-3.1-8B-Instruct | 2023-12 | $15.8_{\pm5.7}$ | $15.9_{\pm3.6}$ | $16.3_{\pm2.0}$ | $14.8_{\pm0.7}$ |
| GPT-4-Turbo | 2023-12 | $\underline{30.0}_{\pm1.9}$ | $\underline{25.8}_{\pm3.1}$ | $32.2_{\pm2.8}$ | $28.9_{\pm3.2}$ |
| GPT-4o-mini | 2023-10 | $\mathbf{32.8}_{\pm2.6}$ | $25.9_{\pm3.2}$ | $\mathbf{33.2}_{\pm0.7}$ | $\mathbf{29.7}_{\pm3.8}$ |

Table 3: F1 (⇑) scores of second-level relation forecasting on different test splits, using "Single Function". The best-performing score is highlighted in **bold** and the second-best is underlined. More results in Appendix D.6.

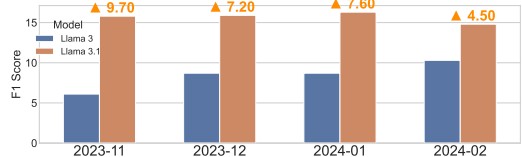

Figure 6: Llama-3.1 vs Llama-3 over four test splits. Llama-3.1 is much higher on 2023-11 (before its training data cutoff date). The gap is smaller afterwards. This shows we need to evaluate model using time-split after its cutoff date.

We compare model forecasting performance across multiple test splits in Table 3. MIRAI's dynamic data construction pipeline enables the creation of data-contamination-free test sets for newly released models, allowing us to study the effect of a model's knowledge cutoff time on forecasting performance.

We focus on comparing two open-sourced Llama3 models (Dubey et al., 2024) with different cutoff dates but similar training processes. Figure 6 illustrates their performance gap across four test splits. Notably, Llama-3.1 shows the largest lead over Llama-3 in the 2023-11 split, which is after Llama-3's cutoff but before Llama-3.1's. This gap diminishes in subsequent months, especially by 2024-02.

This observation indicates potential data contamination favoring the model with more recent training data (especially if eval date is before knowledge-cutoff date). It highlights the necessity of evaluating models using test splits that are later all compared models' cutoffs to ensure a rigorous forecasting task. Our benchmark uniquely supports this requirement, maintaining the integrity of forecasting assessments by reflecting true forecasting capabilities rather than data exposure advantages.

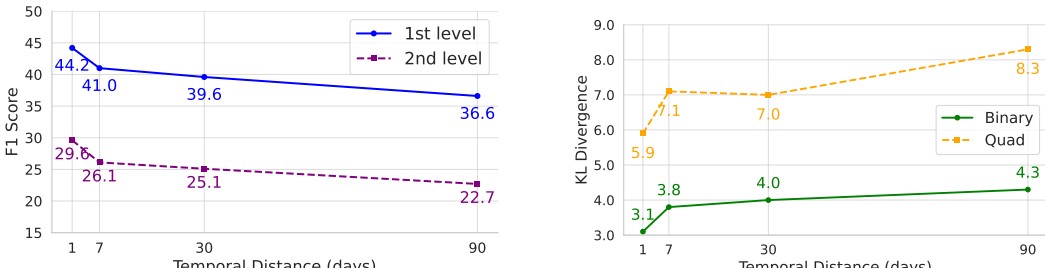

Figure 7: Evaluation of LLM Agents in different temporal distances of the forecasting event.

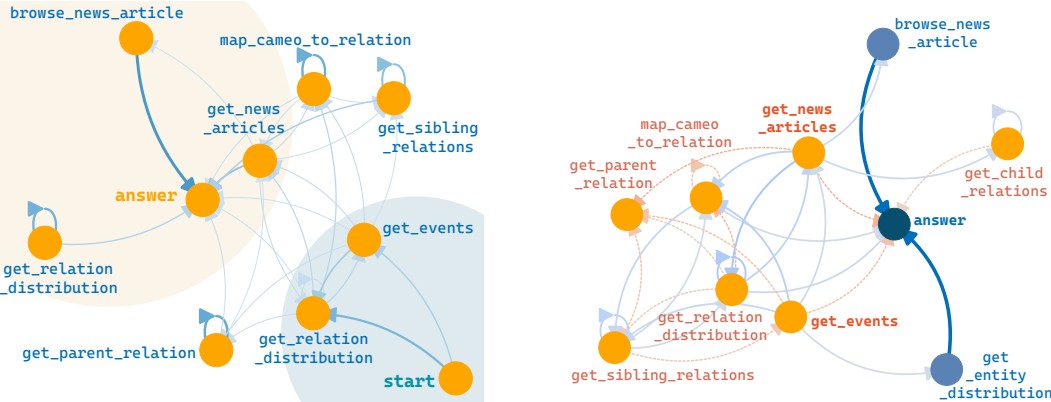

Figure 8: **Action order analysis in LLM agents.** a) Tool-Use Transition Graph of called API functions. Edge thickness indicates transition frequency. b) Freq.(correct) - Freq.(incorrect), in which blue (red) edges indicate positive (negative) contributions.

### 3.4 ANALYZING AGENT BEHAVIOURS

**Impact of temporal distance of the forecasting target.** Our defined event forecasting task varies by temporal distance $l$, which specifies how far into the future we want to predict. We thus conduct an ablation study with $l$ set to 1, 7, 30, and 90 days. Specifically, we fix the query event date and limit the accessible data to $l$ days prior to the query event date. The experimental results depicted in Figure 7 reveal a clear trend: as the temporal distance increases, the F1 score decreases and KL-divergence increases. This indicates that the agent's ability to provide accurate predictions diminishes for events further in the future. When the temporal distance is small, such as 1 or 7 days, the agent has access to more recent and relevant information, providing a strong signal (e.g., human experts' analysis) for making accurate predictions. Thus, to comprehensively benchmark the forecasting capabilities of LLM agents, we should focus on long-term predictions such as those spanning 30 or 90 days. These longer durations require the agents to capture and anticipate potential trend shifts, which may be influenced by a broader range of factors and more complex dependencies.

**Forecasting accuracy on different relation types.** We further split the test events into distinct quadratic relation classes and compute the F1 score for each class, as illustrated in Figure 4b. The results show that all models exhibit significantly higher performance for "verbal cooperation" and "material conflict", while lower in the other two categories. Several factors contribute to these observations. First, "verbal cooperation" events are more prevalent in the dataset, allowing the model to retrieve more such historical events. Second, "material conflicts" has a consistent pattern of extended duration within the same set of countries. Conversely, events categorized under "material cooperation" and "verbal conflicts", such as "057: Sign formal agreement" and "084: Return or release", tend to be more abrupt and unpredictable, demanding subtle trend analysis and contextual knowledge, leading to lower performance in these categories. These observations highlight the need for LLMs capable of understanding the nuances and complexities of different event types.

**How tool-use ordering influences forecasting.** We further investigate the impact of action order on the agent in "Single Function" mode. Figure 8 shows the transition graph from the initial query to the correct final answer, with thicker edges indicating more frequent transitions. Typically, the agent begins with get_relation_distribution or get_event to gather an initial set of recent and frequent events for key information, and often concludes with browse_news_article and get_news_articles, which retrieve news content to make accurate forecasts. To further understand whether each function helps forecasting, we subtract the frequency of incorrect predictions from those of correct predictions, as shown in Figure 8b. Here, blue edges represent sequences typically leading to accurate outcomes, and red edges indicate error-prone paths. Actions

Table 4: **Comparison of MIRAI with other temporal reasoning benchmarks.** "Method" refers to the methodology of original and recent models evaluated on the benchmark. Column "Underst." stands for Understanding. ICL stands for LLM w/ In-Context Learning. FT stands for fine-tuning.

| Benchmark | Temporal Reasoning | | Task Format | Information | | | | Method |
|---|---|---|---|---|---|---|---|---|
| | Underst. | Forecast | | Time Series | KG | Textual | API | |
| TempQuestions (Jia et al., 2018) | ✓ | | KGQA | | ✓ | | | KGQA Systems |
| CRONQuestions (Saxena et al., 2021) | ✓ | | KGQA | | ✓ | | | Bert-based FT |
| TempoQR (Mavromatis et al., 2021) | ✓ | | KGQA | | ✓ | | | Bert-based FT |
| TORQUE (Ning et al., 2020a) | ✓ | | QA | | | ✓ | | Bert-based FT |
| TempReason (Tan et al., 2023a) | ✓ | | QA | | ✓ | ✓ | | ICL + Task FT |
| TRAM (Wang & Zhao, 2024) | ✓ | | MCQ | | | ✓ | | ICL + Bert-based FT |
| TCELongBench (Zhang et al., 2024) | ✓ | ✓ | QA/MCQ | | | ✓ | | ICL |
| ForecastQA (Jin et al., 2021b) | | ✓ | MCQ | | | ✓ | | Bert-based FT |
| IntervalQA (Zou et al., 2022) | | ✓ | QA/MCQ | ✓ | | ✓ | | ICL |
| Approach (Halawi et al., 2024) | | ✓ | QA | | | ✓ | | ICL |
| ForecastBench (Karger et al., 2024) | | ✓ | QA | | | ✓ | | ICL |
| GDELT (Leetaru & Schrodt, 2013b) | | ✓ | Link Prediction | | ✓ | | | Graph FT + ICL + Task FT |
| ICEWS (Boschee et al., 2015b) | | ✓ | Link Prediction | | ✓ | | | Graph FT + ICL + Task FT |
| MIRAI | | ✓ | Relation List | | ✓ | ✓ | ✓ | LLM Agent |

like `browse_news_article` and `get_entity_distribution` typically result in correct answers more frequently. Notably, `get_news_articles` has a direct red link to the answer, suggesting that this function leads more often to incorrect answers because it only returns news titles, which are too vague for accurate prediction. However, when followed by `browse_news_article` and then providing the answer, the agent is more likely to produce correct outcomes. Similar patterns are observed with `get_event`, where adding `get_entity_distribution` turns a negative link to a positive one to the answer. Figure 5a further demonstrates how each function contributes to the final performance, showing `get_{child/sibling}_relation` are more useful for first-level prediction. These results emphasize the importance for strategic action planning in LLM agents for effective temporal forecasting.

## 4 RELATED WORK

Recent benchmarks for evaluating temporal reasoning in AI systems can be categorized into temporal understanding and temporal forecasting. Temporal understanding benchmarks (Jia et al., 2018; Saxena et al., 2021; Mavromatis et al., 2021; Tan et al., 2023a; Wang & Zhao, 2024) assess models' ability to comprehend temporal relations in available data. In contrast, temporal forecasting benchmarks, including our proposed MIRAI, focus on predicting future events based on historical data.

Existing forecasting benchmarks primarily use either QA (Jin et al., 2021b; Zou et al., 2022; Zhang et al., 2024; Halawi et al., 2024; Schoenegger et al., 2024) or link prediction (Boschee et al., 2015b; Leetaru & Schrodt, 2013b) task formulations. QA-based benchmarks typically use textual data, while link prediction tasks often rely on TKGs. MIRAI distinguishes itself by incorporating diverse information sources and employing a multi-relation prediction task format. Additionally, MIRAI introduces an agent-based methodology with intermediate reasoning steps and a construction pipeline that supports dynamic data updates. These features, summarized in Table 4, position MIRAI as a comprehensive and unique benchmark for evaluating temporal forecasting capabilities. We provide further discussions in Appendix C.

## 5 CONCLUSION AND LIMITATION

In conclusion, we introduce MIRAI, a novel benchmark for evaluating LLM agents in temporal forecasting of international events. Our key contributions include: 1) An agentic environment with APIs supporting comprehensive evaluation of agents' capabilities with diverse information sourcing, code-based tool use, and forecasting reasoning. 2) A dynamic data construction pipeline enabling monthly updates for contamination-free test splits for evaluating new models. 3) Extensive benchmarking across various agent methods, prediction horizons, and test-time splits, with in-depth analysis of factors influencing agent behavior. Our results reveal the challenges LLM agents face in generating contextually and syntactically correct code and performing complex temporal reasoning. By providing a standardized, dynamic, and comprehensive evaluation benchmark, MIRAI aims to contribute to the development of more accurate and trustworthy models for event forecasting, ultimately supporting more informed decision-making in international relations.

While our work addresses several key challenges, limitations remain, including the need for broader model coverage, expanded API functionality, and more diverse data types. A full discussion of limitations can be found in Appendix B.

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

APPENDIX

## A    REPRODUCIBILITY STATEMENT

We provide detailed information and explanation of our experimental setup, dataset, and evaluation metrics, ensuring reproducibility. Specifically, we describe the following:

- Experimental setup (Sec. 3): Detailed descriptions of the hardware and software configurations, including libraries and tools used.

- Dataset (Sec. 2: Comprehensive information on the dataset construction, including sources, preprocessing steps, and any techniques applied.

- Evaluation Metrics (Sec. 2.3): Clear definitions for the metrics chosen to evaluate performance.

Additionally, we commit to releasing the following resources for the replication of our results:

- Codebase: The complete codebase, including scripts for dataset construction, model serving, and evaluation. This is currently available on an anonymous repository here.

- Dataset: The processed dataset, along with detailed instructions on how to construct the dataset here.

By providing these resources, we aim to ensure that our work is fully reproducible and can be independently verified by the research community.

## B    LIMITATIONS

Our benchmark addresses several key challenges in evaluating LLM agents for event forecasting, including the integration of diverse information sources, the construction of an agentic interactive environment, and the contamination-free forecasting data and task formulation through its dynamic design. Despite this significant advantage, we acknowledge the following limitations:

1. **Model Coverage:** While we have tested representative open-source and closed-source LLMs, our experiments do not exhaustively cover all available models. Future work could expand to include a wider range of LLMs for more comprehensive evaluation.

2. **API Functionality:** The current API, while functional, has room for expansion. Future iterations could incorporate more sophisticated analytical tools, such as time series analysis functions, to encourage deeper temporal reasoning. Allowing agents to generate and add custom functions during their reasoning process could also lead to more diverse problem-solving approaches.

3. **Experimental Robustness:** The current experiments, while informative, are limited in scope due to cost and time constraints. Increasing the number of experimental rounds and adjusting parameters like model temperature could provide more statistically robust results and insights into model performance variability.

4. **Geopolitical Bias:** Reliance on GDELT as the primary data source may introduce biases in event coverage and interpretation, potentially skewing towards Western or English-language media perspectives. Future work could explore integrating multiple diverse data sources and languages to mitigate this limitation.

Addressing these limitations in future iterations will further enhance the benchmark's robustness and relevance in the rapidly evolving field of AI-driven event forecasting.

## C Additional Related Work

### C.1 Temporal Reasoning Benchmarks

Many benchmarks sensing the temporal reasoning ability of AI models have been constructed, but they have different focuses and settings with MIRAI, particularly in terms of task, information, and method, as shown in Table 4. One line of benchmarks focuses on the *temporal understanding* ability of the model (Jia et al., 2018; Saxena et al., 2021; Mavromatis et al., 2021; Ning et al., 2020a; Tan et al., 2023a; Wang & Zhao, 2024), such as understanding the temporal relations between available facts in knowledge graphs (KGs) or text, either a short piece of text or a document corpus. While the *temporal forecasting* task largely differs from understanding, where the reasoning target is **unseen** in the database for the model, and as such, the model has to not only understand but to reason. For forecasting, there are two main task formulations among previous benchmarks: the QA task format for benchmarks with history information represented in textual format (Zou et al., 2022; Zhang et al., 2024; Jin et al., 2021b; Halawi et al., 2024; Schoenegger et al., 2024; Karger et al., 2024), and graph link prediction task format for temporal knowledge graph (TKG)-based benchmarks (Boschee et al., 2015b; Leetaru & Schrodt, 2013b). However, the uniformat of information sources either lacks of support to clearly structural facts or contextual detail for the model to perform advanced reasoning, while in MIRAI, we provide both information sources. Additionally, we provide carefully constructed API with various data classes and functions that access to various part of the data. With the flexibility provided by code generation, the model is exposed to a broader and more flexible range of information. More importantly, MIRAI distinguishes itself by introducing an agentic environment specifically designed to evaluate LLM agents in the forecasting task. This represents a significant departure from previous work, which has not explored or even considered the potential of performing temporal forecasting tasks using LLM agents.

### C.2 Temporal Event Forecasting Benchmarks and Methods

**Existing Forecasting Benchmarks.** LLMs have been tested for their *temporal understanding* through tasks such as temporal event ordering or storyline comprehension (Tan et al., 2023b; Ning et al., 2020b; Zhou et al., 2019; Zhang & Choi, 2021; Wang & Zhao, 2023). In the context of *temporal forecasting*, LLMs have been evaluated on traditional structured-event-only benchmarks, such as ICEWS (2014) (Boschee et al., 2015b) and GDELT (2018) (Leetaru & Schrodt, 2013a); and also been evaluated on recent text-based temporal forecasting benchmarks, such as IntervalQA (2022) (Zou et al., 2022) and TCELongBench (2022) (Zhang et al., 2024). However, these evaluations typically involve providing LLMs with retrieved-context for in-context learning and then directly answering the forecast question, lacking intermediate reasoning steps and interaction between the LLM and the database crucial for accurate forecasting. Moreover, although experimental results on these benchmarks show that significant challenges and research value remain in forecasting, even when models encounter events before their training cutoff date, these benchmarks still only provide a fixed timeframe for the testing data, earlier than most recent LLMs.

**Traditional TKG and NLP Methods.** Significant research has been conducted in the field of structured event temporal forecasting. Various methods have been proposed, including aggregating temporal and relational information among entities (Jin et al., 2020a; Li et al., 2021b; Park et al., 2022), retrieving relevant historical events (Zhu et al., 2020; Sun et al., 2021; Li et al., 2021a), and modeling the continuous time evolution of events (Trivedi et al., 2017; Ding et al., 2021). Efforts have also been made to incorporate textual event information into Temporal Knowledge Graphs (TKGs). Glean (Deng et al., 2020a) and CMF (Deng et al., 2021) integrate textual embeddings into graph edges, while SeCoGD (Ma et al., 2023a) employs textual topic modeling to separate subgraphs. The MidEast-TE dataset and LoGo model (Ma et al., 2023c) utilize text clustering to construct complex events for forecasting with local and global contexts. However, these methods still perform forecast reasoning solely on graphs using graph-based techniques. Traditional NLP methods form the event forecasting task as MCQ, for example, the method in ForecastQA (Jin et al., 2021a) use text embedding models for retrieving related event news articles and appending them to the forecasting question for a Bert-based classification over answer candidates. Notably, MIRAI contains both structured and textual event data, supporting the test for both traditional TKG and NLP methods. We show more experimental results and illustrations in Appendix D.3.

**LLMs for TKG and NLP Methods.** Recent studies have explored the use of LLMs for temporal event forecasting by transforming the TKG formulation into text sequences and converting missing object prediction into next token prediction (Xu et al., 2023b). GPT-NeoX-ICL (Lee et al., 2023) employs in-context learning of LLMs and constructs prompts as a list of historical events in quadruplet

format. GENTKG (Liao et al., 2023) enhances the selection of historical event inputs using a temporal logical rule-based retrieval strategy, while LAMP (Shi et al., 2023) applies LLMs to perform abductive reasoning to assist the retrieval process. However, these works only investigate LLMs with in-context learning or simple task-specific fine-tuning. In contrast, MIRAI explores forecasting with an LLM agent that supports explicit information gathering and reasoning steps, enabling a hybrid approach that leverages both text and graph data.

### C.3 EVALUATION OF LANGUAGE AGENTS

Previous research has investigated the performance of LLM agents in a variety of domains, including arithmetic reasoning focused on obtaining correct solutions (Roy & Roth, 2015; Cobbe et al., 2021; Patel et al., 2021), proficiency assessment in utilizing tools and reporting results (Li et al., 2023a; Xu et al., 2023a; Zhuang et al., 2023a), evaluation of web navigation skills to find specific websites (Deng et al., 2023; Zhou et al., 2024; Liu et al., 2024), and planning travel itineraries under given constraints (Xie et al., 2024). However, these evaluations do not fully address the challenges posed by tasks involving complex international events with diverse information formats and temporal attributes. MIRAI presents a unique task in this context, where the agent must navigate and reason over the structured events and textual news articles with temporal information. This setup requires the agent to effectively handle multilateral relationships and information spanning different time periods.

### C.4 LLMs FOR TOOL-USE

Large Language Models (LLMs) have demonstrated remarkable language understanding (Radford et al., 2018) and reasoning capabilities (Wei et al., 2022). However, they also possess inherent limitations, such as their inability to provide up-to-date responses based on external knowledge or to perform complex mathematical reasoning. In response to these challenges, recent advancements have seen the integration of LLMs with various external tools (Mialon et al., 2023). Notable examples include TALM (Parisi et al., 2022) and ToolFormer (Schick et al., 2023), which utilize in-context learning to enhance the model's ability to leverage different tools in tasks like question answering and mathematical reasoning. Chameleon (Lu et al., 2023a) employs an LLM as a natural language planner to deduce the optimal sequence of tools to be used, subsequently executing these tools to generate the final output. AVIS (Hu et al., 2023) employs dynamic tree search to synthesize the most effective tool-use sequence. ToolkenGPT (Hao et al., 2023) integrates tool-use operators as special tokens and trains the model through sequence-to-sequence training. ToolLLM (Qin et al., 2023) introduces an instruction tuning dataset encompassing over 16,000 real-world APIs, significantly enhancing the model's capability to utilize these tools effectively.

### C.5 LLMs FOR CODE GENERATION

Early advancements in LLMs have greatly impacted the field of code generation. Notable early models such as GPT-3 (Brown et al., 2020) and PALM (Chowdhery et al., 2023) have demonstrated the potential of LLMs to assist in code completion and generation. Subsequent models have built upon the foundation of pre-trained LLMs, further refining capabilities specific to code generation. This has led to the development of code-specific LLMs such as Codex (Chen et al., 2021), Code T5 (Wang et al., 2021), CodeGen (Nijkamp et al., 2022), AlphaCode (Li et al., 2022), StarCoder (Li et al., 2023c), WizardCoder (Luo et al., 2023), and phi-1 (Gunasekar et al., 2023). Moreover, code from open-source platforms such as GitHub has increasingly been incorporated into the pre-training data for recent LLMs (Touvron et al., 2023; Achiam et al., 2023). This integration has led to improved performance of LLMs such as Code Llama (Roziere et al., 2023), Code-Qwen (Bai et al., 2023), and DeepSeek-Coder (Guo et al., 2024) on popular code-related tasks (Chen et al., 2021; Austin et al., 2021).

# D   ADDITIONAL EXPERIMENTAL RESULTS AND ANALYSIS

## D.1   ANALYSIS ON THE FINAL STATUS OF DIFFERENT LLM AGENTS

Table 5: Experiment results with different base LLMs on 2024-02 test splits: Average number of ReAct iterations and the number of test cases ending in different final statuses.

| Model | Training Data Cutoff Date | Action Type | Avg. Iterations | Final Status | | | |
|---|---|---|---|---|---|---|---|
| | | | | Final Answer | Consecutive Invalid Actions | Consecutive Repetitive Actions | Max Iterations Exceeded |
| Mistral-7B-Instruct-v0.2 | 2023-12 | Single Function | 4.53 | 81 | 13 | 5 | 1 |
| | | Code Block | 2.92 | 46 | 54 | 0 | 0 |
| Llama-3.1-8B-Instruct | 2023-12 | Single Function | 8.18 | 76 | 9 | 9 | 6 |
| | | Code Block | 3.65 | 40 | 56 | 5 | 0 |
| GPT-3.5-Turbo | 2021-09 | Single Function | 2.75 | 99 | 1 | 0 | 0 |
| | | Code Block | 3.52 | 87 | 11 | 1 | 1 |
| GPT-4o-mini | 2023-10 | Single Function | 4.65 | **100** | 0 | 0 | 0 |
| | | Code Block | 4.04 | **100** | 0 | 0 | 0 |

Following the experiment on different base LLMs and analysis of their code execution error in in Sec. 3.2, we further analyze the final status of different agents when they trigger the termination conditions in the ReAct process. The agent-environment interactions were terminated based on four different statuses:

- **Final Answer**: This status is triggered when the model explicitly indicates completion by declaring a "final answer," indicating that it has generated its final forecast.
- **Consecutive Invalid Actions**: Interaction is halted if the model consecutively executes non-executable actions more than three times.
- **Consecutive Repetitive Actions**: A similar threshold of three consecutive repetitive actions prompts termination under this status.
- **Maximum Iterations Exceeded**: Termination occurs if the agent engages in more than 20 rounds of iterations without making a forecast answer.

We observe from the final status: **1) The GPT series demonstrates enhanced capability in concluding interactions with a final answer:** The GPT series frequently concludes interactions with a final answer, indicating its superior ability to comprehend instructions, utilize tools correctly, and perform reasoning to generate the final forecast. In contrast, the smaller open-source models, Mistral-7b-Instruct-v0.2 and Llama-3.1-8B-Instruct, have a round one-tenth of cases and one-half of the cases generating consecutive invalid answers when using single function and code block, respectively, suggesting limited abilities in adhering to data types, functions, Python syntax, and following detailed instructions.

**2) Generating code blocks proves more challenging than generating single functions as action:** Across all models, there is a higher occurrence of errors when generating code blocks compared to single functions. This pattern highlights the greater complexity and increased likelihood of errors associated with composing multiple lines of code over simple function calls.

**3) Advanced models engage in more valid action steps, resulting in better forecasting performance:** Advanced models, such as GPT-4o-mini, engage in significantly more effective action steps, as shown by no case in ending with consecutive invalid or repeated actions. Llama-3.1-8B-Instruct, on the other hand, has the most number of action steps with execution errors in Figure 5 and the most number of cases ending without a direct final answer. This shows the importance of the planning and tool-use ability of LLM agents in effectively gathering historical information and making predictions.

## D.2   ANALYSIS ON AGENT REACT ITERATION SEQUENCE LENGTH

We further show the correlation of the model's forecasting performance to the number of ReAct iterations it takes to reach the final answer. Figure 9 shows the statistics of GPT-3.5-Turbo's performance with the "Single Function" action type and 20 as the maximum action steps allowed. It shows a slightly negative correlation between the two. We further investigate the longest few test cases, and find the model mostly trapped in code generation errors and fails to self-correct from the error message, and therefore, fails to gather useful information.

## D.3   EVALUATION FOR TRADITIONAL TKG AND NLP FORECASTING METHODS

We include the following additional traditional baselines that are fine-tuned:

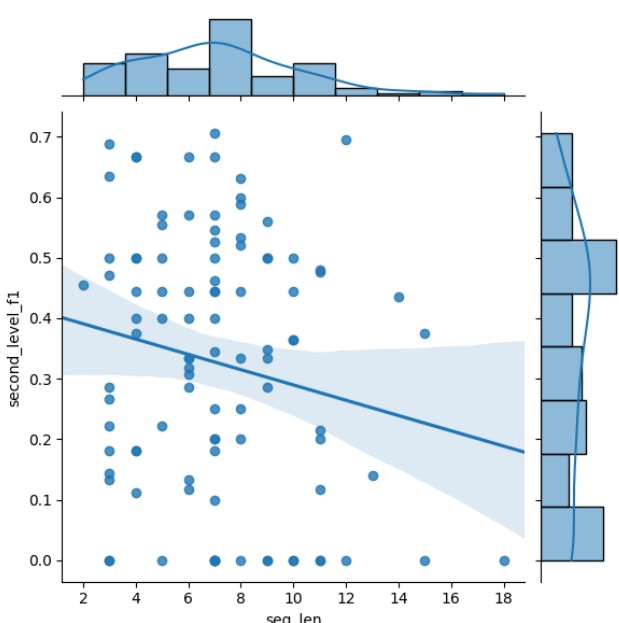

Figure 9: Correlation of F1 Accuracy to Action Sequence Length

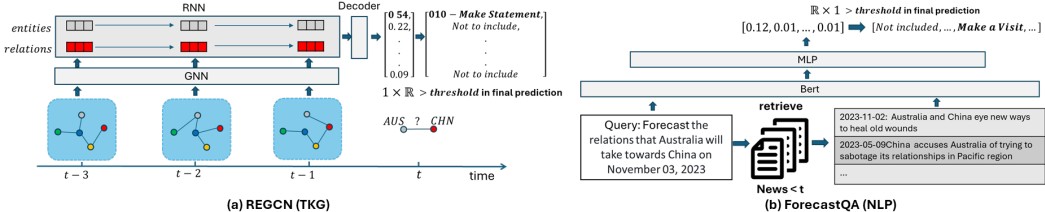

Figure 10: Illustration of RE-GCN and the adapted ForecastQA aggregate historical information and output forecasting probability (0-1) for each relation type. In comparison, examples using LLM agent in Appendix F provide explicit reasoning traces with better interpretability.

- **Structured TKG baseline**, we implemented RE-GCN (Li et al., 2021b), which learns relation in their graph edge embeddings and supports relation prediction.
- **Traditional textual baseline**, we adopted ForecastQA (Jin et al., 2021a), implementing a Bert-based classification model with BM25 to retrieve relevant news articles as additional encoder input.

Figure 10 illustrates the proposed frameworks of REGCN and ForecastQA, along with their threshold-based prediction processes. REGCN learns entity and relation representations in the event knowledge graph at each timestamp, modeling temporal development through a recurrent neural network. The learned representations are then used by a decoder to compute probabilities for each relation type. In contrast, ForecastQA employs a BERT encoder to process both the original query and retrieved news articles, utilizing a classification network for relation type probability computation.

We trained these traditional baselines on data prior to 2023-06 and evaluated them on the 2024-02 test split. Optimal thresholds were determined through a search over 0.3, 0.5, 0.7, and 0.9, resulting in 0.5 for REGCN and 0.3 for ForecastQA. The experimental results of these traditional models are shown in Table 2, along with LLMs' performance.

REGCN demonstrates superior performance in predicting higher-level relation types, particularly at the binary and quadratic levels, suggesting a good level of relation representation learning ability. However, it struggles with accurate forecasting at the more granular second level. ForecastQA achieves higher precision but significantly lower recall compared to REGCN, due to only predicting the relation type "Make a Visit" in most queries.

Notably, data-specific fine-tuning allows these traditional methods to achieve competitive performance with zero-shot smaller LLM agents, though they still largely underperform the most advanced LLMs.

Overall, these experiments and findings highlight MIRAI's value as a comprehensive benchmark for comparing diverse forecasting methods.

## D.4 HUMAN FORECASTING PERFORMANCE AS A REFERENCE

Table 6: Human and LLM agent forecasting performance on the sampled test queries. We have 2 human evaluators and we take the average performance. The best-performing score is highlighted in **bold** and the second-best is underlined.

| Model | Training Data Cutoff Date | Action Type | Binary KL ($\Downarrow$) | Quad KL ($\Downarrow$) | First-level Relation (%) | | | Second-level Relation (%) | | |
|---|---|---|---|---|---|---|---|---|---|---|
| | | | | | Pre. ($\Uparrow$) | Rec. ($\Uparrow$) | F1 ($\Uparrow$) | Pre. ($\Uparrow$) | Rec. ($\Uparrow$) | F1 ($\Uparrow$) |
| **Human evaluators** | — | — | **0.04** | 1.37 | 62.73 | **88.70** | **68.29** | **54.54** | **74.53** | **56.78** |
| Mistral-7B-Instruct-v0.2 | — | Single Function | 10.35 | 13.74 | 25.0 | 14.22 | 13.64 | 10.59 | 12.82 | 5.51 |
| | | Code Block | 8.21 | 11.59 | 30.0 | 7.04 | 10.69 | 23.33 | 4.74 | 7.62 |
| GPT-3.5-Turbo | 2021-09 | Single Function | 1.03 | 3.19 | **69.17** | 62.97 | 54.87 | 53.33 | 57.24 | 45.86 |
| | | Code Block | 3.95 | 8.16 | 36.44 | 48.68 | 28.74 | 23.06 | 36.91 | 13.46 |
| GPT-4-Turbo | 2023-12 | Single Function | 1.94 | 4.09 | 62.5 | 57.04 | 40.83 | 34.31 | 59.74 | 31.72 |
| | | Code Block | 0.18 | 2.77 | 35.33 | 54.48 | 34.03 | 25.46 | 56.05 | 27.83 |
| GPT-4o | 2023-10 | Single Function | 0.17 | 1.46 | 47.0 | 64.12 | 43.17 | 32.62 | 60.3 | 30.92 |
| | | Code Block | 0.16 | **1.22** | 48.06 | 78.2 | 53.02 | 39.0 | 69.88 | 40.32 |

To establish a reference point for our LLM agent evaluations, we conducted a human forecasting task with the following parameters:

- **Scope:** 10 queries $(t, s, ?, o)$ covering 51 distinct events.
- **Participants:** 2 college students (non-political science majors).
- **Task:** Given an event query $(t + 1, s, ?, o)$, generate forecasts for all possible relations.
- **APIs:** Participants used the same API library as the LLM agents.
- **Evaluation:** Applied the same metrics as those used for LLM agents.

As shown in Table 6, human performance surpassed that of LLM agents in most metrics, especially in recall. This highlights significant room for improvement in LLM performance.

It is important to note that our evaluators do not specialize in political science and likely fall short of what domain experts could achieve. This suggests that human forecasting, even at current performance levels, has room for enhancement.

We also acknowledge that event forecasting is an inherently challenging task for both humans and AI. The superior performance of non-expert human evaluators underscores the complexity of this task and the potential for further advancements in both human and AI forecasting capabilities.

## D.5 HUMAN EVALUATION ON THE DATASET QUALITY

To further assess dataset quality, we conducted a human evaluation on a subset of the test set consisting of 10 $(t, s, ?, o)$ queries, corresponding to 51 distinct $(t, s, r, o)$ events. Two human annotators evaluated whether the ground-truth events were correctly extracted based on their source news articles, scoring each event as 0 (incorrect) or 1 (correct). The average score across all evaluated events is 0.82, indicating high accuracy in the sampled test events. This evaluation confirms the reliability of our data cleaning process and the overall quality of our dataset.

## D.6 FORECASTING PERFORMANCE ON DIFFERENT TEST SPLITS

Table 7: Evaluation results of second-level relation forecasting on the different test splits, using "Single Function" as the action type. The best-performing score is highlighted in **bold** and the second-best is underlined.

| Model | Training Data Cutoff Date | 2023-11 | | | 2023-12 | | | 2024-01 | | | 2024-02 | | |
|---|---|---|---|---|---|---|---|---|---|---|---|---|---|
| | | Pre. ($\Uparrow$) | Rec. ($\Uparrow$) | F1 ($\Uparrow$) | Pre. ($\Uparrow$) | Rec. ($\Uparrow$) | F1 ($\Uparrow$) | Pre. ($\Uparrow$) | Rec. ($\Uparrow$) | F1 ($\Uparrow$) | Pre. ($\Uparrow$) | Rec. ($\Uparrow$) | F1 ($\Uparrow$) |
| Llama-3-8B-Instruct | 2023-03 | $10.7_{\pm4.0}$ | $6.1_{\pm2.4}$ | $6.1_{\pm1.5}$ | $13.6_{\pm3.0}$ | $10.2_{\pm2.8}$ | $8.7_{\pm1.8}$ | $16.0_{\pm1.2}$ | $9.0_{\pm0.8}$ | $8.7_{\pm0.1}$ | $15.8_{\pm0.5}$ | $11.8_{\pm0.6}$ | $10.3_{\pm0.1}$ |
| Llama-3.1-8B-Instruct | 2023-12 | $20.6_{\pm9.0}$ | $22.3_{\pm5.4}$ | $15.8_{\pm5.7}$ | $21.2_{\pm7.0}$ | $18.5_{\pm1.5}$ | $15.9_{\pm3.6}$ | $23.2_{\pm1.5}$ | $22.4_{\pm3.2}$ | $16.3_{\pm2.0}$ | $22.7_{\pm3.2}$ | $16.8_{\pm0.5}$ | $14.8_{\pm0.7}$ |
| GPT-4-Turbo | 2023-12 | $33.5_{\pm7.5}$ | $43.5_{\pm5.4}$ | $30.0_{\pm1.9}$ | $31.5_{\pm4.5}$ | $33.9_{\pm0.5}$ | $25.8_{\pm3.1}$ | $36.5_{\pm3.4}$ | $41.9_{\pm4.7}$ | $32.2_{\pm2.8}$ | $33.5_{\pm4.4}$ | $41.6_{\pm1.3}$ | $28.9_{\pm3.2}$ |
| GPT-4o-mini | 2023-10 | $41.3_{\pm9.0}$ | $41.4_{\pm1.4}$ | $32.8_{\pm2.6}$ | $39.4_{\pm7.5}$ | $25.4_{\pm2.9}$ | $25.9_{\pm3.2}$ | $45.9_{\pm3.1}$ | $36.6_{\pm1.7}$ | $33.2_{\pm0.7}$ | $40.0_{\pm5.5}$ | $32.6_{\pm1.6}$ | $29.7_{\pm3.8}$ |

We discuss the effect of models' knowledge cutoff on different test splits in Sec. 3.3. We show the full forecasting performance of second-level relation prediction with different base models over the four test splits in Table 7.

## D.7 ANALYSIS WITH RETRIEVE-AUGMENTED GENERATION (RAG) METHODS

Table 8: Evaluation results of GPT-4o-mini on the 2024-02 test split using different non-agentic methods and the ReAct agent with Single Function action type. The best-performing score is highlighted in **bold** and the second-best is underlined.

| Method | Augmented Context | Agent API | Binary KL ($\Downarrow$) | Quad KL ($\Downarrow$) | First-level Relation (%) | | | Second-level Relation (%) | | |
|---|---|---|---|---|---|---|---|---|---|---|
| | | | | | Pre. ($\Uparrow$) | Rec. ($\Uparrow$) | F1 ($\Uparrow$) | Pre. ($\Uparrow$) | Rec. ($\Uparrow$) | F1 ($\Uparrow$) |
| Direct IO | — | — | $3.6_{\pm1.0}$ | $7.6_{\pm1.9}$ | $39.5_{\pm3.2}$ | $44.8_{\pm3.2}$ | $34.9_{\pm3.5}$ | $15.4_{\pm0.8}$ | $23.9_{\pm3.6}$ | $15.4_{\pm0.2}$ |
| | CAMEO | — | $5.0_{\pm1.5}$ | $7.3_{\pm1.9}$ | $35.5_{\pm4.6}$ | $32.6_{\pm2.2}$ | $28.6_{\pm2.7}$ | $10.0_{\pm1.5}$ | $14.2_{\pm0.9}$ | $10.1_{\pm0.8}$ |
| RAG | *Events-Only* | — | $\mathbf{2.2_{\pm0.9}}$ | $\mathbf{5.9_{\pm2.0}}$ | $57.5_{\pm3.5}$ | $\mathbf{53.4_{\pm3.4}}$ | $\mathbf{50.5_{\pm3.8}}$ | $32.4_{\pm1.1}$ | $\mathbf{43.9_{\pm2.0}}$ | $\mathbf{33.2_{\pm1.4}}$ |
| | *News-Only* | — | $9.1_{\pm2.8}$ | $12.7_{\pm2.9}$ | $47.2_{\pm0.8}$ | $23.2_{\pm2.4}$ | $25.4_{\pm0.2}$ | $19.5_{\pm2.0}$ | $14.9_{\pm2.1}$ | $13.4_{\pm0.8}$ |
| | *All* | — | $\underline{2.3_{\pm1.4}}$ | $\underline{6.3_{\pm2.0}}$ | $59.0_{\pm1.2}$ | $\underline{48.1_{\pm1.2}}$ | $\underline{46.7_{\pm0.4}}$ | $36.4_{\pm5.3}$ | $\underline{38.8_{\pm1.2}}$ | $\underline{32.1_{\pm2.4}}$ |
| ReAct | — | *Event-Only* | $3.3_{\pm0.8}$ | $7.7_{\pm1.4}$ | $\mathbf{62.8_{\pm10.5}}$ | $39.0_{\pm0.8}$ | $41.7_{\pm5.3}$ | $\underline{44.2_{\pm3.3}}$ | $37.0_{\pm0.8}$ | $30.7_{\pm0.9}$ |
| | — | *News-Only* | $6.5_{\pm1.7}$ | $13.0_{\pm2.1}$ | $41.5_{\pm6.1}$ | $16.8_{\pm0.7}$ | $20.2_{\pm1.9}$ | $17.8_{\pm0.2}$ | $12.2_{\pm1.0}$ | $12.5_{\pm0.5}$ |
| | — | *All* | $3.6_{\pm0.9}$ | $8.0_{\pm1.5}$ | $\underline{61.7_{\pm10.1}}$ | $38.6_{\pm1.9}$ | $40.7_{\pm5.6}$ | $\mathbf{46.3_{\pm4.4}}$ | $32.9_{\pm3.8}$ | $31.1_{\pm2.6}$ |

### D.7.1 METHODS AND EXPERIMENTAL SETUP

Besides the **Direct IO / QA** and three **ReAct** agents with different tool-use that we already implemented in the paper, we add the following baselines:

- **Direct QA with Augmentation** (for comment Q4)
  - **QA with CAMEO**: We provide the CAMEO ontology in an ordered dictionary format mapping relation codes to their names and detailed descriptions. This is closer to the QA-format the authors mention, and a more fair comparison as the model can refer to the output vocabulary without needing to memorize CAMEO codes.
- **RAG Methods**
  Following recent work, we implement three RAG variants:
  - **RAG Events-Only**: Following GPT-NeoX-ICL **?**, we explore rule-based approaches for retrieving historical facts. Using the 'Pair' and 'Undirectional' setting, given a query event $(s, ?, o, t)$, we retrieve historical events $(s, r \in \mathcal{R}, o, < t)$ and $(o, r \in \mathcal{R}, s, < t)$. Events are sorted by recency with a cap of 30, aligning with the default cap of the `get_events` API function.
  - **RAG News-Only**: Following TCELongBench **?**, we employ BM25 retrieval to fetch the most query-relevant news articles before the query date. The top 15 news articles are retained, matching the default cap of the `get_news_articles` API function.
  - **RAG All**: Combines both retrieved structured events and textual news articles.

### D.7.2 COMPARISON OF RETRIEVAL STRATEGIES WITH AGENT

Both baseline approaches—CAMEO context augmentation and RAG methods—employ static, predefined retrieval strategies that are fixed for all queries and executed only once per query.

In contrast, our agentic approach enables dynamic, multi-step information gathering and reasoning. The agent **can** replicate the baseline retrieval strategy by fixing certain API parameter values, for example, the agent can use the function call `get_events(head_entities=[s, o], tail_entities=[o,s])` to get the retrieved context as RAG Event-Only, and use the function call `get_news_articles(text_description='(t, s, ?, o)')` to get the retrieved context as RAG News-Only; its capabilities extend far beyond these static approaches through its flexible parameter settings for each function call and multiple steps per query.

This multi-step, adaptive approach represents a fundamental shift from static retrieval to dynamic information gathering and reasoning, though it introduces higher requirements for the LLM's planning ability in:

- Automatically selecting optimal information-gathering strategies
- Integrating and reasoning over information of different formats
- Adjusting strategies based on intermediate findings and current context

### D.7.3 KEY FINDINGS AND ANALYSIS

Our experiment results reveal several important insights:

**Performance of the RAG Baselines:**
RAG demonstrates improved precision over Direct IO (15.4% Pre in second-level) when using either event (32.4% Pre in second-level) or text (19.5% Pre in second-level) information source independently, with event data contributing more significantly to recall (43.9% Rec in second-level). When combining the two information sources, RAG achieves higher precision (36.4% Pre in second-

level) but with a lower recall than RAG Event Only (from 43.9% to 38.8%), leading to a lower overall F1 score (from 33.2% to 32.1%), this suggests that a simple combination of both information in the context not effectively and collaboratively contribute to a better forecasting performance.

**Comparison and Insights for Agent:**

ReAct agents exhibit similar performance patterns with RAG when using different information sources (changed by the type of API functions available for the agent). Specifically, event data also contributes to high precision for the agent (62.8% Pre at first-level and 44.2% at second-level), outperforming RAG baselines. The structured event data consistently provides stronger signals for forecasting across both approaches, likely due to its standardized format.

However, agent baselines generally obtain a lower recall than RAG baselines. Meanwhile, while ReAct with full API access achieves higher average F1 scores in second-level relation prediction than its partial access performance (31.1% than 30.7% and 12.5%), the benefits of combining information sources aren't consistent across all relation hierarchies and methods, similar to RAG, suggesting substantial room for exploring more effective information integration strategies.

This reveals both promises and challenges of the agent's more flexible retrieval approach: RAG's predefined and fixed retrieval strategies can often yield stable performance, while the agent's dynamic and multi-step retrieval allows for flexible information gathering and integration, yet higher requirements for planning and reasoning sometimes also lead to relatively lower performance.

These observations underscore the core purpose of our benchmark: not just to compare current methods but to encourage the development of more advanced agentic forecasting approaches. The current performance patterns suggest significant opportunities for improving agent architectures, particularly in:

- Developing more robust and automatic planning strategies for multi-step information gathering
- Improving information integration capabilities across different information sources, formats, and temporal scales

## D.8 Analysis with Heuristic-based and Traditional TKG methods

Table 9: Evaluation results on the 2024-02 test split for relation prediction using heuristic-based and TKG-based methods and LLM agents based on GPT-4o-mini. The best-performing score is highlighted in **bold** and the second-best is underlined.

| Method | Training Data Cutoff Date | Prompt | MRR (%)(⇑) | Hit@10 (%)(⇑) | Binary KL (⇓) | Quad KL (⇓) | First-level Relation (%) Pre. (⇑) | Rec. (⇑) | F1 (⇑) | Second-level Relation (%) Pre. (⇑) | Rec. (⇑) | F1 (⇑) |
|---|---|---|---|---|---|---|---|---|---|---|---|---|
| RE-GCN | 2023-06 | — | 1.6 | 2.2 | 0.4 | 0.8 | 24.4 | 90.6 | 34.3 | 4.4 | **83.9** | 7.9 |
| | 2023-08 | | 1.9 | 2.8 | 0.4 | 1.1 | 23.9 | 86.1 | 32.9 | 4.6 | 40.0 | 7.0 |
| | 2023-10 | | 1.7 | 2.5 | **0.3** | 1.0 | 24.8 | 78.2 | 32.4 | 3.9 | 25.7 | 5.6 |
| | 2023-12 | | 2.9 | 5.7 | **0.3** | 2.5 | 23.9 | 74.4 | 31.3 | 5.5 | 28.4 | 7.9 |
| Recurrency (Strict) | 2023-06 | — | 17.4 | 45.0 | 3.2 | 3.6 | 32.8 | 77.1 | 42.9 | 18.7 | 67.8 | 27.2 |
| | 2023-08 | | 17.1 | **45.3** | 3.2 | 3.6 | 32.3 | 78.2 | 42.7 | 18.0 | 69.9 | 26.9 |
| | 2023-10 | | 15.8 | 41.0 | 2.4 | 3.1 | 29.7 | 83.5 | 41.3 | 14.3 | 76.8 | 23.0 |
| | 2023-12 | | **17.8** | 43.2 | 2.1 | 2.5 | 29.8 | 86.0 | 41.6 | 14.2 | 80.1 | 23.0 |
| ReAct | 2023-10 | Set Prediction | — | — | 3.6 | 8.0 | **61.7** | 38.6 | 40.7 | **46.3** | 32.9 | 31.1 |
| | | Rank (k=10) | — | 25.7 | 0.6 | 1.4 | 47.5 | 70.2 | **48.9** | 38.1 | 61.8 | **38.2** |
| | | Rank (k=30) | — | 12.0 | **0.3** | 0.8 | 34.9 | **91.2** | 45.8 | 22.5 | 82.8 | 31.7 |
| | | Rank (all) | 13.9 | 14.1 | 2.1 | 2.8 | 27.0 | 86.2 | 37.9 | 12.5 | 81.4 | 20.2 |
| | | Rank w.Prob (k=10) | — | 26.8 | 1.1 | 2.5 | 47.3 | 67.7 | 48.3 | 37.9 | 59.2 | **38.2** |
| | | Rank w.Prob (k=30) | — | 10.8 | **0.3** | **0.6** | 34.8 | 86.6 | 45.3 | 22.2 | 76.4 | 31.0 |
| | | Rank w.Prob (all) | 12.6 | 14.9 | 2.4 | 2.7 | 28.5 | 83.0 | 38.3 | 12.7 | 78.6 | 20.6 |

### D.8.1 Task Focus: Relation Prediction

Previous temporal knowledge graph research has explored both link prediction and relation prediction tasks, with notable works like DynamicGCN (Deng et al., 2020b), Glean (Deng et al., 2020a), and CMF (Deng et al., 2021) focusing on relation prediction. It is important to emphasize that neither task holds inherent priority over the other; rather, each serves distinct analytical purposes tailored to specific research objectives.

In Mirai, we focus on relation prediction as our primary task given our interest in studying dynamic relationship shifts between countries over time. This choice is particularly significant due to the structured nature of CAMEO relations in international event data. The CAMEO ontology offers a hierarchically organized framework that encompasses the entire spectrum of international interactions, ranging from material cooperation (e.g., providing aid, military collaboration) and verbal cooperation (e.g., diplomatic statements, expressions of support) to verbal conflict (e.g., accusations, rejections)

and material conflict (e.g., military actions, sanctions). This natural progression of political interactions—from the most cooperative to the most conflictual—provides a clear framework for analyzing the evolution of international relationships and a nuanced benchmark for assessing models' capacity to forecast shifts in these dynamics over time.

### D.8.2 EXPERIMENTAL SETUP

We evaluate three approaches in detail:

**RE-GCN** (Li et al., 2021b):

- **Model Architecture**: we follow the original method, where the model combines relation-aware graph convolutional layers with recurrent neural networks to jointly model structural dependencies and temporal dynamics in TKGs.
- **Data Cutoffs**: We experiment with four cutoff times (2023-06, 2023-08, 2023-10, 2023-12). For example, with 2023-10 cutoff, the model trains on data until 2023-10, validates on 2023-11, and tests on the 2024-02 split.
- **Test Input**: Uses single-step prediction with a 7-day historical window (t-7 to t-1), aligning with our agent experiments' one-day forecasting horizon.

**Recurrency (Strict)** (Gastinger et al., 2024):

- **Model**: The original work introduces three baselines that place strong inductive bias on fact recurrence over time: strict recurrency, relaxed recurrency, and their combination. While the original work and its scoring functions are specifically designed for link prediction, we adapt the strict recurrency variant for relation prediction. For a query event $(s, ?, o, t)$, we compute scores for all relations $r \in \mathcal{R}$ using:

$$\phi_\Delta((s, r, o, t), G) = \begin{cases} \Delta(t, \max\{k | (s, r, o, k) \in G\}) & \text{if } \exists k \text{ with } (s, r, o, k) \in G \\ 0 & \text{otherwise} \end{cases} \quad (1)$$

where $\Delta(t, k) = k/t$ measures temporal proximity.
- **Data Access**: Cutoff dates determine available historical events, e.g., 2023-10 means using only events before 2023-10-31 for score computation.

**ReAct Agent**:

- **Set Prediction**: Uses original prompt and output format, with the output directly used for set prediction metrics.
- **Ranking**: Modified prompts for ordered relation lists (k=10, 30, all). Only rank=all configurations are used for MRR calculation, while all configurations support Hit@10 evaluation.
- **Probabilistic Ranking**: Additional prompts requesting probability scores for ranked relations.
- **Evaluation**: All ranking evaluations use time-aware filtering to maintain consistency with other methods.

### D.8.3 METRIC SELECTION AND JUSTIFICATION

**Set-based Metrics**: For set prediction tasks, models output a discrete set of predicted relations for each query. We evaluate these predictions using:

- Precision: Proportion of predicted relations that are correct
- Recall: Proportion of actual relations that are predicted
- F1 Score: Harmonic mean of precision and recall

**Ranking-based Metrics**: For ranking tasks, models output an ordered list of relations with associated scores. Following TKG conventions of time-aware filtering (Bordes et al., 2013; Han et al., 2021), we implement:

- Mean Reciprocal Rank (MRR): Average reciprocal of the first correct relation's rank
- Hit@10: Proportion of queries where at least one correct relation appears in top-10 predictions

**Choice of Primary Metrics**: We prioritize set-based metrics as our primary evaluation criteria for Model Capability: The primary consideration is that current LLMs are better suited to generating discrete predictions through natural language reasoning than producing comprehensive ranked lists. This is evidenced by our experimental findings:

- **List Length Sensitivity**: ReAct agent's performance deteriorates with longer list requirements, with k=30 and k=all performing worse than k=10 in ranking metrics (Hit@10: 25.7% for k=10 vs 12.0% for k=30).

- **Prompt Sensitivity**: Performance varies between pure ranking and probability-weighted ranking (Hit@10: 25.7% vs 26.8% for k=10, and MRR: 13.9% vs 12.6% for k=all), suggesting that ranking outputs are sensitive to the prompt formulation and output format.

Given these challenges, we opted for metrics that more directly and reliably assess the agents' ability to predict discrete events without the confounding factors introduced by list generation and ranking.

### D.8.4 RESULTS ANALYSIS AND IMPLICATIONS

Our comprehensive evaluation reveals several significant insights about different forecasting approaches:

**Performance of the Recurrency Model and Insights for Agents**: The experiment results of the Recurrency (strict) model are shown in row 5-8 of the table. We observe that the Recurrency baseline demonstrates strong performance in ranking metrics (17.8% MRR and 43.2% Hit@10 with 2023-12 cutoff), leading other models; it also shows consistently high recall (86.0% Recall at first-level relation, and 80.1% recall at second-level relations with 2023-12 cutoff), suggesting that international events indeed often follow repetitive patterns.

Comparing with the ReAct agent (Set Prediction, the setting we used in the main paper) in row 9, we find that although the LLM agent could achieve much higher forecasting precision, it obtains much lower recall than the Recurrency baseline. We conducted a detailed analysis of this behavior by manually going through the reasoning traces generated by the LLM agent in the test set. One possible reason is that the agent has a strong tendency to select only a subset of the most frequent historical events in its prediction. For example, for the query $(2024\text{-}02\text{-}01, \text{PSE}, ?, \text{EGY})$, it uses function calls like
```
get_relation_distribution(date_range=DateRange(start_date=Date("2023-01-31"),
end_date=Date("2024-01-31")), head_entities=[ISOCode("PSE")],
tail_entities=[ISOCode("EGY")]),
```
and obtained a full frequency list as:
```
{CAMEOCode(code='042'):  32, CAMEOCode(code='192'):  18,
CAMEOCode(code='040'):  13, CAMEOCode(code='043'):  12,
CAMEOCode(code='046'):  8, CAMEOCode(code='080'):  6,
CAMEOCode(code='036'):  4, CAMEOCode(code='010'):  4,
CAMEOCode(code='190'):  3, CAMEOCode(code='073'):  3,
CAMEOCode(code='030'):  3, CAMEOCode(code='084'):  3,
CAMEOCode(code='020'):  3, CAMEOCode(code='172'):  2,
CAMEOCode(code='014'):  2, CAMEOCode(code='070'):  2,
CAMEOCode(code='044'):  2, CAMEOCode(code='086'):  1,
CAMEOCode(code='013'):  1, CAMEOCode(code='051'):  1}.
```
It then has a further step of checking recent news articles and obtains its final prediction as 040, 042, and 192, which are the top three frequent relations.

The effectiveness of simple temporal recurrency heuristics underscores the importance of incorporating more historical pattern analysis in the future development of forecasting agents, in particular, improving their recall of capturing a greater proportion of true relationships between countries.

**Performance of the TKG Baselines and Insights for Agents:** The experiment results of the RE-GCN are shown in rows 1-4 of the table. We observe that the RE-GCN demonstrates strong performance in high-level relation prediction, resulting in 0.3 for binary-level relation KL (cooperation or conflict) and 0.8 for quadratic-level relation KL (verbal/material cooperation/conflict), reflecting its advantage in capturing the high-level dynamics over bilateral relationships. It also shows consistently high recall in more fine-grained relation levels (90.6% Recall at first-level relation, and 83.9% recall at second-level relations with 2023-06 cutoff), suggesting its effectiveness in modeling positive correlation between query and multiple ground-truth relations.

Comparing with the ReAct agent (Set Prediction, the setting we used in the main paper) in row 9, we find that although the LLM agent also could achieve much higher forecasting precision, it obtains much lower recall than the RE-GCN baseline. We manually go through the reasoning traces generated by the LLM agent in the test set, and conclude the following possible insights and future directions of improvement compared with TKG baselines:

One major possible reason is that the current agent mostly focuses on analyzing only the bilateral events between the query entities $s$ and $o$. For instance, it typically sets the function parameter `head_entities` to the query subject, and `tail_entities` to the query object only, obtaining only events and news directly between the two. However, this analysis largely oversimplifies real international relationships where countries have engaged in multi-party and complex interactions.

Events between two countries could be affected by regional or global events. In contrast, TKG methods excel in capturing this multi-party and multi-relational history by leveraging multi-layer graph convolutions, where neighboring information is aggregated to enhance the modeling of each node (entity embedding) and edge (relation embedding). Therefore, when making predictions between two countries, the TKG models consider a much broader relation network than the current LLM agents, leading to higher recall and better generalization, especially when the bilateral history is sparse.

Another problem we observed from the current LLM agent behavior is its tendency to hallucinate, particularly in listing the existence of relations and interpreting the meanings of relations in the CAMEO ontology, which leads to lower precision and recall. For example, in the example we show in Appendix F.3, in its trajectory step 3, the agent attempts to explain and conclude its final prediction: '042' Make a public statement (high frequency in historical data); '036' Negotiate (also high frequency); '057': Express intent to cooperate (indicated by recent news context). However, the correct meanings are *Make a visit* for '042' and *Sign formal agreement* for '057'. This example highlights two issues: firstly, the LLM agent's overconfidence in its understanding of the CAMEO ontology without verifying the relation meanings through function calls (such as `map_cameo_to_relation` and `map_description_to_cameo`); secondly, its over-reliance on the semantic meaning of relations rather than their structural context. In contrast, TKG models learn relations by leveraging the historical graph structure, which inherently learns to capture the contextual meaning of each relation.

To enhance future LLM agents, incorporating a hybrid approach that combines semantic understanding with structural learning from TKGs could help to enhance relation modeling and address certain hallucinations.

### D.9 EVALUATION ON IMPACT OF LLM PARAMETER SIZE

Table 10: Evaluation results on the 2024-02 test split using different base LLMs with different number of model parameters. The best-performing score is highlighted in **bold** and the second-best is underlined.

| Base LLM | Training Data Cutoff Date | Action Type | Binary KL ($\Downarrow$) | Quad KL ($\Downarrow$) | First-level Relation (%) | | | Second-level Relation (%) | | |
|---|---|---|---|---|---|---|---|---|---|---|
| | | | | | Pre. ($\Uparrow$) | Rec. ($\Uparrow$) | F1 ($\Uparrow$) | Pre. ($\Uparrow$) | Rec. ($\Uparrow$) | F1 ($\Uparrow$) |
| Llama-3.2-1B-Instruct | 2023-12 | Single Func | $\underline{9.5}_{\pm1.9}$ | $16.0_{\pm1.7}$ | $23.7_{\pm6.6}$ | $10.0_{\pm1.9}$ | $\underline{11.7}_{\pm2.8}$ | $8.8_{\pm2.6}$ | $7.2_{\pm0.2}$ | $6.1_{\pm1.3}$ |
| | | Code Block | $10.1_{\pm2.2}$ | $16.2_{\pm2.2}$ | $24.0_{\pm5.7}$ | $8.1_{\pm2.1}$ | $10.0_{\pm3.4}$ | $7.6_{\pm1.9}$ | $5.7_{\pm0.7}$ | $5.1_{\pm1.8}$ |
| Llama-3.2-3B-Instruct | 2023-12 | Single Func | $12.1_{\pm2.2}$ | $\underline{15.4}_{\pm1.9}$ | $\mathbf{36.3}_{\pm2.2}$ | $\mathbf{13.1}_{\pm3.7}$ | $\mathbf{16.7}_{\pm3.0}$ | $\mathbf{19.9}_{\pm0.7}$ | $\mathbf{8.3}_{\pm2.6}$ | $\mathbf{9.3}_{\pm0.9}$ |
| | | Code Block | $\mathbf{9.3}_{\pm2.1}$ | $\mathbf{15.1}_{\pm0.5}$ | $\underline{26.7}_{\pm0.7}$ | $\underline{10.5}_{\pm0.5}$ | $11.0_{\pm0.8}$ | $\underline{13.1}_{\pm1.5}$ | $\underline{8.0}_{\pm0.4}$ | $\underline{7.1}_{\pm0.3}$ |

While LLM's parameter count can influence model capabilities, it is not the sole or even primary determinant of performance. Other crucial factors include model architecture, training data quality and recency, and advanced training techniques. For instance, GPT-4o-mini's competitive performance with GPT-3.5-turbo, despite having fewer parameters, can be attributed to more recent architectural improvements, different training data, and advanced training methodologies.

To systematically investigate the impact of parameter size while controlling for other variables, we conduct additional experiments comparing models within the same family. Table 10 presents evaluation results comparing Llama-3.2-1B-Instruct and Llama-3.2-3B-Instruct on the 2024-02 test split, both sharing the same architecture, training data cutoff (2023-12), and training methodology.

Our analysis reveals two key findings:

1) **Parameter size shows a consistent positive correlation with forecasting performance within the same model family.** The 3B model outperforms its 1B counterpart across all prediction levels, from binary (12.1% vs 9.5% in Single Function mode) to second-level relations (9.3% vs 6.1% in Single Function mode).

2) **The impact of parameter size varies across different action types.** While the 3B model maintains its advantage in both modes, the performance gap between 1B and 3B models narrows with Code Block actions. This smaller gap likely reflects the increased complexity of code generation, as our earlier experiments showed that Code Block actions can potentially hurt smaller, less capable models while benefiting more advanced ones.

# E ADDITIONAL INFORMATION ABOUT API

## E.1 OVERVIEW OF API DATA CLASSES AND FUNCTIONS

Table 11: API data classes and their attributes

| Class Name | Attributes and Types |
| --- | --- |
| Date | date: str # 'YYYY-MM-DD' |
| DateRange | start_date: Optional[Date], end_date: Optional[Date] |
| ISOCode | code: str # 3-letter ISO code |
| Country | iso_code: ISOCode, name: str |
| CAMEOCode | code: str # CAMEO code |
| Relation | cameo_code: CAMEOCode, name: str, description: str |
| Event | date: Date, head_entity: ISOCode, relation: CAMEOCode, tail_entity: ISOCode |
| NewsArticle | date: Date, title: str, content: str, events: List[Event] |

Table 12: API functions categorized by functionality

**Functions related to Countries and Relations**

```
map_country_name_to_iso(name: str) -> List[Country]
map_iso_to_country_name(iso_code: ISOCode) -> str
map_relation_description_to_cameo(description: str) -> List[Relation]
map_cameo_to_relation(cameo_code: CAMEOCode) -> Relation
get_parent_relation(cameo_code: CAMEOCode) -> Relation
get_child_relations(cameo_code: CAMEOCode) -> List[Relation]
get_sibling_relations(cameo_code: CAMEOCode) -> List[Relation]
```

**Functions related to Events**

```
count_events(date_range: Optional[DateRange], head_entities: Optional[List[ISOCode]],
tail_entities: Optional[List[ISOCode]], relations: Optional[List[CAMEOCode]]) -> int

get_events(date_range: Optional[DateRange], head_entities: Optional[List[ISOCode]],
tail_entities: Optional[List[ISOCode]], relations: Optional[List[CAMEOCode]],
text_description: Optional[str]) -> List[Event]

get_entity_distribution(date_range: Optional[DateRange], involved_relations: Optional[
List[CAMEOCode]], interacted_entities: Optional[List[ISOCode]], entity_role: Optional[str
]) -> Dict[ISOCode, int]

get_relation_distribution(date_range: Optional[DateRange], head_entities: Optional[List[
ISOCode]], tail_entities: Optional[List[ISOCode]]) -> Dict[CAMEOCode, int]
```

**Functions related to News**

```
count_news_articles(date_range: Optional[DateRange], head_entities: Optional[List[ISOCode
]], tail_entities: Optional[List[ISOCode]], relations: Optional[List[CAMEOCode]],
keywords: Optional[List[str]]) -> int

get_news_articles(date_range: Optional[DateRange], head_entities: Optional[List[ISOCode]],
 tail_entities: Optional[List[ISOCode]], relations: Optional[List[CAMEOCode]], keywords:
Optional[List[str]], text_description: Optional[str]) -> List[Tuple[Date, str]]

browse_news_article(date: Date, title: str) -> str
```

# F ADDITIONAL FORECASTING EXAMPLES OF LLM AGENT

In this section, we show examples of how LLM agents perform reasoning for the forecasting query. The query-specific values are highlighted in yellow in the system prompt and query prompt, which follows the prompt templates shown in Appendix I. For simplicity, we use {api_description} as a placeholder for the API specification in the prompt, which is replaced by the actual contents shown in Appendix **??** in experiments.

## F.1 GPT-4O-BASED AGENT WITH REACT AND "CODE BLOCK" ACTION

### F.1.1 QUERY DETAILS

- **Query Quadruplet**: (2023-11-03, AUS, ?, CHN)
- **Temporal Distance**: 1; therefore, the current date is 2023-11-02
- **Agent Max Steps**: 20

### F.1.2 QUERY PROMPT

Please forecast the relations that Australia will take towards China on November 03, 2023 based on historical information up to November 02, 2023. I.e. forecast the relation CAMEO codes in query event Event(date=2023-11-03, head_entity=ISOCode(AUS), relation=CAMEOCode(?), tail_entity=ISOCode(CHN)).

### F.1.3 SYSTEM PROMPT

You are an expert in forecasting future events based on historical data. The database contains news articles from January 1, 2023 to the current date November 02, 2023 and the events extracted from these articles. The events are in the form of (date, subject country, relation, object country), where the countries are represented by ISO 3166-1 alpha-3 codes and the relations are represented by the CAMEO codes defined in the "Conflict and Mediation Event Observations" ontology. The relations are hierarchical: first-level relations are general parent relations represented by two-digit CAMEO codes, while second-level relations are more specific child relations represented by three-digit CAMEO codes. Child relations have the same first two digits as their parent relations. For example, "01" is a first-level relation, and "010" and "011" are some of its second-level relations. The relations in the database are represented in the second-level form.

Your task is to forecast the future relations between two entities in a given query. You have access to a defined Python API that allows you to query the database for historical events and statistics, and to get precise information about the ISO country codes and CAMEO relation codes. You are also authorized to utilize additional safe, well-established Python libraries such as numpy, pandas, scikit-learn, and NetworkX to enhance your data analysis and forecasting accuracy.

The defined API is described as follows:
``` python
{api_description}
```

You will use an iterative approach, interleaving "Thought", "Action", and "Observation" steps to collect information and perform the forecast. You may perform up to 20 iterations. The steps are as follows:

- "Thought": Analyze the current information and reason about the current situation, and predicts which API you want to use (try to use different APIs to collect diverse information) or make a decision that you want to make a final answer.
- "Action": Use the API to gather more information or provide the final forecast.
    - If gathering more data: the action must be an executable Python code snippet that starts with ```python and ends with ```. It can contain multiple lines of codes and function calls using the defined API or Python libraries. You must use print() to output the results, and only the printed output will be returned in the observation step.
    - If making the final forecast: the action must start immediately with "Final Answer:", and follow with the answer in the expected JSON format. This should not be enclosed within triple backticks.
- "Observation": Return the printed output of the executed code snippet.

To make a reasonable forecast, you should collect both news and relational evidence to support your prediction. When you are fully confident that you accumulate enough information to make the final forecast, you should start the "Thought" with your reasoning using the news and structural information to make the prediction, and then start the "Action" step with "Final Answer:" followed by the answer in the expected JSON format. The answer should be a JSON dictionary where the keys are the forecasted two-digit first-level CAMEO codes and the values are lists of forecasted three-digit second-level CAMEO codes that are child relations of the key. For example, "Action: Final Answer: {"01": ["010", "011", "012"], "02": ["020", "023"]}".

The final answer will be evaluated based on the precision and recall of the forecasted first-level and second-level relations, so only include confident first-level and second-level CAMEO codes in your final forecast.

Try to use different APIs and Python libraries to collect diverse information (including multi-hop relations), such as the precise meaning of CAMEO codes, insights from news content, relational data, and statistical analyses to support your forecasts. Consider not only the frequency of the relations but also the temporal aspects of the data when making your forecast.

### F.1.4 AGENT REACT PROCESS

1. **Thought**: To forecast the relations that Australia will take towards China on November 03, 2023, we need to gather historical data on the interactions between these two countries. This includes examining the frequency and types of relations observed in the past, as well as any recent trends or significant events that might influence future relations.

    We will proceed with the following steps:

    (a) **Retrieve Historical Events**: Gather events involving Australia (AUS) and China (CHN) to understand the context and types of relations that have occurred in the past.

    (b) **Analyze Relation Distribution**: Analyze the distribution of different CAMEO codes in these events to identify the most common types of interactions.

    (c) **Examine Recent Trends**: Focus on recent events to detect any emerging trends or shifts in relations.

    (d) **Contextual Analysis**: Review news articles to gain insights into the context and underlying reasons for these relations.

    Let"s start by retrieving historical events between Australia and China.

    **Action**:

```python
from datetime import datetime

# Define the date range for the historical data collection
start_date = Date("2023-01-01")
end_date = Date("2023-11-02")
date_range = DateRange(start_date=start_date, end_date=end_date)

# Define the ISO codes for Australia and China
aus_code = ISOCode("AUS")
chn_code = ISOCode("CHN")

# Retrieve historical events involving Australia and China
events = get_events(date_range=date_range, head_entities=[aus_code], tail_entities=[
    chn_code])

# Print the retrieved events
for event in events:
    print(event)
```

    **Observation**:

```
Event(date=Date("2023-11-02"), head_entity=ISOCode("AUS"), relation=CAMEOCode("042"),
    tail_entity=ISOCode("CHN"))
Event(date=Date("2023-11-02"), head_entity=ISOCode("AUS"), relation=CAMEOCode("040"),
    tail_entity=ISOCode("CHN"))
Event(date=Date("2023-11-02"), head_entity=ISOCode("AUS"), relation=CAMEOCode("036"),
    tail_entity=ISOCode("CHN"))
Event(date=Date("2023-11-01"), head_entity=ISOCode("AUS"), relation=CAMEOCode("042"),
    tail_entity=ISOCode("CHN"))
Event(date=Date("2023-11-01"), head_entity=ISOCode("AUS"), relation=CAMEOCode("173"),
    tail_entity=ISOCode("CHN"))
```

```
 6  Event(date=Date("2023-11-01"), head_entity=ISOCode("AUS"), relation=CAMEOCode("036"),
        tail_entity=ISOCode("CHN"))
 7  Event(date=Date("2023-10-31"), head_entity=ISOCode("AUS"), relation=CAMEOCode("036"),
        tail_entity=ISOCode("CHN"))
 8  Event(date=Date("2023-10-31"), head_entity=ISOCode("AUS"), relation=CAMEOCode("042"),
        tail_entity=ISOCode("CHN"))
 9  Event(date=Date("2023-10-31"), head_entity=ISOCode("AUS"), relation=CAMEOCode("046"),
        tail_entity=ISOCode("CHN"))
10  Event(date=Date("2023-10-30"), head_entity=ISOCode("AUS"), relation=CAMEOCode("036"),
        tail_entity=ISOCode("CHN"))
11  Event(date=Date("2023-10-30"), head_entity=ISOCode("AUS"), relation=CAMEOCode("042"),
        tail_entity=ISOCode("CHN"))
12  Event(date=Date("2023-10-30"), head_entity=ISOCode("AUS"), relation=CAMEOCode("046"),
        tail_entity=ISOCode("CHN"))
13  Event(date=Date("2023-10-29"), head_entity=ISOCode("AUS"), relation=CAMEOCode("036"),
        tail_entity=ISOCode("CHN"))
14  Event(date=Date("2023-10-27"), head_entity=ISOCode("AUS"), relation=CAMEOCode("036"),
        tail_entity=ISOCode("CHN"))
15  Event(date=Date("2023-10-27"), head_entity=ISOCode("AUS"), relation=CAMEOCode("042"),
        tail_entity=ISOCode("CHN"))
16  Event(date=Date("2023-10-26"), head_entity=ISOCode("AUS"), relation=CAMEOCode("173"),
        tail_entity=ISOCode("CHN"))
17  Event(date=Date("2023-10-26"), head_entity=ISOCode("AUS"), relation=CAMEOCode("061"),
        tail_entity=ISOCode("CHN"))
18  Event(date=Date("2023-10-26"), head_entity=ISOCode("AUS"), relation=CAMEOCode("042"),
        tail_entity=ISOCode("CHN"))
19  Event(date=Date("2023-10-25"), head_entity=ISOCode("AUS"), relation=CAMEOCode("042"),
        tail_entity=ISOCode("CHN"))
20  Event(date=Date("2023-10-23"), head_entity=ISOCode("AUS"), relation=CAMEOCode("042"),
        tail_entity=ISOCode("CHN"))
21  Event(date=Date("2023-10-23"), head_entity=ISOCode("AUS"), relation=CAMEOCode("036"),
        tail_entity=ISOCode("CHN"))
22  Event(date=Date("2023-10-23"), head_entity=ISOCode("AUS"), relation=CAMEOCode("031"),
        tail_entity=ISOCode("CHN"))
23  Event(date=Date("2023-10-23"), head_entity=ISOCode("AUS"), relation=CAMEOCode("061"),
        tail_entity=ISOCode("CHN"))
24  Event(date=Date("2023-10-22"), head_entity=ISOCode("AUS"), relation=CAMEOCode("040"),
        tail_entity=ISOCode("CHN"))
25  Event(date=Date("2023-10-22"), head_entity=ISOCode("AUS"), relation=CAMEOCode("061"),
        tail_entity=ISOCode("CHN"))
26  Event(date=Date("2023-10-22"), head_entity=ISOCode("AUS"), relation=CAMEOCode("036"),
        tail_entity=ISOCode("CHN"))
27  Event(date=Date("2023-10-22"), head_entity=ISOCode("AUS"), relation=CAMEOCode("043"),
        tail_entity=ISOCode("CHN"))
28  Event(date=Date("2023-10-22"), head_entity=ISOCode("AUS"), relation=CAMEOCode("042"),
        tail_entity=ISOCode("CHN"))
29  Event(date=Date("2023-10-21"), head_entity=ISOCode("AUS"), relation=CAMEOCode("042"),
        tail_entity=ISOCode("CHN"))
30  Event(date=Date("2023-10-21"), head_entity=ISOCode("AUS"), relation=CAMEOCode("040"),
        tail_entity=ISOCode("CHN"))
```

2. **Thought**: The retrieved events indicate a variety of relations between Australia and China. To forecast the future relations, we need to analyze the distribution of these relations to identify the most common types and any recent trends.

   Next, we will analyze the distribution of the different CAMEO codes in these events to identify the most frequent interactions. This will help us understand the dominant types of relations and any emerging patterns.

   **Action**:

```
 1  from collections import Counter
 2
 3  # Extract the CAMEO codes from the retrieved events
 4  cameo_codes = [event.relation.code for event in events]
 5
 6  # Count the frequency of each CAMEO code
 7  cameo_code_counts = Counter(cameo_codes)
 8
 9  # Print the distribution of CAMEO codes
10  for code, count in cameo_code_counts.items():
11      print(f"CAMEO Code: {code}, Count: {count}")
```

   **Observation**:

```
 1  CAMEO Code: 042, Count: 10
 2  CAMEO Code: 040, Count: 3
 3  CAMEO Code: 036, Count: 8
 4  CAMEO Code: 173, Count: 2
```

```
5 CAMEO Code: 046, Count: 2
6 CAMEO Code: 061, Count: 3
7 CAMEO Code: 031, Count: 1
8 CAMEO Code: 043, Count: 1
```

3. **Thought**: The distribution of CAMEO codes from the historical events between Australia and China reveals that certain types of interactions are more frequent than others. Specifically, the most common relations are:

   - **042**: 10 occurrences
   - **036**: 8 occurrences
   - **040**: 3 occurrences
   - **061**: 3 occurrences
   - **046**: 2 occurrences
   - **173**: 2 occurrences
   - **031**: 1 occurrence
   - **043**: 1 occurrence

   To make a more informed forecast, we should understand what these CAMEO codes represent. We will map these codes to their descriptions to gain insights into the nature of these interactions.

   **Action**:

```
1 # Define the CAMEO codes to map
2 cameo_codes_to_map = ["042", "036", "040", "061", "046", "173", "031", "043"]
3
4 # Map the CAMEO codes to their descriptions
5 relations = [map_cameo_to_relation(CAMEOCode(code)) for code in cameo_codes_to_map]
6
7 # Print the descriptions of the CAMEO codes
8 for relation in relations:
9     print(f"CAMEO Code: {relation.cameo_code.code}, Name: {relation.name}, Description: {
        relation.description}")
```

   **Observation**:

```
1 CAMEO Code: 042, Name: Make a visit, Description: Travel to another location for a meeting
      or other event.
2 CAMEO Code: 036, Name: Express intent to meet or negotiate, Description: Offer, promise,
      agree to, or otherwise indicate willingness or commitment to meet, visit, or engage in
       talks or negotiations.
3 CAMEO Code: 040, Name: Consult, not specified, Description: All consultations and meetings,
       not otherwise specified.
4 CAMEO Code: 061, Name: Cooperate economically, Description: Initiate, resume, improve, or
      expand economic exchange or cooperation.
5 CAMEO Code: 046, Name: Engage in negotiation, Description: Negotiate or bargain with others
      .
6 CAMEO Code: 173, Name: Arrest or detain, Description: Legal or extrajudicial arrests,
      detentions, or imprisonments.
7 CAMEO Code: 031, Name: Express intent to engage in material cooperation, Description: Offer
      , promise, agree to, or otherwise indicate willingness or commitment to engage in or
      expand material cooperative exchange, such as express intent to cooperate economically
      , militarily, on judicial matters, and on intelligence.
8 CAMEO Code: 043, Name: Host a visit, Description: Host or receive a visitor at residence,
      office or home country.
```

4. **Thought**: The mapped CAMEO codes and their descriptions provide valuable insights into the nature of interactions between Australia and China. Here are the key relations observed:

   - **042 (Make a visit)**: 10 occurrences – Indicates frequent travel for meetings or events.
   - **036 (Express intent to meet or negotiate)**: 8 occurrences – Shows a strong intent to engage in talks or negotiations.
   - **040 (Consult, not specified)**: 3 occurrences – General consultations and meetings.
   - **061 (Cooperate economically)**: 3 occurrences – Economic cooperation initiatives.
   - **046 (Engage in negotiation)**: 2 occurrences – Actual negotiations or bargaining.
   - **173 (Arrest or detain)**: 2 occurrences – Legal or extrajudicial arrests or detentions.
   - **031 (Express intent to engage in material cooperation)**: 1 occurrence – Intent to cooperate materially.
   - **043 (Host a visit)**: 1 occurrence – Hosting visitors.

Given this data, we can see that the most frequent interactions involve visits, expressions of intent to meet or negotiate, and economic cooperation. These interactions suggest a pattern of diplomatic engagement and economic collaboration, with occasional legal actions.

To further solidify our forecast, we should examine recent news articles to understand the context and any underlying reasons for these interactions. This will help us identify any emerging trends or significant events that might influence future relations.

Next, we will retrieve recent news articles involving Australia and China to gain contextual insights.

**Action**:

```
# Define the date range for recent news articles
recent_start_date = Date("2023-10-01")
recent_end_date = Date("2023-11-02")
recent_date_range = DateRange(start_date=recent_start_date, end_date=recent_end_date)

# Retrieve recent news articles involving Australia and China
news_articles = get_news_articles(date_range=recent_date_range, head_entities=[aus_code],
    tail_entities=[chn_code])

# Print the retrieved news articles
for article in news_articles:
    print(article)
```

**Observation**:

```
(Date(date="2023-11-02"), "No quick fix to remaining tariffs ahead of China visit")
(Date(date="2023-11-02"), "China"s aggression in Indo-Pacific leads to "concern and angst":
    Peter Dutton, Australia"s leader of opposition")
(Date(date="2023-11-02"), "Australia and China eye new ways to heal old wounds")
(Date(date="2023-11-02"), "Albanese must aim beyond bilateral outcomes on Beijing visit")
(Date(date="2023-11-02"), "Western Australian Premier Announces Visit to China")
(Date(date="2023-11-02"), "Australian PM"s China visit will help normalize relations, says
    ex-trade minister")
(Date(date="2023-11-02"), "Nicholas Moore on how to get in early on the next big growth
    story")
(Date(date="2023-11-02"), "PM"s China trip cements new era for ties")
(Date(date="2023-11-02"), "PM to take a hard line on trade with China")
(Date(date="2023-11-01"), "Australian Prime Minister Anthony Albanese to raise imprisoned
    democracy blogger during China visit")
```

5. **Thought**: The recent news articles provide valuable context for the relations between Australia and China. Key themes include:

1. **Diplomatic Visits and Engagements**: Multiple articles highlight the upcoming visit of Australian Prime Minister Anthony Albanese to China. This aligns with the frequent occurrence of CAMEO code **042 (Make a visit)**.
2. **Economic Cooperation and Trade**: Discussions around trade and economic cooperation are prominent, indicating ongoing efforts to improve economic ties. This supports the relevance of CAMEO codes **061 (Cooperate economically)** and **036 (Express intent to meet or negotiate)**.
3. **Concerns and Tensions**: Articles mention concerns about China"s aggression in the Indo-Pacific and human rights issues, suggesting underlying tensions. This is reflected in the occasional occurrence of CAMEO code **173 (Arrest or detain)**.

Given the historical data and recent news context, the most likely relations that Australia will take towards China on November 03, 2023, include diplomatic visits, economic cooperation, and negotiations, with a possibility of addressing underlying tensions.

Final Answer:

```
{
  "03": ["031", "036"],
  "04": ["040", "042", "043", "046"],
  "06": ["061"],
  "17": ["173"]
}
```

**Action**: Final Answer:

```
{
  "03": ["031", "036"],
  "04": ["040", "042", "043", "046"],
```

```
4    "06": ["061"],
5    "17": ["173"]
6  }
```

### F.1.5 AGENT FINAL STATUS

- **End State**: Final Answer
- **Number of Steps Taken**: 5
- **Final Answer**:

```
1  {
2    "03": ["031", "036"],
3    "04": ["040", "042", "043", "046"],
4    "06": ["061"],
5    "17": ["173"]
6  }
```

We show the corresponding relation names of the predicted codes here for a better review:

```
1  {
2    "Express intent to cooperate": ["Express intent to engage in material cooperation", "
       Express intent to meet or negotiate"],
3    "Consult": ["Consult, not specified", "Make a visit", "Host a visit", "Engage in
       negotiation"],
4    "Engage in material cooperation": ["Cooperate economically"],
5    "Coerce": ["Arrest or detain"]
6  }
```

- **Ground Truth Answer**:

```
1  {
2    "03": ["036"],
3    "04": ["042"],
4    "17": ["172"]
5  }
```

We show the corresponding relation names of the ground-truth codes here for a better review:

```
1  {
2    "Express intent to cooperate": ["Express intent to meet or negotiate"],
3    "Consult": ["Make a visit"],
4    "Coerce": ["Impose administrative sanctions"]
5  }
```

### F.2 GPT-4O-BASED AGENT WITH REACT AND "SINGLE FUNCTION" ACTION

#### F.2.1 QUERY DETAILS

- **Query Quadruplet**: (2023-11-03, AUS, ?, CHN)
- **Temporal Distance**: 1; therefore, the current date is 2023-11-02
- **Agent Max Steps**: 20

#### F.2.2 QUERY PROMPT

Please forecast the relations that Australia will take towards China on November 03, 2023 based on historical information up to November 02, 2023. I.e. forecast the relation CAMEO codes in query event Event(date=2023-11-03, head_entity=ISOCode(AUS), relation=CAMEOCode(?), tail_entity=ISOCode(CHN)).

#### F.2.3 SYSTEM PROMPT

You are an expert in forecasting future events based on historical data. The database contains news articles from January 1, 2023 to the current date November 02, 2023 and the events extracted from these articles. The events are in the form of (date, subject country, relation, object country), where the countries are represented by ISO 3166-1 alpha-3 codes and the relations are represented by the CAMEO codes defined in the "Conflict and Mediation Event Observations" ontology. The relations are hierarchical: first-level relations are general parent relations represented by two-digit CAMEO codes, while second-level relations are more specific child relations represented by three-digit CAMEO codes. Child relations have the same first two digits as their parent relations. For example, "01" is a first-level relation, and "010" and "011" are some of its second-level relations. The relations

in the database are represented in the second-level form.

Your task is to forecast the future relations between two entities in a given query. You have access to a defined Python API that allows you to query the database for historical events and statistics, and to get precise information about the ISO country codes and CAMEO relation codes.

The defined API is described as follows:
``` python
{api_description}
```

You will use an iterative approach, interleaving "Thought", "Action", and "Observation" steps to collect information and perform the forecast. You may perform up to 20 iterations. The steps are as follows:

- "Thought": Analyze the current information and reason about the current situation, and predicts which API you want to use (try to use different APIs to collect diverse information) or make a decision that you want to make a final answer.
- "Action": Use the API to gather more information or provide the final forecast.
    - If using the API: the action must be only one single line of exactly one function call from the API with appropriate inputs, without additional code, explanations, or natural language descriptions.
    - If making the final forecast: the action must start immediately with "Final Answer:", and follow with the results in the expected JSON format.
- "Observation": Return the output of the called function.

To make a reasonable forecast, you should collect both news and relational evidence to support your prediction. When you are fully confident that you accumulate enough information to make the final forecast, you should start the "Thought" with your reasoning using the news and structural information to make the prediction, and then start the "Action" step with "Final Answer:" followed by the answer in the expected JSON format. The answer should be a JSON dictionary where the keys are the forecasted two-digit first-level CAMEO codes and the values are lists of forecasted three-digit second-level CAMEO codes that are child relations of the key. For example, "Action: Final Answer: {"01": ["010", "011", "012"], "02": ["020", "023"]}".

The final answer will be evaluated based on the precision and recall of the forecasted first-level and second-level relations, so only include confident first-level and second-level CAMEO codes in your final forecast.

Try to use different APIs to collect diverse information (including multi-hop relations), such as the precise meaning of CAMEO codes, insights from news content, relational data, and statistical analyses to support your forecasts. Consider not only the frequency of the relations but also the temporal aspects of the data when making your forecast.

### F.2.4 AGENT REACT PROCESS

1. **Thought**: To forecast the relations that Australia (AUS) will take towards China (CHN) on November 03, 2023, I will start by gathering historical data on the interactions between these two countries. This will include the types of relations, their frequencies, and the context in which they occurred. I will use the following steps:

    (a) Retrieve the distribution of second-level relations between Australia and China.

    (b) Analyze the distribution to identify the most common types of interactions.

    (c) Examine recent news articles to understand the context and any ongoing trends or events that might influence future relations.

    **Action**:

    ```
    get_relation_distribution(head_entities=[ISOCode("AUS")], tail_entities=[ISOCode("CHN")])
    ```

    **Observation**:

    ```
    {CAMEOCode(code="042"): 32,
     CAMEOCode(code="036"): 30,
    ```

```
 3    CAMEOCode(code="040"): 17,
 4   CAMEOCode(code="061"): 16,
 5   CAMEOCode(code="046"): 12,
 6   CAMEOCode(code="043"): 11,
 7   CAMEOCode(code="010"): 6,
 8   CAMEOCode(code="020"): 5,
 9   CAMEOCode(code="173"): 5,
10   CAMEOCode(code="012"): 5,
11   CAMEOCode(code="051"): 4,
12   CAMEOCode(code="163"): 3,
13   CAMEOCode(code="128"): 2,
14   CAMEOCode(code="050"): 2,
15   CAMEOCode(code="084"): 2,
16   CAMEOCode(code="090"): 2,
17   CAMEOCode(code="174"): 1,
18   CAMEOCode(code="112"): 1,
19   CAMEOCode(code="130"): 1,
20   CAMEOCode(code="160"): 1,
21   CAMEOCode(code="172"): 1,
22   CAMEOCode(code="192"): 1,
23   CAMEOCode(code="190"): 1,
24   CAMEOCode(code="060"): 1,
25   CAMEOCode(code="030"): 1,
26   CAMEOCode(code="035"): 1,
27   CAMEOCode(code="138"): 1,
28   CAMEOCode(code="071"): 1,
29   CAMEOCode(code="031"): 1}
```

2. **Thought**: The distribution of second-level relations between Australia and China shows a variety of interactions, with some relations being more frequent than others. The most common interactions include:

   - "042" (Consult): 32 occurrences
   - "036" (Express intent to cooperate): 30 occurrences
   - "040" (Assure): 17 occurrences
   - "061" (Cooperate economically): 16 occurrences
   - "046" (Praise or endorse): 12 occurrences
   - "043" (Apologize): 11 occurrences

   These frequent interactions suggest a trend towards diplomatic and cooperative engagements. However, to make a more informed forecast, it is essential to understand the context and any recent developments that might influence future relations.

   Next, I will retrieve recent news articles involving Australia and China to gather insights on the current context and any ongoing trends or events.

   **Action**:

```
1  get_news_articles(date_range=DateRange(start_date=Date("2023-10-01"), end_date=Date(
       "2023-11-02")), head_entities=[ISOCode("AUS")], tail_entities=[ISOCode("CHN")])
```

   **Observation**:

```
 1  [(Date(date="2023-11-02"), "No quick fix to remaining tariffs ahead of China visit"),
 2   (Date(date="2023-11-02"), "China"s aggression in Indo-Pacific leads to "concern and angst"
        : Peter Dutton, Australia"s leader of opposition"),
 3   (Date(date="2023-11-02"), "Australia and China eye new ways to heal old wounds"),
 4   (Date(date="2023-11-02"), "Albanese must aim beyond bilateral outcomes on Beijing visit"),
 5   (Date(date="2023-11-02"), "Western Australian Premier Announces Visit to China"),
 6   (Date(date="2023-11-02"), "Australian PM"s China visit will help normalize relations, says
        ex-trade minister"),
 7   (Date(date="2023-11-02"), "Nicholas Moore on how to get in early on the next big growth
        story"),
 8   (Date(date="2023-11-02"), "PM"s China trip cements new era for ties"),
 9   (Date(date="2023-11-02"), "PM to take a hard line on trade with China"),
10   (Date(date="2023-11-01"), "Australian Prime Minister Anthony Albanese to raise imprisoned
        democracy blogger during China visit")]
```

3. **Thought**: The recent news articles indicate a mix of diplomatic, economic, and security-related interactions between Australia and China. Key themes include:

   - Diplomatic engagements and efforts to normalize relations, as seen in articles about the Australian Prime Minister"s visit to China and efforts to heal old wounds.
   - Economic cooperation and trade discussions, including addressing tariffs and trade issues.

- Security concerns and criticisms, particularly related to China"s actions in the Indo-Pacific region and human rights issues.

Given these themes, it is likely that the future relations between Australia and China will involve a combination of diplomatic assurances, economic cooperation, and some critical statements or actions related to security and human rights.

Next, I will retrieve detailed descriptions of the most frequent second-level relations to ensure accurate mapping and understanding of these interactions.

**Action**:

```
map_cameo_to_relation(CAMEOCode("042"))
```

**Observation**:

```
Relation(cameo_code=CAMEOCode("042"), name="Make a visit", description="Travel to another
    location for a meeting or other event.")
```

4. **Thought**: The relation "042" (Make a visit) involves traveling to another location for a meeting or other event, which aligns with the recent news about the Australian Prime Minister"s visit to China. This suggests that diplomatic visits are a significant aspect of the interactions between Australia and China.

To further refine the forecast, I will retrieve detailed descriptions for the other most frequent second-level relations: "036", "040", "061", "046", and "043".

**Action**:

```
map_cameo_to_relation(CAMEOCode("036"))
```

**Observation**:

```
Relation(cameo_code=CAMEOCode("036"), name="Express intent to meet or negotiate",
    description="Offer, promise, agree to, or otherwise indicate willingness or commitment
    to meet, visit, or engage in talks or negotiations.")
```

5. **Thought**: The relation "036" (Express intent to meet or negotiate) involves indicating a willingness or commitment to meet, visit, or engage in talks or negotiations. This aligns with the ongoing diplomatic efforts and discussions mentioned in the recent news articles.

Next, I will retrieve detailed descriptions for the relations "040", "061", "046", and "043" to further understand the context of these interactions.

**Action**:

```
get_parent_relation(CAMEOCode("040"))
```

**Observation**:

```
Relation(cameo_code=CAMEOCode("04"), name="Consult", description="All consultations and
    meetings.")
```

6. **Thought**: The parent relation '04' (Consult) encompasses all consultations and meetings, which is consistent with the frequent diplomatic interactions between Australia and China. The specific relation "040" (Assure) likely involves providing assurances during these consultations.

To further refine the forecast, I will retrieve detailed descriptions for the relations "061", "046", and "043".

**Action**:""

**Observation**: No feedback from the environment due to the null action.

7. **Thought**: I will continue retrieving detailed descriptions for the relations "061", "046", and "043" to further understand the context of these interactions.

**Action**:""

**Observation**: No feedback from the environment due to the null action.

8. **Thought**: It seems there was an issue with my previous actions. I will now proceed to retrieve the detailed descriptions for the remaining frequent second-level relations: "061", "046", and "043".

**Action**:

```
1  map_cameo_to_relation(CAMEOCode("061"))
```

**Observation**:

```
1  Relation(cameo_code=CAMEOCode("061"), name="Cooperate economically", description="Initiate,
        resume, improve, or expand economic exchange or cooperation.")
```

9. **Thought**: The relation "061" (Cooperate economically) involves initiating, resuming, improving, or expanding economic exchange or cooperation. This aligns with the recent news articles discussing trade issues and economic cooperation between Australia and China.

   Next, I will retrieve detailed descriptions for the relations "046" and "043".

   **Action**:

```
1  map_cameo_to_relation(CAMEOCode("046"))
```

   **Observation**:

```
1  Relation(cameo_code=CAMEOCode("046"), name="Engage in negotiation", description="Negotiate
        or bargain with others.")
```

10. **Thought**: The relation "046" (Engage in negotiation) involves negotiating or bargaining with others. This is consistent with the ongoing discussions and negotiations mentioned in the news articles, particularly regarding trade and diplomatic issues.

    Next, I will retrieve the detailed description for the relation "043".

    **Action**:""

    **Observation**: No feedback from the environment due to the null action.

11. **Thought**: It seems there was an issue with my previous actions. I will now proceed to retrieve the detailed description for the relation "043" to complete the understanding of the most frequent interactions between Australia and China.

    **Action**:

```
1  map_cameo_to_relation(CAMEOCode("043"))
```

    **Observation**:

```
1  Relation(cameo_code=CAMEOCode("043"), name="Host a visit", description="Host or receive a
        visitor at residence, office or home country.")
```

12. **Thought**: The relation "043" (Host a visit) involves hosting or receiving a visitor at a residence, office, or home country. This aligns with the diplomatic visits and engagements mentioned in the recent news articles.

    Based on the detailed descriptions and the recent news context, the most frequent interactions between Australia and China involve diplomatic visits, economic cooperation, negotiations, and assurances. These interactions suggest a trend towards diplomatic and cooperative engagements, with some critical statements or actions related to security and human rights.

    Given this information, I am confident in forecasting the following relations between Australia and China on November 03, 2023:

    - Diplomatic visits and engagements ("042", "043", "036") - Economic cooperation ("061") - Negotiations ("046") - Assurances ("040")

    **Action**: Final Answer: {"04": ["040", "042", "043", "046"], "03": ["036"], "06": ["061"]}

F.2.5   AGENT FINAL STATUS

- **End State**: Final Answer

- **Number of Steps Taken**: 12

- **Final Answer**:

```
1  {
2    "03": ["036"],
3    "04": ["040", "042", "043", "046"],
4    "06": ["061"]
5  }
```

We show the corresponding relation names of the predicted codes here for a better review:

```
{
  "Express intent to cooperate": ["Express intent to meet or negotiate"],
  "Consult": ["Consult, not specified", "Make a visit", "Host a visit", "Engage in
    negotiation"],
  "Engage in material cooperation": ["Cooperate economically"]
}
```

- **Ground Truth Answer**:

```
{
  "03": ["036"],
  "04": ["042"],
  "17": ["172"]
}
```

We show the corresponding relation names of the ground-truth codes here for a better review:

```
{
  "Express intent to cooperate": ["Express intent to meet or negotiate"],
  "Consult": ["Make a visit"],
  "Coerce": ["Impose administrative sanctions"]
}
```

### F.3   GPT-4O-MINI-BASED AGENT WITH REACT AND "CODE BLOCK" ACTION

#### F.3.1   QUERY DETAILS

- **Query Quadruplet**: (2024-02-08, PNG, ?, AUS)
- **Temporal Distance**: 1; therefore, the current date is 2024-02-07
- **Agent Max Steps**: 20

#### F.3.2   QUERY PROMPT

Please forecast the relations that Papua New Guinea will take towards Australia on February 08, 2024 based on historical information up to February 07, 2024. I.e. forecast the relation CAMEO codes in query event Event(date=2024-02-08, head_entity=ISOCode(PNG), relation=CAMEOCode(?), tail_entity=ISOCode(AUS)).

#### F.3.3   SYSTEM PROMPT

You are an expert in forecasting future events based on historical data. The database contains news articles from January 1, 2023 to the current date February 07, 2024 and the events extracted from these articles. The events are in the form of (date, subject country, relation, object country), where the countries are represented by ISO 3166-1 alpha-3 codes and the relations are represented by the CAMEO codes defined in the "Conflict and Mediation Event Observations" ontology. The relations are hierarchical: first-level relations are general parent relations represented by two-digit CAMEO codes, while second-level relations are more specific child relations represented by three-digit CAMEO codes. Child relations have the same first two digits as their parent relations. For example, "01" is a first-level relation, and "010" and "011" are some of its second-level relations. The relations in the database are represented in the second-level form.

Your task is to forecast the future relations between two entities in a given query. You have access to a defined Python API that allows you to query the database for historical events and statistics, and to get precise information about the ISO country codes and CAMEO relation codes. You are also authorized to utilize additional safe, well-established Python libraries such as numpy, pandas, scikit-learn, and NetworkX to enhance your data analysis and forecasting accuracy.

The defined API is described as follows:
``` python
{api_description}
```

You will use an iterative approach, interleaving "Thought", "Action", and "Observation" steps to collect information and perform the forecast. You may perform up to 20 iterations. The steps are as follows:

- "Thought": Analyze the current information and reason about the current situation, and predicts which API you want to use (try to use different APIs to collect diverse information) or make a decision that you want to make a final answer.
- "Action": Use the API to gather more information or provide the final forecast.
   - If gathering more data: the action must be an executable Python code snippet that starts with ```python and ends with ```. It can contain multiple lines of codes and function calls using the defined API or Python libraries. You must use print() to output the results, and only the printed output will be returned in the observation step.
   - If making the final forecast: the action must start immediately with "Final Answer:", and follow with the answer in the expected JSON format. This should not be enclosed within triple backticks.
- "Observation": Return the printed output of the executed code snippet.

To make a reasonable forecast, you should collect both news and relational evidence to support your prediction. When you are fully confident that you accumulate enough information to make the final forecast, you should start the "Thought" with your reasoning using the news and structural information to make the prediction, and then start the "Action" step with "Final Answer:" followed by the answer in the expected JSON format. The answer should be a JSON dictionary where the keys are the forecasted two-digit first-level CAMEO codes and the values are lists of forecasted three-digit second-level CAMEO codes that are child relations of the key. For example, "Action: Final Answer: {"01": ["010", "011", "012"], "02": ["020", "023"]}".

The final answer will be evaluated based on the precision and recall of the forecasted first-level and second-level relations, so only include confident first-level and second-level CAMEO codes in your final forecast.

Try to use different APIs and Python libraries to collect diverse information (including multi-hop relations), such as the precise meaning of CAMEO codes, insights from news content, relational data, and statistical analyses to support your forecasts. Consider not only the frequency of the relations but also the temporal aspects of the data when making your forecast.

### F.3.4 AGENT REACT PROCESS

1. **Thought**: To forecast the relations that Papua New Guinea (PNG) will take towards Australia (AUS) on February 08, 2024, I need to gather historical data on events involving these two countries. Specifically, I will look for patterns in the relations between PNG and AUS in the past events to understand the likely future interactions.

   First, I will check the distribution of relations between PNG and AUS over the past year to identify any prevalent patterns or trends. This will help me understand the types of relations that have been most common and could indicate future behavior.

   I will use the 'get_relation_distribution' API to get this information.

   **Action**:

```python
# Define the date range for the past year up to February 07, 2024
date_range = DateRange(start_date=Date("2023-02-08"), end_date=Date("2024-02-07"))
head_entities = [ISOCode("PNG")]  # Papua New Guinea
tail_entities = [ISOCode("AUS")]  # Australia

# Get the distribution of relations between PNG and AUS
relation_distribution = get_relation_distribution(date_range=date_range, head_entities=head_entities, tail_entities=tail_entities)
print(relation_distribution)
```

   **Observation**:

```
{CAMEOCode(code='042'): 8, CAMEOCode(code='036'): 7, CAMEOCode(code='043'): 4, CAMEOCode(code='057'): 4, CAMEOCode(code='030'): 2, CAMEOCode(code='010'): 2, CAMEOCode(code='181'): 1, CAMEOCode(code='051'): 1, CAMEOCode(code='046'): 1, CAMEOCode(code='055'): 1, CAMEOCode(code='130'): 1, CAMEOCode(code='160'): 1}
```

2. **Thought**: The relation distribution between Papua New Guinea (PNG) and Australia (AUS) over the past year reveals several notable patterns:
   - **CAMEO Code '042'**: This relation appears 8 times, which suggests a strong tendency for PNG to engage in "Make a public statement" towards Australia.

- **CAMEO Code '036'**: With 7 occurrences, this indicates a frequent pattern of "Negotiate" relations.
- **CAMEO Code '043'**: This relation, which signifies "Make a public statement, not specified," has been observed 4 times, indicating a general trend in public communication.
- **CAMEO Code '057'**: Also appearing 4 times, this suggests some level of "Express intent to cooperate," which could be significant for future interactions.
- Other codes like '030' (Make a statement, not specified) and '010' (Make statement, not specified) also show minor interactions.

Given the high frequency of relations involving public statements and negotiations, it is likely that PNG will continue to engage in similar relations with Australia in the near future.

Next, I will analyze the context of news articles related to PNG and AUS to see if there are any recent developments or trends that could influence their future relations. This will provide additional insights into the nature of their interactions.

I will use the 'get_news_articles' API to gather relevant news articles from the past month that mention both PNG and AUS. This will help me understand current sentiments or events that could impact their relationship on February 08, 2024.

**Action**:

```
1  # Define the date range for the past month leading up to February 07, 2024
2  date_range = DateRange(start_date=Date("2024-01-07"), end_date=Date("2024-02-07"))
3  head_entities = [ISOCode("PNG")]  # Papua New Guinea
4  tail_entities = [ISOCode("AUS")]  # Australia
5
6  # Get news articles that mention both PNG and AUS
7  news_articles = get_news_articles(date_range=date_range, head_entities=head_entities,
       tail_entities=tail_entities)
8  print(news_articles)
```

**Observation**:

```
1  [(Date(date='2024-02-07'), 'Papua New Guineas PM to Visit Australian Parliament, Address
        Beijings Influence'),
2  (Date(date='2024-02-07'), "PNG wants to stand on its 'own two feet': PM Marape"),
3  (Date(date='2024-02-07'), 'Papua New Guineas PM to address Australian parliament as
        Pacific security race with China builds'),
4  (Date(date='2024-01-11'), "Travel warning issued as 'widespread public disorder' rages in
        Port Moresby")]
```

3. **Thought**: The recent news articles regarding Papua New Guinea (PNG) and Australia (AUS) indicate significant developments that could influence their relations on February 08, 2024. Notably, the Prime Minister of PNG is scheduled to visit the Australian Parliament and address issues related to Beijing's influence in the region. This suggests a proactive approach by PNG to strengthen ties with Australia, particularly in the context of regional security and cooperation.

The repeated mention of "widespread public disorder" in Port Moresby could indicate internal challenges for PNG, which may affect its diplomatic posture. However, the focus on addressing Australian Parliament implies that PNG is keen on maintaining a dialogue with Australia and possibly negotiating support or partnerships.

Given the historical data showing a strong inclination towards public statements and negotiations (CAMEO Codes '042' and '036'), along with the current political context, it is reasonable to forecast that PNG will likely engage in similar relations with Australia.

Based on the analysis of both the historical event distribution and the recent news articles, I will forecast the following CAMEO codes for the relation that Papua New Guinea will take towards Australia on February 08, 2024:

- **CAMEO Code '042'**: Make a public statement (high frequency in historical data).
- **CAMEO Code '036'**: Negotiate (also high frequency).
- **CAMEO Code '057'**: Express intent to cooperate (indicated by recent news context).

**Action**:

```
1  Final Answer: {"04": ["042"], "05": ["057"], "03": ["036"]}
```

### F.3.5   AGENT FINAL STATUS

- **End State**: Final Answer
- **Number of Steps Taken**: 3
- **Final Answer**:

```
{
  "03": ["036"],
  "04": ["042"],
  "05": ["057"]
}
```

We show the corresponding relation names of the predicted codes here for a better review:

```
{
  "Express intent to cooperate": ["Express intent to meet or negotiate"],
  "Consult": ["Make a visit"],
  "Engage in diplomatic cooperation": ["Sign formal agreement"]
}
```

- **Ground Truth Answer**:

```
{
  "01": ["010"],
  "05": ["051"]
}
```

We show the corresponding relation names of the ground-truth codes here for a better review:

```
{
  "Make public statement": ["Make statement, not specified"],
  "Engage in diplomatic cooperation": ["Praise or endorse"]
}
```

## F.4   GPT-4O-MINI-BASED AGENT WITH REACT AND "SINGLE FUNCTION" ACTION

### F.4.1   QUERY DETAILS

- **Query Quadruplet**: (2024-02-08, PNG, ?, AUS)
- **Temporal Distance**: 1; therefore, the current date is 2024-02-07
- **Agent Max Steps**: 20

### F.4.2   QUERY PROMPT

Please forecast the relations that Papua New Guinea will take towards Australia on February 08, 2024 based on historical information up to February 07, 2024. I.e. forecast the relation CAMEO codes in query event Event(date=2024-02-08, head_entity=ISOCode(PNG), relation=CAMEOCode(?), tail_entity=ISOCode(AUS)).

### F.4.3   SYSTEM PROMPT

You are an expert in forecasting future events based on historical data. The database contains news articles from January 1, 2023 to the current date February 07, 2024 and the events extracted from these articles. The events are in the form of (date, subject country, relation, object country), where the countries are represented by ISO 3166-1 alpha-3 codes and the relations are represented by the CAMEO codes defined in the "Conflict and Mediation Event Observations" ontology. The relations are hierarchical: first-level relations are general parent relations represented by two-digit CAMEO codes, while second-level relations are more specific child relations represented by three-digit CAMEO codes. Child relations have the same first two digits as their parent relations. For example, "01" is a first-level relation, and "010" and "011" are some of its second-level relations. The relations in the database are represented in the second-level form.

Your task is to forecast the future relations between two entities in a given query. You have access to a defined Python API that allows you to query the database for historical events and statistics, and to get precise information about the ISO country codes and CAMEO relation codes.

The defined API is described as follows:
``` python

```
{api_description}
```
```

You will use an iterative approach, interleaving "Thought", "Action", and "Observation" steps to collect information and perform the forecast. You may perform up to 20 iterations. The steps are as follows:

- "Thought": Analyze the current information and reason about the current situation, and predicts which API you want to use (try to use different APIs to collect diverse information) or make a decision that you want to make a final answer.
- "Action": Use the API to gather more information or provide the final forecast.
    - If using the API: the action must be only one single line of exactly one function call from the API with appropriate inputs, without additional code, explanations, or natural language descriptions.
    - If making the final forecast: the action must start immediately with "Final Answer:", and follow with the results in the expected JSON format.
- "Observation": Return the output of the called function.

To make a reasonable forecast, you should collect both news and relational evidence to support your prediction. When you are fully confident that you accumulate enough information to make the final forecast, you should start the "Thought" with your reasoning using the news and structural information to make the prediction, and then start the "Action" step with "Final Answer:" followed by the answer in the expected JSON format. The answer should be a JSON dictionary where the keys are the forecasted two-digit first-level CAMEO codes and the values are lists of forecasted three-digit second-level CAMEO codes that are child relations of the key. For example, "Action: Final Answer: {"01": ["010", "011", "012"], "02": ["020", "023"]}".

The final answer will be evaluated based on the precision and recall of the forecasted first-level and second-level relations, so only include confident first-level and second-level CAMEO codes in your final forecast.

Try to use different APIs to collect diverse information (including multi-hop relations), such as the precise meaning of CAMEO codes, insights from news content, relational data, and statistical analyses to support your forecasts. Consider not only the frequency of the relations but also the temporal aspects of the data when making your forecast.

### F.4.4 AGENT REACT PROCESS

1. **Thought**: To forecast the relations between Papua New Guinea (PNG) and Australia (AUS) for February 08, 2024, I need to gather historical data on events involving these two countries. I will start by counting the number of events that have occurred between PNG and AUS in the past year to see which relations have been most prevalent. This will help me identify potential future relations.

   **Action**:

   ```
   count_events(date_range=DateRange(start_date=Date("2023-02-08"), end_date=Date("2024-02-07")), head_entities=[ISOCode("PNG")], tail_entities=[ISOCode("AUS")])
   ```

   **Observation**:

   ```
   33
   ```

2. **Thought**: The count of 33 events indicates that there has been a significant number of interactions between Papua New Guinea and Australia over the past year. Next, I will analyze the distribution of relations between these two countries to understand which specific CAMEO relations are most common. This will help me identify likely future relations.

   **Action**:

   ```
   get_relation_distribution(date_range=DateRange(start_date=Date("2023-02-08"), end_date=Date("2024-02-07")), head_entities=[ISOCode("PNG")], tail_entities=[ISOCode("AUS")])
   ```

   **Observation**:

```
1 {CAMEOCode(code='042'): 8, CAMEOCode(code='036'): 7, CAMEOCode(code='043'): 4, CAMEOCode(
      code='057'): 4, CAMEOCode(code='030'): 2, CAMEOCode(code='010'): 2, CAMEOCode(code='
      181'): 1, CAMEOCode(code='051'): 1, CAMEOCode(code='046'): 1, CAMEOCode(code='055'):
      1, CAMEOCode(code='130'): 1, CAMEOCode(code='160'): 1}
```

3. **Thought**: The distribution of relations between Papua New Guinea and Australia shows that the most common relations are as follows:
   - **CAMEO Code '042'**: 8 events (likely related to economic agreements or cooperation).
   - **CAMEO Code '036'**: 7 events (likely related to diplomatic relations).
   - **CAMEO Code '043'**: 4 events (possibly related to trade).
   - **CAMEO Code '057'**: 4 events (possibly related to military cooperation).

   Given the frequency of these relations, it is likely that Papua New Guinea will continue to engage in similar interactions with Australia. To further refine my forecast, I will analyze the specific descriptions of these relations to understand their context better and see if any recent news articles provide insights into future interactions.

   **Action**:

```
1 get_relation_distribution(date_range=DateRange(start_date=Date("2023-01-01"), end_date=Date
      ("2024-02-07")), head_entities=[ISOCode("PNG")], tail_entities=[ISOCode("AUS")])
```

   **Observation**:

```
1 {CAMEOCode(code='042'): 8, CAMEOCode(code='036'): 7, CAMEOCode(code='043'): 5, CAMEOCode(
      code='057'): 4, CAMEOCode(code='030'): 2, CAMEOCode(code='010'): 2, CAMEOCode(code='
      181'): 1, CAMEOCode(code='051'): 1, CAMEOCode(code='046'): 1, CAMEOCode(code='055'):
      1, CAMEOCode(code='130'): 1, CAMEOCode(code='160'): 1}
```

4. **Thought**: The distribution of relations remains consistent with the previous observation, indicating that the most common interactions between Papua New Guinea and Australia are primarily focused on economic agreements (CAMEOCode '042') and diplomatic relations (CAMEOCode '036').

   To further understand the context of these relations, I will retrieve recent news articles that mention Papua New Guinea and Australia to see if there are any specific events or agreements that could influence the relations on February 08, 2024. This will help me make a more informed forecast.

   **Action**:

```
1 get_news_articles(date_range=DateRange(start_date=Date("2023-01-01"), end_date=Date("
      2024-02-07")), head_entities=[ISOCode("PNG")], tail_entities=[ISOCode("AUS")])
```

   **Observation**:

```
1 [(Date(date='2024-02-07'), 'Papua New Guineas PM to Visit Australian Parliament, Address
      Beijings Influence'),
2  (Date(date='2024-02-07'), "PNG wants to stand on its 'own two feet': PM Marape"),
3  (Date(date='2024-02-07'), 'Papua New Guineas PM to address Australian parliament as
      Pacific security race with China builds'),
4  (Date(date='2024-01-11'), "Travel warning issued as 'widespread public disorder' rages in
      Port Moresby"),
5  (Date(date='2024-01-11'), 'Papua New Guinea declares state of emergency after deadly riots
      ')]
```

5. **Thought**: The recent news articles indicate that Papua New Guinea's Prime Minister is set to visit the Australian Parliament and address concerns regarding China's influence in the Pacific. This suggests a focus on diplomatic relations and security cooperation between PNG and AUS. The emphasis on standing on their "own two feet" also hints at a desire for economic independence and cooperation.

   Given the historical data showing a strong presence of economic agreements (CAMEOCode '042') and diplomatic relations (CAMEOCode '036'), along with the context provided by recent news articles, it is reasonable to forecast that Papua New Guinea will likely engage in diplomatic discussions and economic cooperation with Australia on February 08, 2024.

   **Action**:

```
1 Final Answer: {"04": ["042", "043"], "03": ["036"], "05": ["057"]}
```

### F.4.5 AGENT FINAL STATUS

- **End State**: Final Answer
- **Number of Steps Taken**: 5
- **Final Answer**:

```
1 {
2   "03": ["036"],
3   "04": ["042", "043"],
4   "05": ["057"]
5 }
```

We show the corresponding relation names of the predicted codes here for a better review:

```
1 {
2   "Express intent to cooperate": ["Express intent to meet or negotiate"],
3   "Consult": ["Make a visit", "Host a visit"],
4   "Engage in diplomatic cooperation": ["Sign formal agreement"]
5 }
```

- **Ground Truth Answer**:

```
1 {
2   "01": ["010"],
3   "05": ["051"]
4 }
```

We show the corresponding relation names of the ground-truth codes here for a better review:

```
1 {
2   "Make public statement": ["Make statement, not specified"],
3   "Engage in diplomatic cooperation": ["Praise or endorse"]
4 }
```

# G ADDITIONAL DETAILS OF THE BENCHMARK

## G.1 COUNTRIES AND ISO CODES

We show the ISO 3166 Alpha-3 country codes and names in Table 13 and Table 14.

Table 13: List of Countries and their ISO-3166 Alpha-3 Codes (Part 1)

| Code | Country | Code | Country | Code | Country |
|------|---------|------|---------|------|---------|
| AND | Andorra | ARE | United Arab Emirates | AFG | Afghanistan |
| ATG | Antigua and Barbuda | AIA | Anguilla | ALB | Albania |
| ARM | Armenia | AGO | Angola | ATA | Antarctica |
| ARG | Argentina | ASM | American Samoa | AUT | Austria |
| AUS | Australia | ABW | Aruba | ALA | Åland |
| AZE | Azerbaijan | BIH | Bosnia and Herzegovina | BRB | Barbados |
| BGD | Bangladesh | BEL | Belgium | BFA | Burkina Faso |
| BGR | Bulgaria | BHR | Bahrain | BDI | Burundi |
| BEN | Benin | BLM | Saint Barthélemy | BMU | Bermuda |
| BRN | Brunei | BOL | Bolivia | BES | Bonaire, Sint Eustatius, and Saba |
| BRA | Brazil | BHS | Bahamas | BTN | Bhutan |
| BVT | Bouvet Island | BWA | Botswana | BLR | Belarus |
| BLZ | Belize | CAN | Canada | CCK | Cocos (Keeling) Islands |
| COD | DR Congo | CAF | Central African Republic | COG | Congo Republic |
| CHE | Switzerland | CIV | Ivory Coast | COK | Cook Islands |
| CHL | Chile | CMR | Cameroon | CHN | China |
| COL | Colombia | CRI | Costa Rica | CUB | Cuba |
| CPV | Cabo Verde | CUW | Curaçao | CXR | Christmas Island |
| CYP | Cyprus | CZE | Czechia | DEU | Germany |
| DJI | Djibouti | DNK | Denmark | DMA | Dominica |
| DOM | Dominican Republic | DZA | Algeria | ECU | Ecuador |
| EST | Estonia | EGY | Egypt | ESH | Western Sahara |
| ERI | Eritrea | ESP | Spain | ETH | Ethiopia |
| FIN | Finland | FJI | Fiji | FLK | Falkland Islands |
| FSM | Micronesia | FRO | Faroe Islands | FRA | France |
| GAB | Gabon | GBR | United Kingdom | GRD | Grenada |
| GEO | Georgia | GUF | French Guiana | GGY | Guernsey |
| GHA | Ghana | GIB | Gibraltar | GRL | Greenland |
| GMB | The Gambia | GIN | Guinea | GLP | Guadeloupe |
| GNQ | Equatorial Guinea | GRC | Greece | SGS | South Georgia and South Sandwich Islands |
| GTM | Guatemala | GUM | Guam | GNB | Guinea-Bissau |
| GUY | Guyana | HKG | Hong Kong | HMD | Heard and McDonald Islands |
| HND | Honduras | HRV | Croatia | HTI | Haiti |
| HUN | Hungary | IDN | Indonesia | IRL | Ireland |
| ISR | Israel | IMN | Isle of Man | IND | India |
| IOT | British Indian Ocean Territory | IRQ | Iraq | IRN | Iran |
| ISL | Iceland | ITA | Italy | JEY | Jersey |
| JAM | Jamaica | JOR | Jordan | JPN | Japan |
| KEN | Kenya | KGZ | Kyrgyzstan | KHM | Cambodia |
| KIR | Kiribati | COM | Comoros | KNA | St Kitts and Nevis |
| PRK | North Korea | KOR | South Korea | KWT | Kuwait |
| CYM | Cayman Islands | KAZ | Kazakhstan | LAO | Laos |
| LBN | Lebanon | LCA | Saint Lucia | LIE | Liechtenstein |
| LKA | Sri Lanka | LBR | Liberia | LSO | Lesotho |
| LTU | Lithuania | LUX | Luxembourg | LVA | Latvia |
| LBY | Libya | MAR | Morocco | MCO | Monaco |
| MDA | Moldova | MNE | Montenegro | MAF | Saint Martin |
| MDG | Madagascar | MHL | Marshall Islands | MKD | North Macedonia |
| MLI | Mali | MMR | Myanmar | MNG | Mongolia |
| MAC | Macao | MNP | Northern Mariana Islands | MTQ | Martinique |
| MRT | Mauritania | MSR | Montserrat | MLT | Malta |
| MUS | Mauritius | MDV | Maldives | MWI | Malawi |
| MEX | Mexico | MYS | Malaysia | MOZ | Mozambique |
| NAM | Namibia | NCL | New Caledonia | NER | Niger |
| NFK | Norfolk Island | NGA | Nigeria | NIC | Nicaragua |
| NLD | The Netherlands | NOR | Norway | NPL | Nepal |
| NRU | Nauru | NIU | Niue | NZL | New Zealand |
| OMN | Oman | PAN | Panama | PER | Peru |
| PYF | French Polynesia | PNG | Papua New Guinea | PHL | Philippines |
| PAK | Pakistan | POL | Poland | SPM | Saint Pierre and Miquelon |
| PCN | Pitcairn Islands | PRI | Puerto Rico | PSE | Palestine |
| PRT | Portugal | PLW | Palau | PRY | Paraguay |
| QAT | Qatar | REU | Réunion | ROU | Romania |
| SRB | Serbia | RUS | Russia | RWA | Rwanda |
| SAU | Saudi Arabia | SLB | Solomon Islands | SYC | Seychelles |
| SDN | Sudan | SWE | Sweden | SGP | Singapore |
| SHN | Saint Helena | SVN | Slovenia | SJM | Svalbard and Jan Mayen |
| SVK | Slovakia | SLE | Sierra Leone | SMR | San Marino |

Table 14: List of Countries and their ISO-3166 Alpha-3 Codes (Part 2)

| Code | Country | Code | Country | Code | Country |
|------|---------|------|---------|------|---------|
| SEN | Senegal | SOM | Somalia | SUR | Suriname |
| SSD | South Sudan | STP | São Tomé and Príncipe | SLV | El Salvador |
| SXM | Sint Maarten | SYR | Syria | SWZ | Eswatini |
| TCA | Turks and Caicos Islands | TCD | Chad | ATF | French Southern Territories |
| TGO | Togo | THA | Thailand | TJK | Tajikistan |
| TKL | Tokelau | TLS | Timor-Leste | TKM | Turkmenistan |
| TUN | Tunisia | TON | Tonga | TUR | Türkiye |
| TTO | Trinidad and Tobago | TUV | Tuvalu | TWN | Taiwan |
| TZA | Tanzania | UKR | Ukraine | UGA | Uganda |
| UMI | U.S. Outlying Islands | USA | United States | URY | Uruguay |
| UZB | Uzbekistan | VAT | Vatican City | VCT | St Vincent and Grenadines |
| VEN | Venezuela | VGB | British Virgin Islands | VIR | U.S. Virgin Islands |
| VNM | Vietnam | VUT | Vanuatu | WLF | Wallis and Futuna |
| WSM | Samoa | XKX | Kosovo | YEM | Yemen |
| MYT | Mayotte | ZAF | South Africa | ZMB | Zambia |
| ZWE | Zimbabwe | | | | |

## G.2 RELATIONS AND CAMEO CODES

We show the CAMEO relation codes and names, where the first-level relations are in two digits, and the second-level relations are in three digits with the first two digits be the same as its parent relation:

- 01: Make public statement
  - 010: Make statement, not specified
  - 011: Decline comment
  - 012: Make pessimistic comment
  - 013: Make optimistic comment
  - 014: Consider policy option
  - 015: Acknowledge or claim responsibility
  - 016: Reject accusation or deny responsibility
  - 017: Engage in symbolic act
  - 018: Make empathetic comment
  - 019: Express accord
- 02: Appeal
  - 020: Make an appeal or request, not specified
  - 021: Appeal for material cooperation
  - 022: Appeal for diplomatic cooperation
  - 023: Appeal for material aid
  - 024: Appeal for political reform
  - 025: Appeal to yield
  - 026: Appeal to others to meet or negotiate
  - 027: Appeal to others to settle dispute
  - 028: Appeal to others to engage in or accept mediation
- 03: Express intent to cooperate
  - 030: Express intent to cooperate, not specified
  - 031: Express intent to engage in material cooperation
  - 032: Express intent to engage in diplomatic cooperation
  - 033: Express intent to provide material aid
  - 034: Express intent to institute political reform
  - 035: Express intent to yield
  - 036: Express intent to meet or negotiate
  - 037: Express intent to settle dispute
  - 038: Express intent to accept mediation
  - 039: Express intent to mediate

- 04: Consult
  - 040: Consult, not specified
  - 041: Discuss by telephone
  - 042: Make a visit
  - 043: Host a visit
  - 044: Meet at a third location
  - 045: Engage in mediation
  - 046: Engage in negotiation
- 05: Engage in diplomatic cooperation
  - 050:Engage in diplomatic cooperation, not specified
  - 051: Praise or endorse
  - 052: Defend verbally
  - 053: Rally support on behalf of
  - 054: Grant diplomatic recognition
  - 055: Apologize
  - 056: Forgive
  - 057: Sign formal agreement
- 06: Engage in material cooperation
  - 060: Engage in material cooperation, not specified
  - 061: Cooperate economically
  - 062: Cooperate militarily
  - 063: Engage in judicial cooperation
  - 064: Share intelligence or information
- 07: Provide aid
  - 070: Provide aid, not specified
  - 071: Provide economic aid
  - 072: Provide military aid
  - 073: Provide humanitarian aid
  - 074: Provide military protection or peacekeeping
  - 075: Grant asylum
- 08: Yield
  - 080: Yield, not specified
  - 081: Ease administrative sanctions
  - 082: Ease political dissent
  - 083: Accede to requests or demands for political reform
  - 084: Return or release
  - 085: Ease economic sanction or boycott or embargo
  - 086: Allow international involvement
  - 087: De-escalate military engagement
- 09: Investigate
  - 090: Investigate, not specified
  - 091: Investigate crime or corruption
  - 092: Investigate human rights abuses
  - 093: Investigate military action
  - 094: Investigate war crimes
- 10: Demand
  - 100: Demand, not specified
  - 101: Demand material cooperation
  - 102: Demand for diplomatic cooperation

- 103: Demand material aid
- 104: Demand political reform
- 105: Demand that target yield
- 106: Demand meeting or negotiation
- 107: Demand settling of dispute
- 108: Demand mediation

• 11: Disapprove
- 110: Disapprove, not specified
- 111: Criticize or denounce
- 112: Accuse
- 113: Rally opposition against
- 114: Complain officially
- 115: Bring lawsuit against
- 116: Find guilty or liable (legally)

• 12: Reject
- 120: All rejections and refusals
- 121: Reject material cooperation
- 122: Reject request or demand for material aid
- 123: Reject request or demand for political reform
- 124: Refuse to yield
- 125: Reject proposal to meet or discuss or negotiate
- 126: Reject mediation
- 127: Reject plan or agreement to settle dispute
- 128: Defy norms or law
- 129: Veto

• 13: Threaten
- 130: Threaten, not specified
- 131: Threaten non-force
- 132: Threaten with administrative sanctions
- 133: Threaten political dissent
- 134: Threaten to halt negotiations
- 135: Threaten to halt mediation
- 136: Threaten to halt international involvement
- 137: Threaten with repression
- 138: Threaten with military force
- 139: Give ultimatum

• 14: Protest
- 140: Engage in political dissent, not specified
- 141: Demonstrate or rally
- 142: Conduct hunger strike
- 143: Conduct strike or boycott
- 144: Obstruct passage or block
- 145: Protest violently or riot

• 15: Exhibit military posture
- 150: Exhibit military or police power, not specified
- 151: Increase police alert status
- 152: Increase military alert status
- 153: Mobilize or increase police power
- 154: Mobilize or increase armed forces
- 155: Mobilize or increase cyber-forces

- 16: Reduce relations
  - 160: Reduce relations, not specified
  - 161: Reduce or break diplomatic relations
  - 162: Reduce or stop material aid
  - 163: Impose embargo or boycott or sanctions
  - 164: Halt negotiations
  - 165: Halt mediation
  - 166: Expel or withdraw
- 17: Coerce
  - 170: Coerce
  - 171: Seize or damage property
  - 172: Impose administrative sanctions
  - 173: Arrest or detain
  - 174: Expel or deport individuals
  - 175: Use repression
  - 176: Attack cybernetically
- 18: Assault
  - 180: Use unconventional violence, not specified
  - 181: Abduct or hijack or take hostage
  - 182: Physically assault
  - 183: Conduct suicide or car or other non-military bombing
  - 184: Use as human shield
  - 185: Attempt to assassinate
  - 186: Assassinate
- 19: Fight
  - 190: Use conventional military force, not specified
  - 191: Impose blockade or restrict movement
  - 192: Occupy territory
  - 193: Fight with small arms and light weapons
  - 194: Fight with artillery and tanks
  - 195: Employ aerial weapons
  - 196: Violate ceasefire
- 20: Engage in unconventional mass violence
  - 200: Use massive unconventional force, not specified
  - 201: Engage in mass expulsion
  - 202: Engage in mass killings
  - 203: Engage in ethnic cleansing
  - 204: Use weapons of mass destruction

## G.3   HUMAN EVALUATION ON THE DATASET QUALITY

## H   ADDITIONAL DETAILS ABOUT AGENT SETUP

**Final answer extraction.** Our agent has two stopping criteria: 1) The agent makes the final answer. 2) The reasoning process fails to reach an answer, and ends with consecutive invalid actions, consecutive repetitive actions, or exceeded max iterations. We define and analyze this final status in Appendix D.1. Afterwards, we always perform an answer extraction step, which is performed by GPT-3.5-Turbo. If the agent has not generated a final answer during ReAct, we instruct the answer extraction model to make a prediction based on the ReAct reasoning trace. The prompt for answer extraction is shown in Appendix I.3.

# I PROMPTS

## I.1 SYSTEM PROMPTS

The system prompt provides the necessary background information, task description, and guidelines for the LLM agent. In our case, we introduce the forecasting task, basic information of the database, the defined API, and the planning strategies in the system prompt.

### I.1.1 SYSTEM PROMPT FOR REACT AGENT WITH ACTION TYPE AS "SINGLE FUNCTION"

---

**System prompt for ReAct agent with action type as single function (part 1)**

You are an expert in forecasting future events based on historical data. The database contains news articles from January 1, 2023 to the current date {current_date_nlp} and the events extracted from these articles. The events are in the form of (date, subject country, relation, object country), where the countries are represented by ISO 3166-1 alpha-3 codes and the relations are represented by the CAMEO codes defined in the 'Conflict and Mediation Event Observations' ontology. The relations are hierarchical: first-level relations are general parent relations represented by two-digit CAMEO codes, while second-level relations are more specific child relations represented by three-digit CAMEO codes. Child relations have the same first two digits as their parent relations. For example, '01' is a first-level relation, and '010' and '011' are some of its second-level relations. The relations in the database are represented in the second-level form.

Your task is to forecast the future relations between two entities in a given query. You have access to a defined Python API that allows you to query the database for historical events and statistics, and to get precise information about the ISO country codes and CAMEO relation codes.

The defined API is described as follows:
```python
{api_description}
```

You will use an iterative approach, interleaving 'Thought', 'Action', and 'Observation' steps to collect information and perform the forecast. You may perform up to {max_iterations} iterations. The steps are as follows:

- 'Thought': Analyze the current information and reason about the current situation, and predicts which API you want to use (try to use different APIs to collect diverse information) or make a decision that you want to make a final answer.
- 'Action': Use the API to gather more information or provide the final forecast.
    - If using the API: the action must be only one single line of exactly one function call from the API with appropriate inputs, without additional code, explanations, or natural language descriptions.
    - If making the final forecast: the action must start immediately with 'Final Answer:', and follow with the results in the expected JSON format.
- 'Observation': Return the output of the called function.

To make a reasonable forecast, you should collect both news and relational evidence to support your prediction. When you are fully confident that you accumulate enough information to make the final forecast, you should start the 'Thought' with your reasoning using the news and structural information to make the prediction, and then start the 'Action' step with 'Final Answer:' followed by the answer in the expected JSON format. The answer should be a JSON dictionary where the keys are the forecasted two-digit first-level CAMEO codes and the values are lists of forecasted three-digit second-level CAMEO codes that are child relations of the key. For example, 'Action: Final Answer: "01": ["010", "011", "012"], "02": ["020", "023"]'.

---

**System prompt for ReAct agent with action type as single function (part 2)**

The final answer will be evaluated based on the precision and recall of the forecasted first-level and second-level relations, so only include confident first-level and second-level CAMEO codes in your final forecast.

Try to use different APIs to collect diverse information (including multi-hop relations), such as the precise meaning of CAMEO codes, insights from news content, relational data, and statistical analyses to support your forecasts. Consider not only the frequency of the relations but also the temporal aspects of the data when making your forecast.

### I.1.2 SYSTEM PROMPT FOR REACT AGENT WITH ACTION TYPE AS "CODE BLOCK"

**System prompt for ReAct agent with action type as code block (part 1)**

You are an expert in forecasting future events based on historical data. The database contains news articles from January 1, 2023 to the current date {current_date_nlp} and the events extracted from these articles. The events are in the form of (date, subject country, relation, object country), where the countries are represented by ISO 3166-1 alpha-3 codes and the relations are represented by the CAMEO codes defined in the 'Conflict and Mediation Event Observations' ontology. The relations are hierarchical: first-level relations are general parent relations represented by two-digit CAMEO codes, while second-level relations are more specific child relations represented by three-digit CAMEO codes. Child relations have the same first two digits as their parent relations. For example, '01' is a first-level relation, and '010' and '011' are some of its second-level relations. The relations in the database are represented in the second-level form.

Your task is to forecast the future relations between two entities in a given query. You have access to a defined Python API that allows you to query the database for historical events and statistics, and to get precise information about the ISO country codes and CAMEO relation codes. You are also authorized to utilize additional safe, well-established Python libraries such as numpy, pandas, scikit-learn, and NetworkX to enhance your data analysis and forecasting accuracy.

The defined API is described as follows:
``` python
{api_description}
```

You will use an iterative approach, interleaving 'Thought', 'Action', and 'Observation' steps to collect information and perform the forecast. You may perform up to max_iterations iterations. The steps are as follows:

- 'Thought': Analyze the current information and reason about the current situation, and predicts which API you want to use (try to use different APIs to collect diverse information) or make a decision that you want to make a final answer.
- 'Action': Use the API to gather more information or provide the final forecast.
    - If gathering more data: the action must be an executable Python code snippet that starts with ```python' and ends with ````'. It can contain multiple lines of codes and function calls using the defined API or Python libraries. You must use print() to output the results, and only the printed output will be returned in the observation step.
    - If making the final forecast: the action must start immediately with 'Final Answer:', and follow with the answer in the expected JSON format. This should not be enclosed within triple backticks.
- 'Observation': Return the printed output of the executed code snippet.

---

**System prompt for ReAct agent with action type as code block (part 2)**

To make a reasonable forecast, you should collect both news and relational evidence to support your prediction. When you are fully confident that you accumulate enough information to make the final forecast, you should start the 'Thought' with your reasoning using the news and structural information to make the prediction, and then start the 'Action' step with 'Final Answer:' followed by the answer in the expected JSON format. The answer should be a JSON dictionary where the keys are the forecasted two-digit first-level CAMEO codes and the values are lists of forecasted three-digit second-level CAMEO codes that are child relations of the key. For example, 'Action: Final Answer: {"01": ["010", "011", "012"], "02": ["020", "023"]}'.

The final answer will be evaluated based on the precision and recall of the forecasted first-level and second-level relations, so only include confident first-level and second-level CAMEO codes in your final forecast.

Try to use different APIs and Python libraries to collect diverse information (including multi-hop relations), such as the precise meaning of CAMEO codes, insights from news content, relational data, and statistical analyses to support your forecasts. Consider not only the frequency of the relations but also the temporal aspects of the data when making your forecast.

---

### I.1.3 SYSTEM PROMPT FOR DIRECT AGENT

---

**System prompt for direct agent**

You are an expert in forecasting future events based on historical data. The events are in the form of (date, subject country, relation, object country), where the countries are represented by ISO 3166-1 alpha-3 codes and the relations are represented by the CAMEO codes defined in the 'Conflict and Mediation Event Observations' ontology. The relations are hierarchical: first-level relations are general parent relations represented by two-digit CAMEO codes, while second-level relations are more specific child relations represented by three-digit CAMEO codes. Child relations have the same first two digits as their parent relations. For example, '01' is a first-level relation, and '010' and '011' are some of its second-level relations. The relations in the database are represented in the second-level form.

Your task is to forecast the future relations between two entities in a given query. The answer should be a JSON dictionary where the keys are the forecasted two-digit first-level CAMEO codes and the values are lists of forecasted three-digit second-level CAMEO codes that are child relations of the key. For example, 'Final Answer: {{"01": ["010", "011", "012"], "02": ["020", "023"]}}'.

The final answer will be evaluated based on the precision and recall of the forecasted first-level and second-level relations, so only include confident first-level and second-level CAMEO codes in your final forecast.

---

### I.1.4  SYSTEM PROMPT FOR COT AGENT

> **System prompt for CoT**
>
> You are an expert in forecasting future events based on historical data. The events are in the form of (date, subject country, relation, object country), where the countries are represented by ISO 3166-1 alpha-3 codes and the relations are represented by the CAMEO codes defined in the 'Conflict and Mediation Event Observations' ontology. The relations are hierarchical: first-level relations are general parent relations represented by two-digit CAMEO codes, while second-level relations are more specific child relations represented by three-digit CAMEO codes. Child relations have the same first two digits as their parent relations. For example, '01' is a first-level relation, and '010' and '011' are some of its second-level relations. The relations in the database are represented in the second-level form.
>
> Your task is to forecast the future relations between two entities in a given query. **To make a reasonable forecast, you should first think and reason based on your background knowledge. When you are confident that you have conducted enough analysis to make the final answer**, you should start answering by 'Therefore, the final answer is:' followed by the answer in the expected JSON format. The JSON format should be a JSON dictionary where the keys are the forecasted two-digit first-level CAMEO codes and the values are lists of forecasted three-digit second-level CAMEO codes that are child relations of the key. For example, '{{"01": ["010", "011", "012"], "02": ["020", "023"]}}'.
>
> The final answer will be evaluated based on the precision and recall of the forecasted first-level and second-level relations, so only include confident first-level and second-level CAMEO codes in your final forecast.

### I.2  QUERY PROMPT

> **Query prompt**
>
> Please forecast the relations that {actor1_name} will take towards {actor2_name} on {future_date_nlp} based on historical information up to {current_date_nlp}. I.e. forecast the relation CAMEO codes in query event Event(date={future_date}, head_entity=ISOCode({actor1_code}), relation=CAMEOCode(?), tail_entity=ISOCode({actor2_code})).

I.3  ANSWER EXTRACTION PROMPT

> **Answer extraction prompt**
>
> Please help me extract final answer for forecasting the future relations between two entities in a given query: forecast the relations that {actor1_name} will take towards {actor2_name} on {future_date_nlp} based on historical information up to current_date_nlp. I.e. forecast the relation CAMEO codes in query event Event(date={future_date}, head_entity=ISOCode({actor1_code}), relation=CAMEOCode(?), tail_entity=ISOCode({actor2_code})).
>
> I have used interleaving 'Thought', 'Action', and 'Observation' steps to collect information from the database and perform the forecast. The database contains news articles from January 1, 2023 to the current date current_date_nlp and the events extracted from these articles. The events are in the form of (date, subject country, relation, object country), where the countries are represented by ISO 3166-1 alpha-3 codes and the relations are represented by the CAMEO codes defined in the 'Conflict and Mediation Event Observations' ontology. The relations are hierarchical: first-level relations are general parent relations represented by two-digit CAMEO codes, while second-level relations are more specific child relations represented by three-digit CAMEO codes. Child relations have the same first two digits as their parent relations. For example, '01' is a first-level relation, and '010' and '011' are some of its second-level relations. The relations in the database are represented in the second-level form.
>
> The final forecast answer need to forecast both first-level and second-level CAMEO codes, and will be evaluated based on the precision and recall of both levels of relations. The final answer content should be a JSON dictionary where the keys are the forecasted two-digit first-level CAMEO codes and the values are lists of forecasted three-digit second-level CAMEO codes that are child relations of the key. For example, {{"01": ["010", "011", "012"], "02": ["020", "023"]}}.
>
> The latest information and forecast I have collected is as follows:
> {info}
>
> If final forecast answer has been made in the collected information indicated by "Final Answer:", you must only reformat the final forecast answer in the expected JSON dictionary format inside XML tags. For example: <answer>{{"01": ["010", "011", "012"], "02": ["020", "023"]}}</answer>.
>
> Otherwise, if no final forecast is made, you must reason based on the information you have collected and generate a confident final forecast answer to the query, and then reformat your answer in the expected JSON dictionary format inside XML tags.

## J   DATASHEET FOR MIRAI

### J.1   MOTIVATION

1. **For what purpose was the dataset created?**
   The MIRAI dataset was created to evaluate different capabilities of Large Language Model (LLM) agents in forecasting international events.

2. **Who created the dataset and on behalf of which entity?**
   The dataset was developed by a group of university researchers studying LLM reasoning. All rights are held by the individuals themselves, rather than by third-party stakeholders.

3. **Who funded the creation of the dataset?**
   The academic institutions support the university reasearch lab that the authors affiliate in.

### J.2   COMPOSITION

1. **What do the instances that comprise the dataset represent?**
   Each instance in the dataset represents a record of international events, including the date, involved countries, and type of event, along with associated news articles and metadata.

2. **How many instances are there in total?**
   The current database comprises 1,296,991 GDELT event records from January 2023 to February 2024, corresponding to 75,341 unique events and 401,013 news articles. Test splits span over November 2023 to February 2024, each containing 100 balanced-sampled forecasting event queries. Note these statistics reflect the dataset used in this paper's experiments. As a dynamic benchmark, we are committed to periodic updates, continuously expanding the historical database and creating new test splits.

3. **Does the dataset contain all possible instances or is it a sample of instances from a larger set?**
   The dataset represents a curated sample from the entire GDELT database. It has been created through meticulously designed data cleaning and preprocessing steps on GDELT raw data, aimed at enhancing the quality and reliability of the event data.

4. **Is there a label or target associated with each instance?**
   Yes, each instance in the dataset is an event labeled with a relation type derived from the CAMEO[7] event taxonomy.

5. **Is any information missing from individual instances?**
   No, all instances are complete with all available information.

6. **Are there recommended data splits (e.g., training, development/validation, testing)?**
   The dataset is flexible in splitting data and expanding future events to new test splits. In this paper, we provide multiple test splits, spanning from November 2023 to February 2024. Each test instance utilizes all preceding records as its historical dataset to ensure accuracy and relevance in analysis. For training and development, users can flexibly use the data prior to the test splits based on their specific research needs.

7. **Are there any errors, sources of noise, or redundancies in the dataset?**
   The dataset has undergone extensive cleaning and structuring to minimize errors and noise. However, residual noise from the original GDELT database may still be present.

8. **Is the dataset self-contained, or does it link to or otherwise rely on external resources (e.g., websites, tweets, other datasets)?**
   While the dataset is primarily self-contained, it includes URLs to news articles, providing links to external resources for further context and verification.

9. **Does the dataset contain data that might be considered confidential?**
   No, the dataset contains publicly available data, and does not include confidential information. It follows the term of use for GDELT, which is an open platform for research and analysis of global society.

10. **Does the dataset contain data that, if viewed directly, might be offensive, insulting, threatening, or might otherwise cause anxiety?**
    The dataset may include descriptions of sensitive events, such as global conflicts, due to its focus on international relations. However, We conduct rigorous text cleaning procedures to reduce noise from web content while enhancing the reliability and ethical integrity of the textual information,

---

[7]Conflict and Mediation Event Observations (CAMEO): https://parusanalytics.com/eventdata/data.dir/cameo.html

following the OBELICS protocol (Laurençon et al., 2023). Thorough checks such as flagging word ratios are employed during the data cleaning process to identify and potentially exclude inappropriate paragraphs or entire news articles and events. This ensures the minimization of distressing content while maintaining the integrity and relevance of the dataset for academic study.

### J.3 COLLECTION PROCESS

1. **How was the data associated with each instance acquired?**
   Data for each instance was sourced from the GDELT project, which aggregates global event data and news articles from various worldwide media. Detailed information can be found in Section 2.3 of the paper.

2. **What mechanisms or procedures were used to collect the data (e.g., hardware apparatus or sensor, manual human curation, software program, software API)?**
   Data collection was facilitated through software programs that aggregate event data and news articles from various sources. This automated collection is followed by data cleaning processes to enhance completeness and reliability. For a detailed description of the collection and cleaning methods, please refer to Section 2.3 of the paper. All dataset construction scripts are available in Github and also described in its README.

3. **Who was involved in the data collection process? (e.g., students, crowdworkers, contractors) and how were they compensated (e.g., how much were crowdworkers paid)**
   The data collection and preprocessing were conducted by the authors, who are university researchers. This work was performed as part of their academic research activities.

4. **Does the dataset relate to people?**
   No. The dataset does not contain personal data or directly relate to individual people.

5. **Did you collect the data from the individuals in questions directly, or obtain it via third parties or other sources (e.g., websites)?**
   The dataset does not involve data collected from individuals directly. Instead, it is curated from the GDELT project, which aggregates information from various global news media sources.

### J.4 USES

1. **Has the dataset been used for any tasks already?**
   The dataset has not been used for any tasks other than the ones proposed and examined in this current paper, specifically for benchmarking the forecasting capabilities of LLM agents in predicting international relations. The dataset has also supported evaluating forecasting performance of traditional temporal knowledge graph-based methods and natural langugage-based methods.

2. **What (other) tasks could the dataset be used for?**
   In addition to benchmarking LLM agents and traditional forecasting methods, the dataset could be valuable for research in geopolitics, the development of other event prediction algorithms, sentiment analysis of international events, and trend analysis in global political dynamics.

3. **Is there anything about the composition of the dataset or the way it was collected and preprocessed/cleaned/labeled that might impact future uses?**
   The dataset was meticulously collected and preprocessed to ensure high quality and reliability. However, the reliance on the GDELT project as the primary data source and the specific cleaning process applied might influence its applicability to certain tasks. Users should consider these factors when applying the dataset to different research areas or methodologies.

4. **Are there tasks for which the dataset should not be used?**
   The dataset should not be used for any tasks that violate the terms of use associated with the GDELT project. We clearlt cite the terms of use in Appendix J.7.

### J.5 DISTRIBUTION

1. **Will the dataset be distributed to third parties outside of the entity?**
   Yes, the dataset, evaluation codes and leaderboards are intended to be publicly available to foster future research and development.

2. **How will the dataset be distributed?**
   The database and codebase are currently available via an academic website, Google Drive, and Github. To enhance the accessibility and utility, the distribution of current version of data and its future updates will be enhanced by uploading the dataset to Hugging Face, and refining the API into a more user-friendly library format in the future.

3. **Have any third parties imposed IP-based or other restrictions on the data associated with the instances?**
No, there are no IP-based or other restrictions on the data: MIRAI is curated based on the GDELT[8] Event Database, which is an open platform for research and analysis of global society and all datasets released by the GDELT Project are available for unlimited and unrestricted use for any academic, commercial, or governmental use of any kind without fee; Data are allowed for any redistribution, rehost, republish, and mirror of the GDELT datasets in any form, with necessary citations[9].

4. **Do any export controls or other regulatory restrictions apply to the dataset or to individual instances?**
No, there are no export controls or other regulatory restrictions applied to this dataset.

5. **When will the dataset be distributed?**
The dataset will be made publicly available after the review process is completed, with the current academic website, Google Drive, and Github, and additional release/updates of arXiv, Hugging Face, and leaderboard to facilitate a more comprehensive accessibility to the research community.

6. **Will the dataset be distributed under a copyright or other IP license, and/or under applicable terms of use (ToU)?**
The dataset will be distributed under the CC BY-NC 4.0 license, allowing for use and distribution for non-commercial purposes with appropriate attribution.

### J.6 MAINTENANCE

1. **Who will be supporting/hosting/maintaining the dataset?**
The dataset maintenance will be supporting/hosting/maintaining by the authors.

2. **How can the owner/curator/manager of the dataset be contacted?**
The owner/curator/manager of the dataset can be contacted through the authors' emails.

3. **Will the dataset be updated? (e.g., to correct labeling errors, add new instances, delete instances)?**
Yes, the dataset is designed as a dynamic benchmark with periodic updates. We are committed to regularly expanding the historical database and creating new test splits. Updates will include adding new event data, creating new test splits, and potentially correcting any identified errors. Announcements regarding updates will be made through the project's official channels.

4. **If the dataset relates to people, are there applicable limits on the retention of the data associated with the instances (e.g., were the individuals in question told that their data would be retained for a fixed period of time and then deleted?)**
NA. This dataset does not contain data related to individuals or personal identifiers. It consists of aggregated event data and news articles related to international events.

5. **Will older version of the dataset continue to be supported/hosted/maintained?**
Our dataset is designed as a cumulative, evolving benchmark. Historical event data is continuously incorporated into the database and remains available for future event forecasting. While we strongly recommend using the latest test splits to ensure data-contamination-free benchmarking for all LLM models, older historical data remains an integral part of the dataset. For reproducibility purposes, we will maintain records of the specific data versions used in published experiments. Researchers can request access to particular historical snapshots of the dataset, subject to resource availability. However, for ongoing research and comparisons, we encourage using the most current version of the dataset and its latest test splits.

6. **If others want to extend/augment/build on/contribute to the dataset, is there a mechanism for them to do so?**
Researchers and developers interested in extending, augmenting, or contributing to the dataset are encouraged to submit their changes through GitHub pull requests. For additional inquiries or detailed discussions, contacting the authors via email is recommended.

### J.7 TERM OF USE FOR GDELT

Based on https://www.gdeltproject.org/about.html#termsofuse, GDELT dataset "*is an open platform for research and analysis of global society and thus all datasets*

---

[8]GDELT Project: https://www.gdeltproject.org/
[9]GDELT Term of Use: https://www.gdeltproject.org/about.html#termsofuse

*released by the GDELT Project are available for unlimited and unrestricted use for any aca-demic, commercial, or governmental use of any kind without fee.*", as long as "*any use or re-distribution of the data must include a citation to the GDELT Project and a link to this website (`https://www.gdeltproject.org/`).*", which we've cited in abstract.

