# OpenReview forum: "MIRAI: Evaluating LLM Agents for International Event Forecasting"
_ICLR.cc/2025/Conference — Submitted to ICLR 2025_

### Official Review · Reviewer_SzJ2 · 2024-11-01

**Soundness:** 3
**Presentation:** 3
**Contribution:** 3
**Rating:** 6
**Confidence:** 4

**Summary:**

This paper introduces MIRAI, a practical benchmark designed to predict future interactions between two countries. It constructs a dataset based on news articles and existing information extraction and representation tools, and then exposes an API that allows an LLM agent to interact with the different data sources. The authors' extensive experiments provide valuable insights into how LLMs perform on this task.

**Strengths:**

S1. The paper provides the code and data. Besides, it comes with an extensive description of the data and code, ensuring reproducibility and reuse.

S2. The authors ran a lot of experiments testing different scenarios and configurations. They obtained very insightful and broad results about the task.

S3. The paper is well illustrated, with many examples showcasing the performance of the systems. It is also well-written and easy to follow.

**Weaknesses:**

W1. The paper heavily relies on the appendix (the paper is 73 pages long!). As a reviewer, I am not supposed to read it, but in many cases, I had to. For example, the previous work is mostly in the appendix (two small paragraphs are in the main paper) and the human evaluation is in the appendix.

W2. The technical contribution of the paper could be clearer. The dataset construction pipeline is standard and relies on many existing tools with additional cleaning. The baselines are also coming from previous works.

W3. The scope of the task is very narrow: Predicting the future interaction between two countries.

Others/Typos

O1. Some figures are hard to read, especially in black and white (2, 4, 5, 7, 8)

O2. Human evaluation of the dataset is done in the appendix. Is it enough to only evaluate 51 events? Besides, 82% seems quite low to me.

O3. The authors include the confidence intervals but do not discuss them. In many cases, there are overlaps between the baseline, making the conclusions unclear.

O4. Adding the human evaluation in Table 2 is only one line, and it would be insightful.

O5. Having the RAG baseline would be very insightful in knowing if one needs the API interface.

T6. Line 420, problem citation (et al, 2024)

O7. Figure 8.b: In general, red is used to show incorrect things. Here, it is counter-intuitive.

**Questions:**

Q1. Are the news articles freely available? Are there copyright issues?

Q2. Line 260: Is balancing this way fair? Some months are naturally more active than others, and some relationships are more frequent than others.

Q3. How does the number of parameters in the LLM impact the performance? In Table 2, GPT-3.5-turbo has more parameters than the other baselines but is competitive with GPT-4o-mini, which the oldness of GPT-3.5 can explain.

Q4. For the Direct IO baseline, isn't a main issue outputting the output following the standard of the answer? Do you provide the possible relationships in any way?

---

> ### Author Response · Authors · 2024-11-23
>
> We are grateful for your support and suggestions, for which we have included additional experiments and discussions accordingly. Please find our responses below:
>
> ---
>
> ### W1. The paper heavily relies on the appendix. Previous work and human evaluation are in the appendix.
> Thank you for this suggestion. We have streamlined the Appendix by:
> 1. Removing the full API documentation and implementation details and providing a link to anonymous github repo.
> 2. Retaining only the key list of data classes and functions defined in the API that directly support the paper’s contribution in the agentic environment and the code-based tool use setting.
>
> For related works, as our paper primarily introduces a new **benchmark** designed to facilitate the development and evaluation of agent forecasting methods, the primary competitors in the related work are existing benchmarks in this domain. Therefore, in the main content of paper, we made comparison  to these most relevant benchmarks in Table 4 and its following section. By comparing our work with these existing benchmarks, we demonstrated how our benchmark is unique and addresses current gaps in the lack of dynamic data updates, diverse information sources, and agentic evaluation. Extended related work in the appendix focus more on the background of previous non-agentic forecasting methods and agentic/ code-based  LLM work.
>
> For human evaluation, as suggested, we have incorporated the human forecasting performance results into main paper (page 7 Table 2). Additionally, we note that the human evaluation was conducted on a subset of 51 test events due to resource constraints and the time-intensive nature of expert evaluation. This is also noted in the caption of Table 2.
>
> ---
>
> ### W2. The technical contribution of the paper could be clearer. The baselines are coming from previous works.
> Thank you for suggesting we clarify the technical contributions of our paper. We would like to emphasize that as a benchmark work, our primary contribution is not in developing novel data processing tools or baseline methods, but rather in constructing the first comprehensive benchmark for evaluating and advancing LLM agents' capabilities in event forecasting. Our benchmark makes several key technical contributions:
>
> 1. We design and implement the first agentic environment specifically for temporal forecasting, featuring carefully constructed APIs that enable systematic evaluation of agents' abilities in information integration, tool use, and reasoning. This fills a critical gap in the field where no previous benchmark has specifically targeted the evaluation of LLM agents' forecasting capabilities.
> 2. Our benchmark uniquely combines structured events and unstructured news within an agentic framework, allowing us to assess how effectively agents can gather and integrate diverse information sources. While we leverage existing data sources, our technical contribution lies in creating an interactive environment that enables agent-driven information exploration and integration—a capability not present in traditional static datasets.
> 3. We implement a dynamic pipeline that supports regular updates and ensures contamination-free test splits. This technical design choice directly addresses a critical challenge in LLM evaluation: maintaining benchmark relevance as models evolve while preventing data contamination.
>
> You may refer to our global rebuttal comments for a comprehensive discussion of our [contributions](https://openreview.net/forum?id=gzzX4ZeErx&noteId=ZueduytQF6).
>
> ---
>
> ### W3. The scope of the task is very narrow: Predicting the future interaction between two countries.
> We respectfully disagree that the task scope is narrow. Our benchmark evaluates forecasting across the comprehensive CAMEO ontology, which provides a hierarchically organized framework encompassing the full spectrum of international interactions: The relation space is rich and diverse, including a progression of relation types from cooperative to conflictual interactions, providing a nuanced space for evaluating models' ability to forecast shifts in international dynamics.
>
> Additionally, our experimental results demonstrate that even this foundational task formulation poses significant challenges for both traditional and agentic methods. The performance analysis has already yielded valuable insights about current LLM agents' capabilities and limitations in temporal reasoning, tool utilization, and information integration.
>
> While we acknowledge the potential for more complex task formulations (e.g., multi-party events, intra-country dynamics) and consider exploring them as future work, we believe establishing strong benchmarks for fundamental bilateral interactions is crucial before advancing to more complex scenarios. By constructing MIRAI as a focused yet comprehensive evaluation framework, we aim to drive systematic progress in agentic forecasting methods while maintaining clear metrics for measuring advancement.

---

> ### Author Response · Authors · 2024-11-23
>
> ### O1. Some figures are hard to read, especially in black and white (2, 4, 5, 7, 8)
>
> Thanks for the suggestion. We have revised these figures using more distinguishable color schemes, hatching patterns, line style, and markers in the updated manuscript.
>
> ---
>
> ### O2. Human evaluation of the dataset is done in the appendix. Is it enough to only evaluate 51 events? Besides, 82% seems quite low.
>
> Thank you for raising this important point about our human evaluation process. The 51 events evaluated correspond directly to the ground truth events derived from our human forecasting queries, encompassing 103 associated news articles. This selection ensures a diverse and meaningful subset of the dataset for evaluation.
>
> While we acknowledge that expanding the evaluation scope would provide additional insights, the current sample size reflects the resource-intensive nature of this process. Each event requires careful analysis of multiple news articles within their temporal and contextual nuances. Furthermore, accurate application of the CAMEO ontology is a non-trivial task, necessitating meticulous attention to detail to appropriately classify real-world events.
>
> Our evaluators, while equipped with the CAMEO ontology, were not domain experts in political science, which likely limited their ability to achieve the precision that specialized experts could provide.
>
> It is worth noting that previous work [1] evaluated raw GDELT event data using GPT-4 on a sample of 100 randomly selected news articles from the Middle East, reporting a precision of only 42.5%. In contrast, the 82% accuracy achieved in our human evaluation demonstrates the effectiveness of our data cleaning process and highlights the quality of our dataset. While there is room for further improvement, these results provide strong evidence for the validity of our methods.
>
> [1] SCTc-TE: A Comprehensive Formulation and Benchmark for Temporal Event Forecasting.
>
> ---
>
> ### O3.  The authors include the confidence intervals but do not discuss them. In many cases, there are overlaps between the baseline, making the conclusions unclear.
>
> Thank you for highlighting the need to discuss confidence intervals in our results. We acknowledge that overlapping confidence intervals between models warrant careful interpretation. We have conducted additional statistical analysis to provide more rigorous support for our key conclusions from the paper:
>
> 1. Our first main finding that "Code Block benefits stronger LLMs but hurts weaker models" is supported by clear patterns even when considering confidence intervals:
>    - For GPT-4o-mini, the adoption of Code Block shows consistently higher mean values with smaller standard deviations in second-level relation prediction tasks, indicating both improved performance and increased stability.
>    - In contrast, smaller open-source LLMs demonstrate consistently decreased performance with Code Block, with the negative impact remaining significant even when accounting for confidence intervals.
>
> 2. Our second main finding that "GPT-4o-mini outperforms other models" is supported by statistical evidence. While there are some overlaps between confidence intervals (e.g., between GPT-3.5 and GPT-4o-mini), additional statistical testing validates this conclusion. For instance, a Welch's t-test comparing second-level F1 scores between these models yields p=2.45e-06 < 0.05, confirming that GPT-4o-mini's superior performance is statistically significant.
>
> This additional statistical analysis provides a clearer and more rigorous interpretation of our results.
>
> ---
>
> ### O4. Adding the human evaluation in Table 2 is only one line, and it would be insightful.
>
> In response to your comment, we have incorporated the human forecasting performance results into the main paper (page 7 Table 2). Additionally, we note that the human evaluation was conducted on a subset of 51 test events due to resource constraints and the time-intensive nature of expert evaluation. This is also noted in the caption of Table 2.
> Moreover, for fair comparison, we have run all LLM agent experiments on this same subset, with results presented in Table 6 of the Appendix D.4. The results show that  human performance surpassed that of LLM agents in most metrics, especially in recall. This highlights significant room for improvement in LLM agents.

---

> ### Author Response · Authors · 2024-11-23
>
> ### O5. Having the RAG baseline would be very insightful in knowing if one needs the API interface.
>
> Thank you for this valuable suggestion for incorporating RAG-based methods into our dataset’s evaluation. We have now added RAG evaluations to provide additional context. We also want to note that retrieval is inherently a critical tool for LLM agents [2, 3]. RAG represents a specific type of augmented LLM, and the key difference is that RAG always performs retrieval augmentation for generation while agents automatically choose when to use the tool (see our detailed discussion below).
>
> Follow the suggestion,  we conduct additional experiments with different RAG methods. The result table is as follows:
>
> **Table:** Evaluation results of GPT-4o-mini on the 2024-02 test split using different non-agentic methods and the ReAct agent with Single Function action type. The best-performing score is highlighted in **bold** and the second-best is in *italics*.
>
> | Method      | Augmented Context | Agent API     | Binary KL (↓)      | Quad KL (↓)       | First-level Relation (%) - Pre. (↑) | Rec. (↑)       | F1 (↑)       | Second-level Relation (%) - Pre. (↑) | Rec. (↑)       | F1 (↑)       |
> |-------------|-------------------|---------------|--------------------|-------------------|-------------------------------------|----------------|--------------|-------------------------------------|----------------|--------------|
> | Direct IO   | ---               | ---           | 3.6±1.0           | 7.6±1.9          | 39.5±3.2                           | 44.8±3.2       | 34.9±3.5     | 15.4±0.8                           | 23.9±3.6       | 15.4±0.2     |
> |             | CAMEO            | ---           | 5.0±1.5           | 7.3±1.9          | 35.5±4.6                           | 32.6±2.2       | 28.6±2.7     | 10.0±1.5                           | 14.2±0.9       | 10.1±0.8     |
> | RAG         | *Events-Only*    | ---           | **2.2±0.9**       | **5.9±2.0**      | 57.5±3.5                           | **53.4±3.4**   | **50.5±3.8** | 32.4±1.1                           | **43.9±2.0**   | **33.2±1.4** |
> |             | *News-Only*      | ---           | 9.1±2.8           | 12.7±2.9         | 47.2±0.8                           | 23.2±2.4       | 25.4±0.2     | 19.5±2.0                           | 14.9±2.1       | 13.4±0.8     |
> |             | *All*            | ---           | *2.3±1.4*         | *6.3±2.0*        | 59.0±1.2                           | *48.1±1.2*     | *46.7±0.4*   | 36.4±5.3                           | *38.8±1.2*     | *32.1±2.4*   |
> | ReAct       | ---              | *Event-Only*  | 3.3±0.8           | 7.7±1.4          | **62.8±10.5**                       | 39.0±0.8       | 41.7±5.3     | *44.2±3.3*                         | 37.0±0.8       | 30.7±0.9     |
> |             | ---              | *News-Only*   | 6.5±1.7           | 13.0±2.1         | 41.5±6.1                           | 16.8±0.7       | 20.2±1.9     | 17.8±0.2                           | 12.2±1.0       | 12.5±0.5     |
> |             | ---              | *All*         | 3.6±0.9           | 8.0±1.5          | *61.7±10.1*                         | 38.6±1.9       | 40.7±5.6     | **46.3±4.4**                        | 32.9±3.8       | 31.1±2.6     |
>
>
> **Methods and Experimental Setup:** Besides the **Direct IO / QA**and three **ReAct** agent with different tool-use that we already implemented in the paper, we add the following baselines:
>
> - **Direct QA with Augmentation**  (for comment Q4)
>    - **QA with CAMEO**: We provide the CAMEO ontology in an ordered dictionary format mapping relation codes to their names and detailed descriptions. This is closer to the QA-format the authors mention, and a more fair comparison as the model can refer to the output vocabulary without needing to memorize CAMEO codes.
>
> - **RAG Methods**
> Following recent work, we implement three RAG variants:
>    - **RAG Events-Only**: Following GPT-NeoX-ICL[4], we explore rule-based approaches for retrieving historical facts. Using the 'Pair' and 'Undirectional' setting, given a query event $(s, ?, o, t)$, we retrieve historical events $(s, r \in \mathcal{R}, o, <t)$ and $(o, r \in \mathcal{R}, s, <t)$. Events are sorted by recency with a cap of 30, aligning with the default cap of the `get_events` API function.
>    - **RAG News-Only**: Following TCELongBench[5], we employ BM25 retrieval to fetch the most query-relevant news articles before the query date. The top 15 news articles are retained, matching the default cap of the `get_news_articles` API function.
>    - **RAG All**: Combines both retrieved structured events and textual news articles.
>
> [2] Tool Learning with Large Language Models: A Survey
> [3] A Survey on Large Language Model based Autonomous Agents
> [4] Temporal Knowledge Graph Forecasting Without Knowledge Using In-Context Learning.
> [5] TCELongBench: Analyzing Temporal Complex Events with Large Language Models? A Benchmark towards Temporal, Long Context Understanding.

---

> ### Author Response · Authors · 2024-11-23
>
> **Comparison of Retrieval Strategies with Agent:** Both baseline approaches—CAMEO context augmentation and RAG methods—employ static, predefined retrieval strategies that are fixed for all queries and executed only once per query.
>
> In contrast, our agentic approach enables dynamic, multi-step information gathering and reasoning. The agent **can** replicate the baseline retrieval strategy by fixing certain API parameter values, for example, the agent can use function call `get_events(head_entities=[s, o], tail_entities=[o,s])` to get the retrieved context as RAG Event-Only, and use function call `get_news_articles(text_description='(t, s, ?, o)')` to get the retrieved context as RAG News-Only; its capabilities extend far beyond these static approaches through its flexible parameter settings for each function call and multiple steps per query.
>
> This multi-step, adaptive approach represents a fundamental shift from static retrieval to dynamic information gathering and reasoning, though it introduces higher requirements for the LLM's planning ability in:
> - Automatically selecting optimal information-gathering strategies
> - Integrating and reasoning over information of different formats
> - Adjusting strategies based on intermediate findings and current context
>
>
> **Key Findings and Analysis:** Our experiment results reveal several important insights:
>
> - **Performance of the RAG Baselines**
> RAG demonstrates improved precision over Direct IO (15.4% Pre in second-level) when using either event (32.4% Pre in second-level) or text (19.5% Pre in second-level) information source independently, with event data contributing more significantly to recall (43.9% Rec in second-level). When combining the two information sources, RAG achieves higher precision (36.4% Pre in second-level) but with a lower recall than RAG Event Only (from 43.9% to 38.8%), leading to a lower overall F1 score (from 33.2% to 32.1%), this suggests that a simple combination of both information in the context not effectively and collaboratively contribute to a better forecasting performance.
>
>
> - **Comparison and Insights for Agent**
> ReAct agents exhibit similar performance patterns with RAG when using different information sources (changed by the type of API functions available for the agent). Specifically, event data also contributes to high precision for the agent (62.8% Pre at first-level and 44.2% at second-level), outperforming RAG baselines.
> The structured event data consistently provides stronger signals for forecasting across both approaches, likely due to its standardized format.
>
>   However, agent baselines generally obtain a lower recall than RAG baselines. Meanwhile, while ReAct with full API access achieves higher average F1 scores in second-level relation prediction than its partial access performance(31.1% than 30.7% and 12.5%), the benefits of combining information sources aren't consistent across all relation hierarchies and methods, similar to RAG, suggesting substantial room for exploring more effective information integration strategies.
>
>   This reveals both promises and challenges of the agent's more flexible retrieval approach: RAG's predefined and fixed retrieval strategies can often yield stable performance, agent's dynamic and multi-step retrieval allows for flexible information gathering and integration, yet higher requirements for planning and reasoning sometimes also lead to relatively lower performance.
>
> These observations underscore the core purpose of our benchmark: not just to compare current methods but to encourage the development of more advanced agentic forecasting approaches. The current performance patterns suggest significant opportunities for improving agent architectures, particularly in:
>
> - Developing more robust and automatic planning strategies for multi-step information gathering
> - Improving information integration capabilities across different information sources, formats, and temporal scales

---

> > ### Author Response · Authors · 2024-11-23
> >
> > ### T6. Line 420, problem citation (et al, 2024).
> >
> > Thank you for catching the typo. We have corrected this in the updated revision.
> >
> > ---
> >
> > ### O7. Figure 8.b: In general, red is used to show incorrect things. Here, it is counter-intuitive.
> >
> > Thanks for the suggestion. We have revised Figure 8b to use another color scheme.
> >
> > ---
> >
> > ### Q1. Are the news articles freely available? Are there copyright issues?
> >
> > Thank you for bringing up the question regarding the availability of the news articles and potential copyright issues. Currently, the agents get news articles by utilizing only the metadata and titles of the news articles, which are typically permissible to use and share under fair use provisions. To distribute the full content of the articles while minimizing potential copyright concerns, before releasing any additional content beyond news metadata and titles, we will conduct a thorough review of the policies of the respective news agencies and adhere to all applicable copyright laws.
> >
> > ---
> >
> > ### Q2. Is balancing test split fair? Some months are naturally more active than others, and some relationships are more frequent than others.
> >
> > Thank you for this insightful question about the fairness of our balanced sampling approach. We acknowledge that some months are naturally more active than others and that certain relationships occur more frequently. We want to clarify several important points about our sampling approach:
> >
> > The balancing is performed solely on the selection of test queries, while the historical data used for model predictions remains unaltered and reflects the true activity levels and relationship frequencies. This approach aims to create a diverse and comprehensive test set that evaluates models across a wide range of scenarios, ensuring coverage of varied relation types, entity geographical distributions, and event temporal distributions that might be underrepresented in random selection.
> >
> > From an evaluation perspective, this balanced sampling prevents the test set from being dominated by highly active periods or frequently occurring relationships, which could skew the assessment. This design choice is validated by our experimental results in Figure 4b, where LLM models show a common performance variation—achieving higher accuracy in "verbal cooperation" and "material conflict" categories while struggling with "verbal conflict" and "material cooperation".
> >
> > We have made the detailed implementation of our balanced sampling approach available in our public repository: https://anonymous.4open.science/r/ForecastAgent-3419/dataset_construction/8_generate_test_subset.py
> >
> > ---
> > ### Q3. How does the number of parameters in the LLM impact the performance?
> >
> > Thank you for raising this suggestion of studying the impact of LLM parameter size to the forecasting performance. While LLM's parameter count can influence model capabilities, it is not the sole or even primary determinant of performance. Other crucial factors include model architecture, training data quality and recency, and advanced training techniques. For instance, GPT-4o-mini's competitive performance with GPT-3.5-turbo, despite having fewer parameters, can be attributed to more recent architectural improvements, different training data, and advanced training methodologies.
> >
> > To study the impact of parameter size, we conduct experiments with settings and results as below:

---

> > > ### Author Response · Authors · 2024-11-23
> > >
> > > **Table:** Evaluation results on the 2024-02 test split using different base LLMs with different numbers of model parameters.
> > > *The best-performing score is highlighted in **bold**, and the second-best is in *italics*.*
> > >
> > > | Base LLM             | Training Data Cutoff Date | Action Type   | Binary KL (↓)      | Quad KL (↓)        | First-level Relation (%) - Pre. (↑) | Rec. (↑)       | F1 (↑)         | Second-level Relation (%) - Pre. (↑) | Rec. (↑)       | F1 (↑)         |
> > > |-----------------------|---------------------------|---------------|--------------------|--------------------|-------------------------------------|----------------|----------------|-------------------------------------|----------------|----------------|
> > > | **Llama-3.2-1B-Instruct** | 2023-12                  | Single Func   | *9.5*±1.9         | 16.0±1.7           | 23.7±6.6                           | 10.0±1.9       | *11.7*±2.8     | 8.8±2.6                           | 7.2±0.2        | 6.1±1.3        |
> > > |                       |                           | Code Block    | 10.1±2.2          | 16.2±2.2           | 24.0±5.7                           | 8.1±2.1        | 10.0±3.4       | 7.6±1.9                           | 5.7±0.7        | 5.1±1.8        |
> > > | **Llama-3.2-3B-Instruct** | 2023-12                  | Single Func   | 12.1±2.2          | *15.4*±1.9         | **36.3**±2.2                       | **13.1**±3.7   | **16.7**±3.0   | **19.9**±0.7                       | **8.3**±2.6    | **9.3**±0.9    |
> > > |                       |                           | Code Block    | **9.3**±2.1       | **15.1**±0.5       | *26.7*±0.7                         | *10.5*±0.5     | 11.0±0.8       | *13.1*±1.5                         | *8.0*±0.4      | *7.1*±0.3      |
> > >
> > >
> > > To study the impact of LLM parameter size, we choose models from the same model family. Specifically, the above experiment table presents evaluation results comparing Llama-3.2-1B-Instruct and Llama-3.2-3B-Instruct on the 2024-02 test split, both sharing the same architecture, training data cutoff (2023-12), and training methodology. We include these discussions in Appendix D.8.
> > >
> > > Our analysis reveals two key findings:
> > >
> > > 1. **Parameter size shows a consistent positive correlation with forecasting performance within the same model family.**
> > >    The 3B model outperforms its 1B counterpart across all prediction levels, from binary (12.1% vs 9.5% in Single Function mode) to second-level relations (9.3% vs 6.1% in Single Function mode).
> > >
> > > 2. **The impact of parameter size varies across different action types.**
> > >    While the 3B model maintains its advantage in both modes, the performance gap between 1B and 3B models narrows with Code Block actions. This smaller gap likely reflects the increased complexity of code generation, as our earlier experiments showed that Code Block actions can potentially hurt smaller, less capable models while benefiting more advanced ones.
> > >
> > > ---
> > >
> > > ### Q4. For the Direct IO baseline, would outputting the output following the standard of the answer be the main issue? Do you provide the possible relationships in any way?
> > >
> > > In our experiments, the input context of Direct IO does not include a listing of CAMEO ontology, therefore, the Direct IO baseline relies on the model's inherent knowledge of CAMEO ontology and world event knowledge to predict relationships between entities. Notably, we observed that the Direct IO approach successfully generates answers in the required JSON format, demonstrating strong instruction-following capabilities and a basic understanding of the CAMEO ontology.
> > >
> > > To study whether the limited knowledge about CAMEO is a limiting factor for DirecIO performance, we conduct additional experiments that augment the CAMEO ontology to the context, as shown by DirectIO + CAMEO in the table in our reply to O5. We find that adding the CAMEO dictionary with full relation descriptions does not always improve the forecasting performance. The lengthy CAMEO dictionary might actually complicate the task and affect LLM performance.

---

> ### Author Response · Authors · 2024-11-25
> **Inquery for Discussion**
>
> Dear Reviewer SzJ2
>
> We greatly appreciate your thorough review and the insightful comments you've provided. Here is an overview of our responses to your feedback:
>
> - **Adjustment of Appendix**: We have streamlined the appendix by removing extensive API details and moving them to a GitHub repository, while keeping essential information in the paper. Human evaluation results have been moved to the main document for visibility.
> - **Highlight of Technical Contribution**: We've clarified that our core contribution is the establishment of an agentic benchmark for event forecasting, focusing on the design of an interactive environment with diverse information sources and a dynamic data pipeline. We summarized our contribution in our [global comments](https://openreview.net/forum?id=gzzX4ZeErx&noteId=ZueduytQF6).
> - **Scope of the Task**: We clarified that the task of hierarchical multi-relation forecasting is broad in its diversity, challenging for current models, and provides valuable insights. We see this as a foundational step before scaling to more complex forecasting.
> - **Figure Readability**: Figures have been revised for better readability, especially for black and white printing.
> - **Human Evaluation**: We justified the current evaluation scope, emphasizing the resource-intensive nature and complexity of the task. We compared to prior benchmarks, the accuracy highlights the effectiveness of our data cleaning process, demonstrating the quality of our dataset. We also moved human forecasting performance to Table 2 of the maim paper.
> - **Confidence Intervals Discussion**: We've added statistical analysis to clarify the significance of our experimental results, addressing the issues with overlapping confidence intervals.
> - **Inclusion of RAG Baseline**: We've evaluated multiple RAG baselines, which reveals that, unlike RAG’s fixed retrieval strategies, agents dynamically adapt their information-gathering and reasoning, offering greater flexibility, but also coming with higher challenges in multi-step planning and information integration.
> - **Minor Corrections**: We've fixed the citation error and updated the color scheme in Figure 8b for better intuitiveness.
> - **News Articles**: We highlighted the use metadata and titles with plans for further review of news content if full texts are needed.
> - **Balancing Test Split**: We've explained how our balanced sampling maintains fairness and diversity in the test set while still reflecting real-world event distribution.
> - **Impact of Model Parameters**: We've included experiments with different model sizes from the same family to demonstrate the influence of parameter count on performance.
> - **Direct IO Baseline**: We've discussed how providing the CAMEO ontology affects performance, suggesting that model knowledge and instruction-following are critical, not just the format of the output.
>
> We hope these clarifications and additional experiments address your concerns and provide a stronger foundation for our work. If there are any further points you would like us to address or clarify, please let us know. Thank you again for your time and effort in reviewing our work.
>
> Sincerely,
> Authors

---

> ### Author Response · Authors · 2024-11-29
> **Gentle Reminder: Awaiting Your Response to Our Rebuttal**
>
> Dear Reviewer SzJ2,
>
> I hope this message finds you well. We recently submitted our rebuttal and would like to kindly request your feedback on our responses.
>
> We understand that your schedule is demanding and greatly appreciate the time and effort you dedicate to the review process. Your insights are invaluable to us, and we are eager to address any further questions or concerns you may have.
>
> Thank you for your attention to this matter. We look forward to your response.
>
> Best regards,
>
> Authors

---

> ### Author Response · Authors · 2024-12-03
>
> Dear Reviewer SzJ2,
>
> Thank you again for taking the time to review our paper. We greatly appreciate your thoughtful feedback and your support and recognition of our work, particularly regarding the comprehensive description of the data and code, extensive experiments, insightful analysis, and clear overall writing with many illustrative examples.
>
> In response to your feedback, we have conducted extensive experiments with augmented Direct QA and various RAG methods, and studied the effect of model parameter sizes. We have also provided detailed clarifications to each of your concerns in our rebuttal replies, with our paper revised accordingly. We hope our responses have comprehensively addressed your concerns.
>
> With the discussion deadline approaching, we would appreciate your attention to our rebuttal and any further feedback you may have, as it will give us the opportunity to provide more details before the author-reviewer discussion session ends and help us continue to improve our work. Thank you again for your time and valuable comments!
>
> Best regards,
> Authors

---

### Official Review · Reviewer_uDbL · 2024-11-02

**Soundness:** 3
**Presentation:** 3
**Contribution:** 2
**Rating:** 5
**Confidence:** 4

**Summary:**

This paper introduces MIRAI, a benchmark to test LLMs for events forecasting. The article provides a detailed description of the process for generating predictions from LLMs which are based on recently extracted news and geopolitical events. This is based on a strategy called ReAct that includes the three steps of thinking, acting, and observing. An ontology called CAMEO is used for the geopolitical events analysis. Extensive experiments have been performed to compare MIRAI to other approaches and strategies. Different Large Language Models have been used to evaluate their performance.

**Strengths:**

- The paper is well-written.
- Extensive experiments have been provided. These include comparing MIRAI process against other strategies and temporal reasoning
  benchmarks and comparing base LLMs and smaller Language Models using MIRAI.
- The results of these experiments seem promising.

**Weaknesses:**

- Only one concrete example is provided and discussed in the paper. More examples would facilitate a better understanding of the results.  For instance, examples that showcase different types of forecasting scenarios or that highlight the strengths of MIRAI could have been included.
- The work should have been also compared with RAG-based (Retrieval-Augmented generation) approaches. It is unclear how MIRAI's approach differs from RAG-based methods.
- Typos  line 088: ".. is able to better utilizes and benefits... " ->  ".. is able to better utilize and benefit... "

**Questions:**

How LLMs would perform if the news were used as a direct source instead of following the ReAct strategy?

---

> ### Author Response · Authors · 2024-11-23
>
> We sincerely thank you for your constructive feedback. We have carefully considered each concern you raised and present detailed clarifications below:
>
> ---
>
> ### W1.  Only one concrete example is provided and discussed in the paper.
> We would like to clarify that we provided comprehensive examples throughout the paper:
> - In the main paper, Figure 1 provides a demonstrative example with discussions provided in lines 151-157.
> - In the footnote of page 1, we further provided an interactive agent demo with MIRAI at https://drive.google.com/file/d/1kKvYdAYv5hed-sbF_QE1moP-dQbfYoXI/view.
> - In Appendix F "Additional Forecasting Examples of LLM Agent", we detailed the entire LLM outputs showcasing the agent ReAct framework with code block (Appendix F.1)  as well as with single-function (Appendix F.2). Each example includes the full context, agent's step-by-step reasoning, and final predictions.
>
>   We placed these detailed additional examples in the Appendix to maintain focus in the main paper while providing interested readers with comprehensive documentation.
>
> Specifically, these examples have well demonstrated the strength and uniqueness of our MIRAI benchmark and using LLM agents for forecasting. For exmaple, in the first step of the agent in example Appendix F.1, it generates the following thoughts:
>
> """To forecast the relations that Australia will take towards China on November 03, 2023, we need to gather historical data on the interactions between these two countries. This includes examining the frequency and types of relations observed in the past, as well as any recent trends or significant events that might influence future relations.
>
> We will proceed with the following steps:
>
> 1. **Retrieve Historical Events**: Gather events involving Australia (AUS) and China (CHN) to understand the context and types of relations that have occurred in the past.
> 2. **Analyze Relation Distribution**: Analyze the distribution of different CAMEO codes in these events to identify the most common types of interactions.
> 3. **Examine Recent Trends**: Focus on recent events to detect any emerging trends or shifts in relations.
> 4. **Contextual Analysis**: Review news articles to gain insights into the context and underlying reasons for these relations.
>
> Let's start by retrieving historical events between Australia and China."""
>
> This example highlights how the MIRAI benchmark's rich, agentic environment enables LLM agents to:
>
> - **Automatically Plan Forecasting Strategies**: The agent crafts a tailored approach based on the specific query, demonstrating an ability to dynamically adjust its strategy.
> - **Analyse and Integrate Diverse Information Sources**: It leverages a variety of data from structured event to statistical distribution, and to textual news, showcasing its capability to synthesize different information through MIRAI's rich database for detailed analysis. This is also supported by the comprehensive APIs defined in MIRAI's agentic environment.
>
>
> In the follow-up steps in this example, by adhering to the outlined steps, the agent not only gathers and analyzes data but also engages in reasoning to its forecasting answer, which is crucial for predictive accuracy.
>
>
> In our revision, we additionally included two concrete examples with full reasoning traces of GPT-4o-mini based LLM agents in Appendix F.3 and F.4. We also make all of its reasoning logs and results, using Single Function or Code Block on the 2024-02 test split, publically available under this anonymous drive folder: https://drive.google.com/drive/folders/1fIVmv5EE-1qxvD8QRcj9K3QIIRpeTQPx?usp=sharing
>
> ---
>
> ### W2. The work should have been also compared with RAG-based (Retrieval-Augmented generation) approaches. It is unclear how MIRAI's approach differs from RAG-based methods.
> Thank you for this valuable suggestion for incorporating RAG-based methods into our dataset’s evaluation. We have now added RAG evaluations to provide additional context (see detailed results below). However, we respectfully note that our manuscript's core contribution is not proposing a forecasting method, but rather a benchmark dataset with an agentic environment for evaluating LLM agents' forecasting capabilities. We had explained our scope in the manuscript (lines 10-14, 21-25, and 90-96). Our goal is to establish a foundational evaluation framework that can drive future developments in this emerging field for agent-based approaches. Please refer to our global comment for a detailed discussion on our contribution.
>
> We also want to note that retrieval is inherently a critical tool for LLM agents [1, 2]. RAG represents a specific type of augmented LLM, and the key difference is that RAG always performs the retrieval augmentation as part of generation while agents automatically choose whether and when to use the tool (see our detailed discussion below).
>
> [1] Tool Learning with Large Language Models: A Survey
> [2] A Survey on Large Language Model based Autonomous Agents

---

> ### Author Response · Authors · 2024-11-23
>
> Follow the suggestion,  we conduct additional experiments with different retrieval augmented generation (RAG) methods. The result table is as follows:
>
> **Table:** Evaluation results of GPT-4o-mini on the 2024-02 test split using different non-agentic methods and the ReAct agent with Single Function action type. The best-performing score is highlighted in **bold** and the second-best is in *italics*.
>
> | Method      | Augmented Context | Agent API     | Binary KL (↓)      | Quad KL (↓)       | First-level Relation (%) - Pre. (↑) | Rec. (↑)       | F1 (↑)       | Second-level Relation (%) - Pre. (↑) | Rec. (↑)       | F1 (↑)       |
> |-------------|-------------------|---------------|--------------------|-------------------|-------------------------------------|----------------|--------------|-------------------------------------|----------------|--------------|
> | Direct IO   | ---               | ---           | 3.6±1.0           | 7.6±1.9          | 39.5±3.2                           | 44.8±3.2       | 34.9±3.5     | 15.4±0.8                           | 23.9±3.6       | 15.4±0.2     |
> | RAG         | *Events-Only*    | ---           | **2.2±0.9**       | **5.9±2.0**      | 57.5±3.5                           | **53.4±3.4**   | **50.5±3.8** | 32.4±1.1                           | **43.9±2.0**   | **33.2±1.4** |
> |             | *News-Only*      | ---           | 9.1±2.8           | 12.7±2.9         | 47.2±0.8                           | 23.2±2.4       | 25.4±0.2     | 19.5±2.0                           | 14.9±2.1       | 13.4±0.8     |
> |             | *All*            | ---           | *2.3±1.4*         | *6.3±2.0*        | 59.0±1.2                           | *48.1±1.2*     | *46.7±0.4*   | 36.4±5.3                           | *38.8±1.2*     | *32.1±2.4*   |
> | ReAct       | ---              | *Event-Only*  | 3.3±0.8           | 7.7±1.4          | **62.8±10.5**                       | 39.0±0.8       | 41.7±5.3     | *44.2±3.3*                         | 37.0±0.8       | 30.7±0.9     |
> |             | ---              | *News-Only*   | 6.5±1.7           | 13.0±2.1         | 41.5±6.1                           | 16.8±0.7       | 20.2±1.9     | 17.8±0.2                           | 12.2±1.0       | 12.5±0.5     |
> |             | ---              | *All*         | 3.6±0.9           | 8.0±1.5          | *61.7±10.1*                         | 38.6±1.9       | 40.7±5.6     | **46.3±4.4**                        | 32.9±3.8       | 31.1±2.6     |
>
>
> **Methods and Experimental Setup:** Besides the **Direct IO** (Direct QA) and three **ReAct** agents with different tool-use that we already implemented in the paper, we add the following three RAG baselines following recent work:
>
>    - **RAG Events-Only**: Following GPT-NeoX-ICL[3], we explore rule-based approaches for retrieving historical facts. Using the 'Pair' and 'Undirectional' setting, given a query event $(s, ?, o, t)$, we retrieve historical events $(s, r \in \mathcal{R}, o, <t)$ and $(o, r \in \mathcal{R}, s, <t)$. Events are sorted by recency with a cap of 30, aligning with the default cap of the `get_events` API function.
>    - **RAG News-Only**: Following TCELongBench[4], we employ BM25 retrieval to fetch the most query-relevant news articles before the query date. The top 15 news articles are retained, matching the default cap of the `get_news_articles` API function.
>    - **RAG All**: Combines both retrieved structured events and textual news articles.
>
>
> **Comparison of Retrieval Strategies with Agent:** RAG methods employ static, predefined retrieval strategies that are fixed for all queries and executed only once per query.
>
> In contrast, our agentic approach enables dynamic, multi-step information gathering and reasoning. The agent **can** replicate the baseline retrieval strategy by fixing certain API parameter values, for example, the agent can use function call `get_events(head_entities=[s, o], tail_entities=[o,s])` to get the retrieved context as RAG Event-Only, and use function call `get_news_articles(text_description='(t, s, ?, o)')` to get the retrieved context as RAG News-Only; its capabilities extend far beyond these static approaches through its flexible parameter settings for each function call and multiple steps per query.
>
> This multi-step, adaptive approach represents a fundamental shift from static retrieval to dynamic information gathering and reasoning, though it introduces higher requirements for the LLM's planning ability in:
> - Automatically selecting optimal information-gathering strategies
> - Integrating and reasoning over information of different formats
> - Adjusting strategies based on intermediate findings and current context
>
>
> [3] Temporal Knowledge Graph Forecasting Without Knowledge Using In-Context Learning. EMNLP 2023.
>
> [4] TCELongBench: Analyzing Temporal Complex Events with Large Language Models? A Benchmark towards Temporal, Long Context Understanding. ACL 2024.

---

> ### Author Response · Authors · 2024-11-23
>
> **Key Findings and Analysis:** Our experiment results reveal several important insights:
>
> - **Performance of the RAG Baselines**
> RAG demonstrates improved precision over Direct IO (15.4% Pre in second-level) when using either event (32.4% Pre in second-level) or text (19.5% Pre in second-level) information source independently, with event data contributing more significantly to recall (43.9% Rec in second-level). When combining the two information sources, RAG achieves higher precision (36.4% Pre in second-level) but with a lower recall than RAG Event Only (from 43.9% to 38.8%), leading to a lower overall F1 score (from 33.2% to 32.1%), this suggests that a simple combination of both information in the context not effectively and collaboratively contribute to a better forecasting performance.
>
>
> - **Comparison and Insights for Agent**
> ReAct agents exhibit similar performance patterns with RAG when using different information sources (changed by the type of API functions available for the agent). Specifically, event data also contributes to high precision for the agent (62.8% Pre at first-level and 44.2% at second-level), outperforming RAG baselines.
> The structured event data consistently provides stronger signals for forecasting across both approaches, likely due to its standardized format.
>
>   However, agent baselines generally obtain a lower recall than RAG baselines. Meanwhile, while ReAct with full API access achieves higher average F1 scores in second-level relation prediction than its partial access performance(31.1% than 30.7% and 12.5%), the benefits of combining information sources aren't consistent across all relation hierarchies and methods, similar to RAG, suggesting substantial room for exploring more effective information integration strategies.
>
>   This reveals both promises and challenges of the agent's more flexible retrieval approach: RAG's predefined and fixed retrieval strategies can often yield stable performance, agent's dynamic and multi-step retrieval allows for flexible information gathering and integration, yet higher requirements for planning and reasoning sometimes also lead to relatively lower performance.
>
> These observations underscore the core purpose of our benchmark: not just to compare current methods but to encourage the development of more advanced agentic forecasting approaches. The current performance patterns suggest significant opportunities for improving agent architectures, particularly in:
>
> - Developing more robust and automatic planning strategies for multi-step information gathering
> - Improving information integration capabilities across different information sources, formats, and temporal scales
>
> ---
>
> ### W3. Typos line 088: ".. is able to better utilizes and benefits... " -> ".. is able to better utilize and benefit... "
>
> Thank you for catching this grammatical error. We made the correction in the updated revision.
>
>
> ---
>
> ### Q1. How LLMs would perform if the news were used as a direct source instead of following the ReAct strategy?
>
> We have investigated how LLMs perform with direct news usage through both RAG News-Only and ReAct News-Only approaches, as shown in the above table. Our analysis reveals several key findings:
>
> When using news as the only information source, RAG News-Only achieves better performance (47.2% precision, 23.2% recall, 25.4% F1 at first-level relations) compared to ReAct News-Only (41.5% precision, 16.8% recall, 20.2% F1). This suggests that for unstructured news information, a fixed but focused retrieval strategy might be more effective than flexible but complex agent reasoning.
>
> However, both methods show relatively lower performance compared to their event-based counterparts (RAG Event-Only: 50.5% F1, ReAct Event-Only: 41.7% F1 at first-level relations), indicating that structured event data provides stronger signals for forecasting. Notably, when given access to both information sources, the agent method demonstrates potential for effective information integration (achieving 46.3% F1 at first-level prediction with full API access), though the challenges in reliably combining different information types suggest opportunities for improving future agentic approaches.

---

> ### Author Response · Authors · 2024-11-25
> **Inquery for Discussion**
>
> Dear Reviewer uDbL,
>
> Thank you once again for your thoughtful feedback and for the opportunity to further clarify our work. We hope our responses and additional experiments have addressed your questions comprehensively. Specifically::
>
> - **Comprehensive Examples**: We have clarified the presence of various examples in our paper, including a demonstrative example in Figure 1, an interactive agent demo, and extensive examples in Appendix F. We've also made additional data available publicly to further illustrate our methodology.
> - **Comparison with RAG Methods**: We've included comparisons with multiple RAG-based methods. Our experimental results, detailed in the table provided, show how our agentic approach complements and differs from RAG strategies, particularly in terms of dynamic information handling and reasoning. We've reiterated that our primary contribution is the benchmark itself, aimed at fostering future research in agent-based forecasting.
> - **Grammatical Correction**: We appreciate your attention to detail and have corrected the typo on line 88 in our revised manuscript.
> - **Direct Use of News**: We explored the performance implications of using news directly as a source, both through RAG and our ReAct agent approaches. Our findings suggest that while RAG might perform better with its predefined retrieval strategy, the agent method shows promise in automatic planning and integrating multiple data types, highlighting areas for future improvement.
>
> We hope our clarifications and additional experiments have addressed your concerns comprehensively. If there are any further points you would like clarification on, please do not hesitate to let us know.
>
> We look forward to any further feedback and sincerely appreciate the time and effort you've put into reviewing our work.
>
> Sincerely,
> Authors

---

> ### Author Response · Authors · 2024-11-29
> **Gentle Reminder: Awaiting Your Response to Our Rebuttal**
>
> Dear Reviewer uDbL,
>
> I hope this message finds you well. We recently submitted our rebuttal and would like to kindly request your feedback on our responses.
>
> We understand that your schedule is demanding and greatly appreciate the time and effort you dedicate to the review process. Your insights are invaluable to us, and we are eager to address any further questions or concerns you may have.
>
> Thank you for your attention to this matter. We look forward to your response.
>
> Best regards,
>
> Authors

---

> ### Comment · Reviewer_uDbL · 2024-12-02
>
> Thanks to the authors for the thorough response. Having reviewed the authors' response I stand by my score for this work.

---

> > ### Author Response · Authors · 2024-12-02
> > **Seeking Further Feedback to Address Any Remaining Concerns**
> >
> > Dear Reviewer uDbL,
> >
> > Thank you for taking the time to review our responses. We greatly appreciate your engagement with our work and the opportunity to address your initial concerns.
> >
> > We noticed that you’ve chosen to maintain your initial score. If there are any specific points or concerns that you feel were not fully addressed in our rebuttal, we would be very grateful if you could share them with us. We are eager to ensure that all your concerns are comprehensively addressed.
> >
> > Thank you again for your time and effort in reviewing our submission. We deeply value your time and expertise.
> >
> > Best regards,
> > Authors

---

### Official Review · Reviewer_AZdo · 2024-11-04

**Soundness:** 3
**Presentation:** 3
**Contribution:** 3
**Rating:** 6
**Confidence:** 4

**Summary:**

The paper proposes a new LLM benchmark for short- and longterm forcasting international events, which is essentialy trying to predict relations between international parties. The general and fine-grained relations stem from an ontology, namely CAMEO. The authors define a think, act, oberserve loop, where the agent processes the query at hand and can give an answer right away, otherwise can act to get more information via coding and eventually observes the executed code. The benchmark equips agents to act by either single functions for data retrieval or code blocks for writing more complex code.

The results show that the task is challenging for LLMs by evaluating a set of baseline LLMs, where GPT-4o mini ends up performing the best. Next to the baseline LLMs, the authors also show the performance of fine-tuned methods from other benchmarks, underlining the benchmark is challenging. In addition, the authors run additional experiments and analyses, including observed generated code errors, boosting smaller architectures or investigating chosen action orders.

**Strengths:**

- Novel relation prediction task formulation (wrt other benchmarks)
- Sensibe LLM system design, enabling the to query for information or to write own code, and defining an agent-based behavioural loop
- Sufficiently broad evaluation, including diverse analyses and also including baselines from other competing benchmarks
- Empirical results are promising

**Weaknesses:**

- Unclear if, beyond the integrated fine-tuned approaches from other works, the results for LLMs are superior if compared to other task formulations such as QA. Since LLMs have been used there as well, the question remains open if other types of system components or prompts significantly worked better in the past

**Questions:**

- Did you also try a fine-tuned the baseline LLMs (given their are open-source)?
- Which competitor in the related work overview would be strongest compared to your approach?
- From the related work it becomes quite clear what the conceptual differences to prior LLM QA works are, but how much better does the new agent design actually work compared to more straightforward prior approaches?

---

> ### Author Response · Authors · 2024-11-23
>
> We sincerely appreciate your support and constructive feedback on our MIRAI benchmark.
>
> First of all, we’d like to emphasize that the main contribution of MIRAI is to provide a **benchmark** dataset to evaluate all kinds of LLM forecasting systems, including Simple QA format (the Direct IO in our experiment), LLM with tool-use, with and without fine-tuning, etc. In our rebuttal, we added recurrency-based baselines, fine-tuned TKG baselines (as shown in Response [Table 1](https://openreview.net/forum?id=gzzX4ZeErx&noteId=YQRn01vvCq)) and RAG baselines as shown below.
>
> Please find our point-by-point responses to your questions below:
>
> ---
>
> ### W1.  Are LLMs in your approach demonstrably better than in other task formulations like QA, considering prior methods or prompts?
>
> Thank you for this valuable question. In our submission-version, we have already evaluated direct question answering (direct IO) and question answering with zero-shot chain-of-thought prompting (ZS-COT) in experiment Table 1. Our results show that allowing LLMs to interact in an agentic environment with diverse information sources improves performance by providing richer context for predictions.
>
> To further address the raised concern, we additionally include more baseline methods that are used in previous QA-based studies, particularly augmenting context of direct IO (i.e. direct QA) and QA with retrieval augmented generation (RAG) methods. The result table is as follows:
>
> **Table:** Evaluation results of GPT-4o-mini on the 2024-02 test split using different non-agentic methods and the ReAct agent with Single Function action type. The best-performing score is highlighted in **bold** and the second-best is in *italics*.
>
> | Method      | Augmented Context | Agent API     | Binary KL (↓)      | Quad KL (↓)       | First-level Relation (%) - Pre. (↑) | Rec. (↑)       | F1 (↑)       | Second-level Relation (%) - Pre. (↑) | Rec. (↑)       | F1 (↑)       |
> |-------------|-------------------|---------------|--------------------|-------------------|-------------------------------------|----------------|--------------|-------------------------------------|----------------|--------------|
> | Direct IO   | ---               | ---           | 3.6±1.0           | 7.6±1.9          | 39.5±3.2                           | 44.8±3.2       | 34.9±3.5     | 15.4±0.8                           | 23.9±3.6       | 15.4±0.2     |
> |             | CAMEO            | ---           | 5.0±1.5           | 7.3±1.9          | 35.5±4.6                           | 32.6±2.2       | 28.6±2.7     | 10.0±1.5                           | 14.2±0.9       | 10.1±0.8     |
> | RAG         | *Events-Only*    | ---           | **2.2±0.9**       | **5.9±2.0**      | 57.5±3.5                           | **53.4±3.4**   | **50.5±3.8** | 32.4±1.1                           | **43.9±2.0**   | **33.2±1.4** |
> |             | *News-Only*      | ---           | 9.1±2.8           | 12.7±2.9         | 47.2±0.8                           | 23.2±2.4       | 25.4±0.2     | 19.5±2.0                           | 14.9±2.1       | 13.4±0.8     |
> |             | *All*            | ---           | *2.3±1.4*         | *6.3±2.0*        | 59.0±1.2                           | *48.1±1.2*     | *46.7±0.4*   | 36.4±5.3                           | *38.8±1.2*     | *32.1±2.4*   |
> | ReAct       | ---              | *Event-Only*  | 3.3±0.8           | 7.7±1.4          | **62.8±10.5**                       | 39.0±0.8       | 41.7±5.3     | *44.2±3.3*                         | 37.0±0.8       | 30.7±0.9     |
> |             | ---              | *News-Only*   | 6.5±1.7           | 13.0±2.1         | 41.5±6.1                           | 16.8±0.7       | 20.2±1.9     | 17.8±0.2                           | 12.2±1.0       | 12.5±0.5     |
> |             | ---              | *All*         | 3.6±0.9           | 8.0±1.5          | *61.7±10.1*                         | 38.6±1.9       | 40.7±5.6     | **46.3±4.4**                        | 32.9±3.8       | 31.1±2.6     |

---

> ### Author Response · Authors · 2024-11-23
>
> **Methods and Experimental Setup:** Besides the **Direct IO / QA**, that we already implemented in the paper, where the LLM directly provides answers using only its internal knowledge without chain-of-thought, we add the following baselines:
>
> - **Direct QA with Augmentation**
>    - **QA with CAMEO**: We provide the CAMEO ontology in an ordered dictionary format mapping relation codes to their names and detailed descriptions. This is closer to the QA-format the authors mention, and a more fair comparison as the model can refer to the output vocabulary without needing to memorize CAMEO codes.
>
> - **RAG Methods**
> Following recent work, we implement three RAG variants:
>    - **RAG Events-Only**: Following GPT-NeoX-ICL[1], we explore rule-based approaches for retrieving historical facts. Using the 'Pair' and 'Undirectional' setting, given a query event $(s, ?, o, t)$, we retrieve historical events $(s, r \in \mathcal{R}, o, <t)$ and $(o, r \in \mathcal{R}, s, <t)$. Events are sorted by recency with a cap of 30, aligning with the default cap of the `get_events` API function.
>    - **RAG News-Only**: Following TCELongBench[2], we employ BM25 retrieval to fetch the most query-relevant news articles before the query date. The top 15 news articles are retained, matching the default cap of the `get_news_articles` API function.
>    - **RAG All**: Combines both retrieved structured events and textual news articles.
>
>
> **Comparison of Retrieval Strategies with Agent:** Both baseline approaches—CAMEO context augmentation and RAG methods—employ static, predefined retrieval strategies that are fixed for all queries and executed only once per query.
>
> In contrast, our agentic approach enables dynamic, multi-step information gathering and reasoning. The agent **can** replicate the baseline retrieval strategy by fixing certain API parameter values, for example, the agent can use function call `get_events(head_entities=[s, o], tail_entities=[o,s])` to get the retrieved context as RAG Event-Only, and use function call `get_news_articles(text_description='(t, s, ?, o)')` to get the retrieved context as RAG News-Only; its capabilities extend far beyond these static approaches through its flexible parameter settings for each function call and multiple steps per query.
>
> This multi-step, adaptive approach represents a fundamental shift from static retrieval to dynamic information gathering and reasoning, though it introduces higher requirements for the LLM's planning ability in:
> - Automatically selecting optimal information-gathering strategies
> - Integrating and reasoning over information of different formats
> - Adjusting strategies based on intermediate findings and current context

---

> > ### Author Response · Authors · 2024-11-23
> >
> > **Key Findings and Analysis:** Our experiment results reveal several important insights:
> >
> > - **Comparision of Direct QA with/without CAMEO**
> > Both Direct IO variants successfully generate valid three-digit second-level relation codes, but adding the CAMEO dictionary with full relation descriptions does not improve forecasting performance (15.4% vs. 10.1% F1 score). The lengthy CAMEO dictionary might complicate the task and affect LLM performance. Our manual analysis of Direct IO with CAMEO (we make the all results of gpt-4o-mini on the 20224-02 test split available in https://drive.google.com/drive/folders/1fIVmv5EE-1qxvD8QRcj9K3QIIRpeTQPx?usp=sharing), reveals an interesting pattern: with detailed descriptions, the LLM tends to polarize or intensify relationship predictions between countries based on its world knowledge, favoring significant relations under Material Cooperation and Material Conflict categories over the more common Verbal Cooperation and Verbal Conflict categories.
> >
> > - **Performance of the RAG Baselines**
> > RAG demonstrates improved precision over Direct IO (15.4% Pre in second-level) when using either event (32.4% Pre in second-level) or text (19.5% Pre in second-level) information source independently, with event data contributing more significantly to recall (43.9% Rec in second-level). When combining the two information sources, RAG achieves higher precision (36.4% Pre in second-level) but with a lower recall than RAG Event Only (from 43.9% to 38.8%), leading to a lower overall F1 score (from 33.2% to 32.1%), this suggests that a simple combination of both information in the context not effectively and collaboratively contribute to a better forecasting performance.
> >
> >
> > - **Comparison and Insights for Agent**
> > ReAct agents exhibit similar performance patterns with RAG when using different information sources (changed by the type of API functions available for the agent). Specifically, event data also contributes to high precision for the agent (62.8% Pre at first-level and 44.2% at second-level), outperforming RAG baselines.
> > The structured event data consistently provides stronger signals for forecasting across both approaches, likely due to its standardized format.
> >
> >   However, agent baselines generally obtain a lower recall than RAG baselines. Meanwhile, while ReAct with full API access achieves higher average F1 scores in second-level relation prediction than its partial access performance(31.1% than 30.7% and 12.5%), the benefits of combining information sources aren't consistent across all relation hierarchies and methods, similar to RAG, suggesting substantial room for exploring more effective information integration strategies.
> >
> >   This reveals both promises and challenges of the agent's more flexible retrieval approach: RAG's predefined and fixed retrieval strategies can often yield stable performance, agent's dynamic and multi-step retrieval allows for flexible information gathering and integration, yet higher requirements for planning and reasoning sometimes also lead to relatively lower performance.
> >
> > These observations underscore the core purpose of our benchmark: not just to compare current methods but to encourage the development of more advanced agentic forecasting approaches. The current performance patterns suggest significant opportunities for improving agent architectures, particularly in:
> >
> > - Developing more robust and automatic planning strategies for multi-step information gathering
> > - Improving information integration capabilities across different information sources, formats, and temporal scales
> >
> >
> > [1] Temporal Knowledge Graph Forecasting Without Knowledge Using In-Context Learning. EMNLP 2023.
> >
> > [2] TCELongBench: Analyzing Temporal Complex Events with Large Language Models? A Benchmark towards Temporal, Long Context Understanding. ACL 2024.

---

> > > ### Author Response · Authors · 2024-11-23
> > >
> > > ### Q1. Did you also try a fine-tuned the baseline LLMs (given they are open-source)?
> > >
> > > Thank you for this insightful question. Fine-tuning LLM agents is indeed an exciting direction for future research. However, we did not fine-tune the baseline LLMs for this study due to the unique challenges posed by agentic fine-tuning. Unlike standard supervised learning tasks, fine-tuning agentic reasoning requires optimizing complex, multi-step sequences involving tool selection, information gathering, and reasoning. This process is particularly challenging in the absence of high-quality, ground-truth reasoning traces—a prerequisite for effective fine-tuning. Collecting such data, whether from human experts or advanced LLMs, is non-trivial and remains an open research problem. Even with the availability of data, fine-tuning for agentic reasoning is challenging, with a few recent works beginning to explore its potential [3-5].
> > >
> > > Moreover, our main contribution is establishing MIRAI as a new benchmark for evaluating forecasting capabilities, rather than proposing optimal methods. We view MIRAI as a platform for developing novel agentic forecasting methods, including future approaches to agent finetuning. Future directions could involve collecting high-quality reasoning data, leveraging advanced LLMs for self-training, or exploring other innovative methods to advance this field.
> > >
> > > [3] Re-ReST: Reflection-Reinforced Self-Training for Language Agents. EMNLP 2024.
> > >
> > > [4] Optima: Optimizing Effectiveness and Efficiency for LLM-Based Multi-Agent System
> > >
> > > [5] AgentTuning: Enabling Generalized Agent Abilities for LLMs. ACL 2024.
> > >
> > > ---
> > >
> > > ### Q2. Which competitor in the related work overview would be strongest compared to your approach?
> > >
> > > We would like to clarify that, our paper primarily introduces a new benchmark designed to facilitate the development and evaluation of agent forecasting methods. As such, the primary competitors in the related work are **existing benchmarks in this domain** (as we pointed out in Table 4) rather than specific state-of-the-art prediction methods.
> > >
> > > Among the existing benchmarks, the most relevant comparison can be made with TempLongBench[2] and the TKG dataset GDELT-2015. These benchmarks have contributed significantly to the field by providing platforms for evaluating temporal forecasting in QA-based formulations or link prediction tasks. However, they have limitations in uniform information source (either textual or structured) and lack of dynamic updates, and their task formulation cannot be easily adapted to the evaluation of agentic method.
> > >
> > > Our benchmark distinguishes itself by offering diverse and updated data sources, and introducing an agentic environment with comprehensive tools. This allows for the first comprehensive assessment of LLM agents in event forecasting and promotes further advancement in the field. By comparing our work with these existing benchmarks, we demonstrated how our benchmark is unique and addresses current gaps in the lack of dynamic data updates, diverse information sources, and agentic evaluation.
> > >
> > > ---
> > >
> > > ### Q3. From the related work it becomes quite clear what the conceptual differences to prior LLM QA works are, but how much better does the new agent design actually work compared to more straightforward prior approaches?
> > >
> > > Thank you for your question. As clarified in W1, we evaluated direct question answering (direct IO) and question answering with zero-shot chain-of-thought prompting (ZS-COT) in Table 1. Our results show that allowing LLMs to interact in an agentic environment with diverse information sources improves performance by providing richer context to make final predictions.
> > >
> > > Additionally, RAG is a common technique in prior LLM forecasting benchmarks formatted as QA tasks. For example, TCELongBench [2] explores LLMs with various textual embedding-based retrievers. In line with this, we employed a BM25 retriever to retrieve the most query-relevant news articles (with publication dates and titles) prior to the query date. These results are discussed further in our response to Q1.
> > >
> > > We also want to note that our goal is not to claim that the agentic method universally outperforms prior approaches. Instead, our objective is to establish a benchmark that promotes the development of advanced agentic forecasting methods while offering valuable insights into the benefits of agentic reasoning.

---

> ### Author Response · Authors · 2024-11-25
> **Inquery for Discussion**
>
> Dear Reviewer AZdo,
>
> Thank you again for your support and thoughtful feedback. We deeply appreciate the time you have dedicated to evaluating our work. Below, we provide a summary of our additional experiments and clarifications, hope these have addressed your questions comprehensively:
>
> 1. **Comparison with QA-based Task Formulations:** In addition to direct question answering (Direct IO) and with zero-shot chain-of-thought prompting (ZS-COT) we evaluated, we've expanded our evaluation to include more baselines from the QA domain, including Direct QA with Augmentation and various RAG methods. These comparisons reveal that, unlike RAG’s fixed retrieval strategies, agents dynamically adapt their information-gathering and reasoning, offering greater flexibility, but also coming with higher challenges in multi-step planning and information integration
>
> 2. **Fine-tuning Baseline LLMs:**  We acknowledged the potential of fine-tuning baseline LLMs but have clarified the complexities involved, particularly for agentic systems. We've cited recent works exploring this area and emphasized that MIRAI serves as a platform for such innovations. We will include a discussion on this in our revision to encourage further research.
>
> 3. **Comparison with Related Benchmarks:** We've clarified that our main competitors are benchmarks rather than methods, specifically comparing with TempLongBench and GDELT-2015.  We've highlighted how MIRAI fills gaps in these benchmarks by introducing dynamic updates, diverse information sources, and an agentic environment with tool-based reasoning.
>
> 4. **Performance of Agentic Design Compared to Prior Approaches:** We've reiterated the findings from our initial experiments and added comparisons with RAG and other QA methods, showing improvements in contexts where agentic reasoning provides additional value.
>
> Additional Points:
>
> - **Dynamic Information Use**: We've demonstrated that our agentic approach allows for more automatic and adaptive information retrieval, which is not with static, one-shot retrieval methods used in previous works.
> - **Benchmark Contribution**: We've underscored that our primary contribution is the benchmark itself, promoting new research directions in agentic forecasting.
>
> We hope these responses and experiments in our rebuttal have been helpful. If there are any further questions or areas you feel need clarification, please let us know. We are eager to refine our work based on your feedback. Thank you once again for your time and the detailed review of our work.
>
> Sincerely,
> Authors

---

> > ### Comment · Reviewer_AZdo · 2024-11-26
> >
> > Thank you for providing such thorough answers to my questions. I believe the novel experimental results provide added value, especially since they seem to be integrated into the novel version of the paper. The clarifications wrt benchmark comparisons is sensible.

---

> > > ### Author Response · Authors · 2024-11-26
> > > **Reply to Reviewer's Rebuttal Feedback**
> > >
> > > Dear Reviewer AZdo,
> > >
> > > Thank you sincerely for your positive feedback and acknowledgment of our rebuttal responses. As we addressed your concerns, would you like to increase the rating to support our paper?
> > >
> > > Best,
> > > Authors

---

### Official Review · Reviewer_ZMPf · 2024-11-07

**Soundness:** 2
**Presentation:** 3
**Contribution:** 2
**Rating:** 5
**Confidence:** 3

**Summary:**

The paper presents MIRAI, a benchmark designed to assess the performance of LLM agents in predicting international event relations. It emphasizes APIs to enable LLM agents to utilize different tools, a refined GDELT event database, and a dynamic data construction pipeline to ensure contamination-free test sets, aiming to provide a reliable standard for assessing the capabilities of LLM agents in forecasting international events.

**Strengths:**

•	Clarity and Structure: The paper is clearly written and easy to follow, which facilitates understanding and replicability.
•	Contamination-Free Test Sets: The focus on providing contamination-free test sets is a significant strength, ensuring unbiased performance evaluation. Regular updates to the test sets also add to its robustness and relevance over time, when newer models should be evaluated.
•	Insightful Visuals: Figure 2 offers a valuable overview of the distribution of relation types within the dataset.
•	Detailed Analysis: The authors provide an in-depth result analysis across diverse categories, such as action order and accuracy across different relation types

**Weaknesses:**

1.	Absence of Baseline Comparisons: As a benchmark paper, a comparison with simple (heuristic) baselines is critical to contextualize the agents performance in the MIRAI benchmark. The lack of such comparisons with simple baselines is a notable drawback, reducing the ability to assess the evaluated agents true effectiveness.  See suggestion 3).
2.	Focus Solely on Relation Prediction: The exclusive focus on relation forecasting seems narrow. Incorporating link prediction tasks (i.e., predicting specific subjects and objects for given queries) would align it better with existing TKG forecasting approaches and provide a more comprehensive benchmark. See question 1)
3.	Doubts in Experimental Setting when comparing to "traditional" models:
o	The "traditional" models that you compare to are a bit old, which may not reflect current advancements in temporal knowledge graph forecasting. Inclusion of more recent models, such as TiRGN (Time-Guided Recurrent Graph Network with Local-Global Historical Patterns for Temporal Knowledge Graph Reasoning), would enhance the relevance of comparisons.
o	I have some doubts about the evaluation on these models, see question 5. For example, why finetune it only until 2023-06, even though some agents have more recent cutoff-dates.

1.	Integrate Human Performance in Main Paper: Including the presented human forecasting performance in the main paper (currently in Appendix G.4) would strengthen the benchmark's validation, given the strong performance of human evaluators on most metrics.  I strongly suggest to include the human study in main paper.
2.	Condense the Appendix: I suggest to potentially not include the full API specification and implementation, and README of the github repo, but instead a link to the github repo containing these. This could streamline the paper, allowing readers to focus on MIRAI's core contributions.
3.	Include a simple Baseline for Comparison: To provide a more meaningful comparison, I strongly suggest to include a simple/ heuristic baseline to compare the agents to. For example, a baseline that predicts the same relations as previously occurred, e.g. a modification from [1] to relation prediction.
4.	References: AutoGPT Documentation: this should at least contain a date and a link.
 [1] Gastinger, J., Meilicke, C., Errica, F., Sztyler, T., Schuelke, A., & Stuckenschmidt, H. (2024). History repeats itself: A Baseline for Temporal Knowledge Graph Forecasting. International Joint Conferences on Artificial Intelligence.

**Questions:**

1.	Focus on Relation Forecasting: Why did the authors choose to focus solely on relation forecasting rather than including link prediction (predicting subjects and objects), which is more standard in TKG forecasting, and KG completion evaluation?
2.	Metric Selection: What motivated the choice of the current metrics, and why were more commonly reported metrics, like MRR and Hits@k, excluded?
3.	Dataset Update Commitment: The authors commit to updating the dataset split monthly. How long is this commitment expected to last, and will it be sustained indefinitely?
4.	Higher Thresholds for Test Data: What is the rationale behind applying higher thresholds to the test data (100 daily mentions, 5 news articles) than to the training data?
5.	Clarification on the "traditional models" experiments: Section 3.2 3):
o	What do you mean with "fine-tuned" in this case? Did you train the models on the provided GDELT event data?
o	Assuming that with "fine-tune" you mean train: Why call it fine-tuned?
o	Why do you fine-tune it only until 2023-6? considering that the query date is 2024-02, this is a very large gap. It seems also unfair, considering that e.g. for Llama-3.1-8B-Instruct the cutoff date is 2023-12.
o	Do you apply RE-GCN in single-step or multi-step prediction mode, i.e. for predicting t, do you feed ground truth up until t-1?

---

> ### Author Response · Authors · 2024-11-23
>
> **Table 1:** Evaluation results on the 2024-02 test split for relation prediction using TKG-based methods and LLM agents based on GPT-4o-mini. The best-performing score is highlighted in **bold**, and the second-best is in *italics*.
>
> | Method               | Training Data Cutoff Date | Setting               | MRR (%) (↑)   | Hit@10 (%) (↑) | Binary KL (↓) | Quad KL (↓) | First-level Relation (%) - Pre. (↑) | Rec. (↑)       | F1 (↑)         | Second-level Relation (%) - Pre. (↑) | Rec. (↑)       | F1 (↑)         |
> |----------------------|---------------------------|----------------------|---------------|----------------|---------------|-------------|-------------------------------------|----------------|----------------|-------------------------------------|----------------|----------------|
> | **RE-GCN**          | 2023-06                  | ---                  | 1.6           | 2.2            | *0.4*         | *0.8*       | 24.4                               | *90.6*         | 34.3           | 4.4                               | **83.9**       | 7.9           |
> |                      | 2023-08                  | ---                  | 1.9           | 2.8            | *0.4*         | 1.1         | 23.9                               | 86.1           | 32.9           | 4.6                               | 40.0           | 7.0           |
> |                      | 2023-10                  | ---                  | 1.7           | 2.5            | **0.3**       | 1.0         | 24.8                               | 78.2           | 32.4           | 3.9                               | 25.7           | 5.6           |
> |                      | 2023-12                  | ---                  | 2.9           | 5.7            | **0.3**       | 2.5         | 23.9                               | 74.4           | 31.3           | 5.5                               | 28.4           | 7.9           |
> | **Recurrency (Strict)** | 2023-06               | ---                  | **17.4**      | *45.0*         | 3.2           | 3.6         | 32.8                               | 77.1           | 42.9           | 18.7                              | 67.8           | 27.2          |
> |                      | 2023-08                  | ---                  | *17.1*        | **45.3**       | 3.2           | 3.6         | 32.3                               | 78.2           | 42.7           | 18.0                              | 69.9           | 26.9          |
> |                      | 2023-10                  | ---                  | 15.8          | 41.0           | 2.4           | 3.1         | 29.7                               | 83.5           | 41.3           | 14.3                              | 76.8           | 23.0          |
> |                      | 2023-12                  | ---                  | 17.8          | 43.2           | 2.1           | 2.5         | 29.8                               | 86.0           | 41.6           | 14.2                              | 80.1           | 23.0          |
> | **ReAct**            | 2023-10                  | **Set Prediction**   | ---           | ---            | 3.6           | 8.0         | **61.7**                            | 38.6           | 40.7           | **46.3**                          | 32.9           | 31.1          |
> |                      |                           | Rank (k=10)         | ---           | 25.7           | 0.6           | 1.4         | *47.5*                              | 70.2           | **48.9**       | *38.1*                             | 61.8           | **38.2**      |
> |                      |                           | Rank (k=30)         | ---           | 12.0           | **0.3**       | *0.8*       | 34.9                               | **91.2**       | 45.8           | 22.5                              | *82.8*         | *31.7*        |
> |                      |                           | Rank (all)          | 13.9          | 14.1           | 2.1           | 2.8         | 27.0                               | 86.2           | 37.9           | 12.5                              | 81.4           | 20.2          |
> |                      |                           | Rank w.Prob (k=10)  | ---           | 26.8           | 1.1           | 2.5         | 47.3                               | 67.7           | *48.3*         | 37.9                              | 59.2           | **38.2**      |
> |                      |                           | Rank w.Prob (k=30)  | ---           | 10.8           | **0.3**       | **0.6**     | 34.8                               | 86.6           | 45.3           | 22.2                              | 76.4           | 31.0          |
> |                      |                           | Rank w.Prob (all)   | 12.6          | 14.9           | 2.4           | 2.7         | 28.5                               | 83.0           | 38.3           | 12.7                              | 78.6           | 20.6          |
>
> ---

---

> ### Author Response · Authors · 2024-11-23
>
> We sincerely thank you for your thoughtful and detailed feedback, for which we have included additional discussions and experiments accordingly. Please find our experiment results in the table and detailed responses for each of your questions as follows:
>
> ---
>
> ## Comparison with Heuristic-based Method:
> ### W1&W6&W7. Comparisons with simple heuristic baselines to contextualize the agents' performance in the MIRAI benchmark (W1). Include a heuristic baseline, such as predicting relations based on historical data (modification from [1]), to improve evaluation. (W6, W7)
>
> Thank you for raising this valuable suggestion of including the heuristic-based baseline, specifically the Reccurrency [1] model. The main idea in this paper is ‘history repeats itself’: events that happened in the past are likely to re-occur in the future.
>
> First of all, in our agent API design, we already have functions inspired by such re-currency heuristics. Specifically, we’ve defined functions like `count_events`, which count the number of events in the knowledge graph based on customizable conditions on date, entities, and relations. To provide more statistical support for the agent, we also include functions like `get_entity_distribution` and `get_relation_distribution` which return a frequency-ranked list of the most commonly interacted entities or happened relations.
>
> Secondly, we appreciate your suggestions on directly including the recurrency-based baseline to compare with LLM Agent, so we can better understand the limitations of existing agent design. Thus, we added the Reccurrency [1] model as one baseline. The original work introduces three settings: **strict recurrency, relaxed recurrency, and their combination**. While the original work and its scoring functions are specifically designed for link prediction, we adapt the strict recurrency variant for relation prediction. For a query event $(s,?,o,t)$, we compute scores for all relations $r \in \mathcal{R}$ using:
>
> $\phi_{\Delta}((s,r,o,t),G) =
> \Delta(t, \max\{k \mid (s,r,o,k) \in G\}) \text{ if } \exists k \text{ with } (s,r,o,k) \in G, \text{ else } 0,$
>
> where $\Delta(t,k) = k/t$ measures temporal proximity.
>
> **Cutoff Dates**: To fairly compare the performance with LLM agents and study the effect of cutoff date, we evaluate the recurrency model on 4 different cutoff dates, 2023-06, 2023-08, 2023-10, and 2023-12, where 2023-06 uses around half of the training data in MIRAI; 2023-10 is the knowledge cutoff date of GPT-4o-mini, the best-performing LLM we tested; 2023-12 is the knowledge cutoff date of Mistral and LLama, the open-sourced LLMs we tested. The cutoff date for the Recurrency model determines available historical events, e.g., 2023-10 means using only events before 2023-10-31 for score computation.
>
> **Evaluation Metrics**: For evaluation, the Recurrency model supports a full ranking list for all relation types with their prediction scores. Therefore, for **ranking-based metrics**, we apply MRR and Hit@10. We follow TKG forecasting conventions of time-aware filtering [2-3] for the ranking-based metrics. For **set-based evaluation**, we acknowledge that there are an infinite number of possible thresholds to select the set of relations with their score above the threshold. In the experiment table, we report the set prediction result with the threshold set to be 0 - this has an intuitive meaning: we predict the relations that happened at least once between the two countries.
>
> **Performance of the Recurrency Model**: The experiment results of the Reccurency (strict) model are shown in row 5-8 of the table. We observe that the Recurrency baseline demonstrates strong performance in ranking metrics (17.8% MRR and 43.2% Hit@10 with 2023-12 cutoff), leading other models; it also shows consistently high recall (86.0% Recall at first-level relation, and 80.1% recall at second-level relations with 2023-12 cutoff), suggesting that international events indeed often follow repetitive patterns.

---

> ### Author Response · Authors · 2024-11-23
>
> **Recurrency's Comparison and Insights for Agents**: Comparing with the ReAct agent (Set Prediction, the setting we used in the main paper) in row 9, we find that although the LLM agent could achieve much higher forecasting precision, it obtains much lower recall than the Recurrency baseline. We conducted a detailed analysis of this behavior by manually going through the reasoning traces generated by the LLM agent in the test set. One possible reason is that the agent has a strong tendency to select only a subset of the most frequent historical events in its prediction. For example, for query (2024-02-01, PSE, ? , EGY), it uses function calls like `get_relation_distribution(date_range=DateRange(start_date=Date("2023-01-31"), end_date=Date("2024-01-31")), head_entities=[ISOCode("PSE")], tail_entities=[ISOCode("EGY")])’, and obtained a full frequency list as: {CAMEOCode(code='042'): 32, CAMEOCode(code='192'): 18, CAMEOCode(code='040'): 13, CAMEOCode(code='043'): 12, CAMEOCode(code='046'): 8, CAMEOCode(code='080'): 6, CAMEOCode(code='036'): 4, CAMEOCode(code='010'): 4, CAMEOCode(code='190'): 3, CAMEOCode(code='073'): 3, CAMEOCode(code='030'): 3, CAMEOCode(code='084'): 3, CAMEOCode(code='020'): 3, CAMEOCode(code='172'): 2, CAMEOCode(code='014'): 2, CAMEOCode(code='070'): 2, CAMEOCode(code='044'): 2, CAMEOCode(code='086'): 1, CAMEOCode(code='013'): 1, CAMEOCode(code='051'): 1}`. It then has a further step of checking recent news articles, and obtains its final prediction as 040, 042, and 192, which are the top three frequent relations.
>
> The effectiveness of simple temporal recurrency heuristics underscores the importance of incorporating more historical pattern analysis in the future development of forecasting agents, in particular, improving its recall of capturing a greater proportion of true relationships between countries.
>
> ---
>
> [1] History Repeats Itself: A Baseline for Temporal Knowledge Graph Forecasting. IJCAI 2024.
>
> [2] Translating embeddings for modeling multi-relational data. Neurips 2013.
>
> [3]  xERTE: Explainable Reasoning on Temporal Knowledge Graphs for Forecasting Future Link. ICLR 2021.

---

> > ### Author Response · Authors · 2024-11-23
> >
> > ## Comparison with TKG Methods:
> >
> > Thank you for raising questions about TKG baselines that help us to clarify our experimental settings and analysis for the TKG baselines.
> >
> > ---
> >
> > ### W3.1. When compared to "traditional" models, include more recent advancements like TiRGN in evaluation.
> > **TKG Baseline Models**: We appreciate your suggestion to add another more recent TKG baseline TiGCN [2] to enhance our study. We have selected RE-GCN [1] as the traditional TKG baseline because it is a well-established method widely compared across TKG studies, and its training objectives encompass both link prediction and relation prediction. Compared to RE-GCN, which uses an RNN combined with graph convolutions to model event sequential dependencies, TiRGN mainly enhances the capability to capture not just local and sequential events, but also repetitive and cyclical global patterns by having local and global encoders.
> >
> > We are currently reproducing TiRGN results using their [official code](https://github.com/Liyyy2122/TiRGN) on MIRAI. However, TiRGN's setup differs from the agent method in two key aspects: a) it outputs ranking lists rather than direct predictions, and b) requires parameter tuning for optimal history lengths. We aim to complete this adaptation and comparison by the rebuttal deadline or share results via public comments thereafter.
> >
> > ---
> >
> > ### W3.2. (part of Q5) Train the traditional models on more recent cutoff dates.
> > **Training Cutoff Date**: To ensure fair comparison and study cutoff date effects, we further evaluated TKG baselines across four cutoff dates: 2023-06 (using ~50\% of MIRAI data), 2023-08, 2023-10 (matching GPT-4o-mini’s cutoff), and 2023-12 (matching Mistral/Llama’s cutoff). For each cutoff data, we train TKG models on events up to that month’s end and validate on the followin month. For example, for 2023-10 cutoff, training uses events through 2023-10-31, validating on 2023-11 data.
> >
> > However, we note three key differences in how these cutoff dates apply to TKG methods versus LLMs:
> > 1. LLMs' pretraining data extends before MIRAI's 2023-01 start date.
> > 2. LLMs' cutoffs reflect pretraining on general knowledge for next token prediction, while TKG methods are trained specifically for our task
> >
> > ---
> >
> >
> > ### Q5.1 What does "fine-tuned" mean in this context?
> > **Clarification on Training**: While we originally have included more training and evaluation details for the traditional baselines in Appendix D.3, we acknowledge that the use of the term ‘fine-tuned’ in the main table might be misleading. We have corrected it to task-specifically ‘trained’ in Table 2 and its following discussions in the revised manuscript. We specify this setting to highlight its difference with LLM-based methods: all base LLMs for agents in our experiment table are not task-specifically fine-tuned on MIRAI’s data for the forecasting task; instead, they are zero-shot with their pre-trained model weights from general knowledge.
> >
> > ---
> >
> > ### Q5.2 Were the models trained on GDELT event data?
> > **Training Data**: TKG baselines are trained on the temporal knowledge graphs constructed from the structured events in MIRAI, which is a cleaned-up version of raw GDELT data.
> >
> > ---
> >
> > ### Q5.3. How was RE-GCN applied, in single-step or multi-step prediction mode?
> > **Prediction Settings**: To align the same setting of a forecasting horizon of 1 day in our agent experiments, we use the single-step prediction model for TKG methods. This means that during testing time, both the LLM agents and TKG model are provided with all ground-truth historical events up to 1 day before the query event date. However, following RE-CGN and other TKG prediction settings, we use a sliding window of w days, where the trained model primarily aggregates the temporal and relational information within this window to predict future events. We select w from 3, 7, and 10.
> >
> > **Evaluation Metrics**: For evaluation, the TKG model also supports a full ranking list for all relation types with their prediction scores. In particular, RE-CGN obtains the prediction score of each relation by first using ConTransR to obtain the query embedding for (s, o) and then multiplying it with each relation embedding. Therefore, with this score, we are able to use **ranking-based metrics**. We apply MRR and Hit@10. We follow TKG forecasting conventions of time-aware filtering [2-3] for the ranking-based metrics. For **set-based evaluation**, we also acknowledge that there are an infinite number of possible thresholds to select the set of relations with their score above the threshold. In the experiment table, we report the set prediction results with the threshold set to be 0 - which indicates a relatively favorable alignment between the query embedding (s,o) and the relation embedding r.

---

> > > ### Author Response · Authors · 2024-11-23
> > >
> > > **Performance of the TKG Baselines**: The experiment results of the RE-GCN are shown in row 1-4 of the table. We observe that the RE-GCN demonstrates strong performance in high-level relation prediction, resulting in 0.3 for binary-level relation KL (cooperation or conflict) and 0.8 for quadratic-level relation KL (verbal/material cooperation/conflict), reflecting its advantage in capturing the high-level dynamics over bilateral relationships. It also shows consistently high recall in more fine-grained relation levels (90.6% Recall at first-level relation, and 83.9% recall at second-level relations with 2023-06 cutoff), suggesting its effectiveness in modeling positive correlation between query and multiple ground-truth relations.
> > >
> > > **TKG’s Comparison and Insights for Agents**: Comparing with the ReAct agent (Set Prediction, the setting we used in the main paper) in row 9, we find that although the LLM agent also could achieve much higher forecasting precision, it obtains much lower recall than the RE-GCN baseline. We manually go through the reasoning traces generated by the LLM agent in the test set, and conclude the following possible insights and future directions of improvement compared with TKG baselines:
> > >
> > > One major possible reason is that the current agent mostly focuses on analyzing only the bilateral events between the query entities s and o. For instance, it typically sets the function parameter `head_entities` to the query subject, and `tail_entities` to the query object only, obtaining only events and news directly between the two. However, this analysis largely oversimplifies real international relationships where countries have engaged in multi-party and complex interactions. Events between two countries could be affected by regional or global events. In contrast, TKG methods excel in capturing this multi-party and multi-relational history by leveraging multi-layer graph convolutions, where neighboring information is aggregated to enhance the modeling of each node (entity embedding) and edge (relation embedding). Therefore, when making predictions between two countries, the TKG models consider a much broader relation network than the current LLM agents, leading to higher recall and better generalization, especially when the bilateral history is sparse.
> > >
> > > Another problem we observed from the current LLM agent behavior is its tendency to hallucinate, particularly in listing the existence of relations and interpreting the meanings of relations in CAMEO ontology, which leads to lower precision and recall. For example, in the example we show in Appendix F.3, in its trajectory step 3, the agent attempts to explain and conclude its final prediction: ‘042’ Make a public statement (high frequency in historical data); ‘036’ Negotiate (also high frequency); ‘057': Express intent to cooperate (indicated by recent news context).  However, the correct meanings are ‘Make a visit’ for ‘042’ and ‘Sign formal agreement’ for ‘057’. This example highlights two issues: firstly, the LLM agent's overconfidence in its understanding of the CAMEO ontology without verifying the relation meanings through function calls (such as `map_cameo_to relation` and `map_description_to_cameo`); secondly, its over-reliance on the semantic meaning of relations rather than their structural context. In contrast, TKG models learn relations by leveraging the historical graph structure, which inherently learns to capture the contextual meaning of each relation.
> > >
> > > To enhance future LLM agents, incorporating a hybrid approach that combines semantic understanding with structural learning from TKGs could help to enhance relation modeling and address certain hallucinations.
> > >
> > > ---
> > >
> > > [1] REGCN: Temporal Knowledge Graph Reasoning Based on Evolutional Representation Learning, SIGIR 2021.
> > >
> > > [2] TiRGN: Time-Guided Recurrent Graph Network with Local-Global Historical Patterns for Temporal Knowledge Graph Reasoning. IJCAI 2022.

---

> > > > ### Author Response · Authors · 2024-11-23
> > > >
> > > > ### W2&Q1. Why did the authors choose to focus solely on relation forecasting rather than including link prediction (predicting subjects and objects), which is more standard in TKG forecasting, and KG completion evaluation?
> > > >
> > > > Thank you for your valuable question. We want to clarify that previous temporal knowledge graph research has explored both link prediction and relation prediction tasks, with notable works like RE-GCN[1], TiRGN[2], RE-GCN[3], DynamicGCN[4], Glean[5], and CMF[6] focusing on relation prediction. It is important to emphasize that neither task holds inherent priority over the other; rather, each serves distinct analytical purposes tailored to specific research objectives.
> > > >
> > > > In MIRAI, we focus on relation prediction as our primary task given our interest in studying dynamic relationship shifts between countries over time. This choice is particularly significant due to the structured nature of CAMEO relations in international event data. The CAMEO ontology offers a hierarchically organized framework that encompasses the entire spectrum of international interactions, ranging from material cooperation (e.g., providing aid, military collaboration) and verbal cooperation (e.g., diplomatic statements, expressions of support) to verbal conflict (e.g., accusations, rejections) and material conflict (e.g., military actions, sanctions). This natural progression of political interactions—from the most cooperative to the most conflictual—provides a clear framework for analyzing the evolution of international relationships and a nuanced benchmark for assessing models' capacity to forecast shifts in these dynamics over time.
> > > >
> > > > Meanwhile, we also acknowledge the importance of entity prediction in forecasting. MIRAI technically can support entity prediction tasks and this formulation. In our future work, we plan to explore how MIRAI's framework could be extended to evaluate LLM agents on entity prediction, enhancing our understanding of how entities might engage in future international events based on their historical behaviors and current geopolitical climate.
> > > >
> > > > [1] History Repeats Itself: A Baseline for Temporal Knowledge Graph Forecasting. IJCAI 2024.
> > > >
> > > > [2] TiRGN: Time-Guided Recurrent Graph Network with Local-Global Historical Patterns for Temporal Knowledge Graph Reasoning. IJCAI 2022.
> > > >
> > > > [3] REGCN: Temporal Knowledge Graph Reasoning Based on Evolutional Representation Learning, SIGIR 2021.
> > > >
> > > > [4] Dynamic knowledge graph based multi-event forecasting. KDD 2020.
> > > >
> > > > [5] Learning dynamic context graphs for predicting social events. KDD 2019.
> > > >
> > > > [6] Understanding event predictions via contextualized multilevel feature learning. CIKM 2021.
> > > >
> > > > ---
> > > >
> > > > ### W4. Include the presented human forecasting performance from the appendix into the main paper
> > > >
> > > > In response to your comment, we have incorporated the human forecasting performance results into the main paper (page 7 Table 2). Additionally, we note that the human evaluation was conducted on a subset of 51 test events due to resource constraints and the time-intensive nature of expert evaluation. This is also noted in the caption of Table 2.
> > > >
> > > > Moreover, for fair comparison, we have run all LLM agent experiments on this same subset, with results presented in Table 6 of the Appendix D.4. The results show that human performance surpassed that of LLM agents in most metrics, especially in recall. This highlights significant room for improvement in LLM agents.
> > > >
> > > > ---
> > > >
> > > >
> > > > ### W5. Condense the Appendix by not include the full API specification and implementation, and README of the github repo, but instead a link to the github repo
> > > >
> > > > Thank you for this suggestion. We have streamlined the Appendix by:
> > > > 1. Removing the full API documentation and implementation details and providing a link to anonymous github repo.
> > > > 2. Retaining only the key list of data classes and functions defined in the API that directly support the paper’s contribution in the agentic environment and the code-based tool use setting.

---

> ### Author Response · Authors · 2024-11-23
>
> ### Q2. What motivated the choice of the current metrics, and why were more commonly reported metrics, like MRR and Hits@k, excluded?
>
> Thank you for this important question about metric selection. We'd first point out that the expected outputs for the two types of metrics are different:
> **Set-based Metrics**:
> Models output a discrete set of predicted relations for each query. We evaluate these predictions using:
> - **Precision**: Proportion of predicted relations that are correct
> - **Recall**: Proportion of actual relations that are predicted
> - **F1 Score**: Harmonic mean of precision and recall
>
> **Ranking-based Metrics**:
> Models output an ordered list of relations with associated scores. Following TKG conventions of time-aware filtering, we implement:
> - **Mean Reciprocal Rank (MRR)**: Average reciprocal of the first correct relation's rank
> - **Hit@10**: Proportion of queries where at least one correct relation appears in top-10 predictions
>
> With this understanding, we acknowledge that agent-based methods also can support ranking-based output and evaluation, but necessary prompt modifications need to be applied. In added experiments, we implemented the ReAct ranking-based variants as presented in the last 6 rows of the table.  For *React (Rank)*: we instruct the agent to generate an ordered relation list; For *React (Rank w. Prob)*:  we additionally instruct the agent to generate probability scores for ranked relations. We tested on different list lengths k=10, 30, all, where only *k=all* is valid to be used for MRR calculation, while all configurations support Hit@10 evaluation.
>
> We prioritize set-based metrics as our primary evaluation criteria for LLM Agent-based methods mainly considering the current LLM model capabilities. Our primary consideration is that current LLMs are better suited to generating discrete predictions through natural language reasoning than producing comprehensive ranked lists. This is evidenced by our experimental findings:
> - **List Length Sensitivity**:
>      ReAct agent's performance deteriorates with longer list requirements, with k=30 and k=all performing worse than k=10 in ranking metrics (Hit@10: 25.7% for k=10 vs 12.0% for k=30).
> - **Prompt Sensitivity**:
>      Performance varies between pure ranking and probability-weighted ranking (Hit@10: 25.7% vs 26.8% for k=10, and MRR: 13.9% vs 12.6% for k=all), suggesting that ranking outputs are sensitive to the prompt formulation and output format.
>
> Given these challenges, we opted for metrics that more directly and reliably assess the agents' ability to predict discrete events without the confounding factors introduced by list generation and ranking.
>
> ---
>
> ### Q3. The authors commit to updating the dataset split monthly. How long is this commitment expected to last, and will it be sustained indefinitely?
>
> We commit to maintaining monthly dataset split updates for at least the next year through our fully automated pipeline. The update process requires minimal human intervention, as the pipeline is designed to handle data collection, processing, and splitting autonomously. As long as the data format from CAMEO remains consistent, the pipeline can continue to generate updates reliably.
>
> To ensure sustainability, we have made the automation scripts publicly available in our repository, allowing researchers to independently verify and execute the update process. Beyond this, we plan to continue updates indefinitely as long as computational resources and data availability permit. To ensure sustainability, we have also documented the entire update process and will actively maintain the codebase, making it resilient to potential API or data source changes.
>
> ---
>
> ### Q4. What is the rationale behind applying higher thresholds to the test data (100 daily mentions, 5 news articles) than to the training data?
>
> The different thresholds between training and test data serve distinct methodological purposes. The higher test thresholds (100 daily mentions, 5 news articles) enable **precise and robust evaluation** of our benchmark by:
> 1. Ensuring test events have sufficient contextual information for reliable ground truth labeling
> 2. Reducing noise from potential false positives or misreported events
> 3. Allowing thorough verification across multiple independent sources
> The higher test thresholds align with real-world applications where stakeholders focus on detecting and analyzing significant international events.
>
> Meanwhile, the lower thresholds for training data are deliberately chosen to maximize the dataset's utility. First, they create a comprehensive training set that captures both major and minor events, enabling models to learn the full spectrum of international relations. Second, by providing a broader training set, we enable future researchers to either use the complete dataset or apply custom filtering criteria based on their specific research needs, enhancing the dataset's long-term value to the community.

---

> ### Author Response · Authors · 2024-11-25
> **Inqury for discussion**
>
> Dear Reviewer ZMPf,
>
> Thank you again for your detailed and constructive feedback. We greatly value the time and effort you invested in reviewing our work, and we hope that our additional experiments and discussions addressed your concerns comprehensively. Specifically:
>
> 1. **Heuristic Baseline Comparisons:** We included the Recurrency [1] heuristic model and provided a detailed analysis compared to LLM agents (see Table 1). This addition highlighted the strong recall of heuristic methods, emphasizing areas for future agents.
>
> 2. **TKG Baseline Analysis:** We have detailed the training settings used for TKG models, incorporated additional comparisons with agents, and explored the effects of training cutoff dates. The updated analysis reinforces the strengths and areas of improvement of LLM agents as compared to TKG methods.
>
> 3. **Metric Selection and Extensions:** We provided results on ranking-based metrics (MRR and Hit@k) for the ReAct agent with necessary modifications to a ranking list output. We underscored the trade-offs between set-based and ranking-based evaluations for current LLM methods limited by current model capacity, explaining our rationale for prioritizing set-based metrics.
>
> 4. **Task Focus on Relation Forecasting:** We clarified our focus on relation prediction as the primary task due to its alignment with the structured nature of CAMEO relations and the goal of capturing fine-grained dynamic shifts in international relationships.
>
> 5. **Rationale for Test Data Thresholds:**  We clarified that the higher thresholds applied to the test data were designed to ensure precise and robust evaluation by focusing on the contextual-rich and cross-verified international events; while the lower thresholds for training data enable models to learn from a broader spectrum of event types.
>
> 6. **Appendix and Human Baselines:** We streamlined the appendix by linking to the full API documentation and included human forecasting performance directly in the main paper (Table 2).
>
> 7. **Future Dataset Updates:** We have clarified our commitment to maintaining monthly dataset updates is supported by an automated pipeline. Additionally, we have documented and open-sourced this process to ensure its sustainability and usability for the broader research community.
>
> We would like to inquire if there are any questions about our rebuttal, for which we're happy to provide additional information and further clarifications. We deeply appreciate your time and effort in reviewing our work.
>
> Sincerely,
> Authors

---

> > ### Author Response · Authors · 2024-11-29
> > **Gentle Reminder: Awaiting Your Response to Our Rebuttal**
> >
> > Dear Reviewer ZMPf,
> >
> > I hope this message finds you well. We recently submitted our rebuttal and would like to kindly request your feedback on our responses.
> >
> > We understand that your schedule is demanding and greatly appreciate the time and effort you dedicate to the review process. Your insights are invaluable to us, and we are eager to address any further questions or concerns you may have.
> >
> > Thank you for your attention to this matter. We look forward to your response.
> >
> > Best regards,
> >
> > Authors

---

> ### Author Response · Authors · 2024-12-03
>
> Dear Reviewer ZMPf,
>
> Thank you again for taking the time to review our paper. We greatly appreciate your detailed feedback and your recognition of key strengths of our benchmark work, such as the clarity and structure, the contamination-free test sets, insightful visuals, and the in-depth analysis across diverse aspects. Your thoughtful comments are invaluable to us.
>
> In response to your feedback, we have conducted extensive experiments with heuristic and TKG baselines, and agentic methods with both set-predictions and ranking-based metrics. We have also provided detailed clarifications on our contribution ([global comment](https://openreview.net/forum?id=gzzX4ZeErx&noteId=ZueduytQF6)) and on your concerns in our rebuttal replies with our paper revised accordingly. We hope our detailed responses adequately address your concerns.
>
> With the discussion deadline approaching, we would appreciate your  attention to our rebuttal and any further feedback you may have, as it will give us the opportunity to provide more details before the author-reviewer discussion session ends and help us continue to improve our work. Thank you again for your time and valuable comments!
>
> Best regards,
> Authors

---

### Author Response · Authors · 2024-11-23
**Global Comment**

We sincerely thank all the reviewers for their insightful and encouraging feedback on our benchmark manuscript. We appreciate the recognition of the clarity and reproducibility (noted by Reviewers ZMPf and SzJ2), featuring contamination-free test sets with regular updates (ZMPf) and comprehensive code documentation (SzJ2). We also thank the reviewers for noting the the extensive experimental evaluations (mentioned by all reviewers), including diverse analyses across different relation types (ZMPf), comparisons with competing benchmarks (AZdo, uDbL) and the well-crafted visualizations (ZMPf, SzJ2).

We have addressed the reviewers' comments with new experiments and substantial revisions. Our key additions include:
- Adding experiments for heuristics-based forecasting baseline, Recurrency, with different data cutoffs (Reviewer ZMPf’s W1, W6, W7)
- Adding experiments for traditional temporal knowledge graph-based method, RE-CGN, with different training data cutoffs (Reviewer ZMPf’s W3, Q5)
- Adding ranking-based forecasting task formulation and corresponding experiments with LLM agents, ReAct (Rank) and ReAct (Rank w.Prob), studying the effect of rank list length and format (Reviewer ZMPf’s Q2)
- Adding retrieve-augmented generation (RAG) baselines, including structured event retrieval, textual news retrieval, and hybrid of the two (Reviewer AZdo’s W1, Q3; Reviewer uDbL’s W2, Q1; Reviewer SzJ2’s O5)
- Adding more agent forecasting examples and analysis (Reviewer uDbL’s W1)
- Adding experiments studying the effect of LLM parameter size for forecasting (Reviewer SzJ2’s Q3)
- Adding a new baseline of augmenting CAMEO ontology to LLM Direct IO (Reviewer SzJ2’s Q4)


### Contributions of this paper

We further make the following clarification to highlight the contribution of our paper.
- **Scope.** Most importantly, our manuscript clearly defined its scope in lines 10-14, 21-25, and 90-96: we introduce **a benchmark dataset** with an **agentic environment** to systematically evaluate LLM agents' forecasting capabilities, which is a newly emerging and promising direction in LLM development.
- **Benchmark compatibility.** To facilitate meaningful comparisons, we have made further efforts so that our benchmark is compatible with traditional TKG methods, NLP based approaches and LLM based methods like retrieval-augmented generation (RAG), allowing us to better contextualize the LLM agents' performance.
- **Focus.** Our contribution is not proposing state-of-the-art methods or advocating for current LLM agent baseline performances. Rather, we aim to **provide a foundational benchmark that can drive future development in this field**, particularly given LLMs' demonstrated strengths in tool utilization and environmental interaction.
- **Findings and analysis.** Our extensive experiments reveal challenges and opportunities for LLM agents in forecasting: diverse information gathering and tool-use are critical for temporal forecasting, advanced action types like Code Block benefit stronger LLM agents, scaling inference-time compute improves small LLMs, and there is substantial room for LLM agents to improve in code generation, tool-use planning, reasoning, and long-horizontal forecasting.
- **Fostering future research.** LLM agents have become a prominent focus in recent research [1-8], driven by interest in their ability to interact with dynamic environments through tool use, complex reasoning, and decision-making. This highlights the need for benchmarks tailored to real-world applications. MIRAI thus offers a uniquely designed platform centered around event forecasting, enabling researchers to explore and develop LLM agents in a focused yet flexible setting.
To further reflect this, we have also added a contribution statement in lines 91-98 of the revised manuscript.

[1] Large Language Model based Multi-Agents: A Survey of Progress and Challenges

[2] AgentGym: Evolving Large Language Model-based Agents across Diverse Environments.

[3] SWE-agent: Agent-Computer Interfaces Enable Automated Software Engineering

[4] OpenHands: An Open Platform for AI Software Developers as Generalist Agents

[5] WorkArena: How Capable Are Web Agents at Solving Common Knowledge Work Tasks?

[6] Voyager: An Open-Ended Embodied Agent with Large Language Models

[7] Approaching Human-Level Forecasting with Language Models

[8] From News to Forecast: Integrating Event Analysis in LLM-Based Time Series Forecasting with Reflection

---

### Meta-Review · Area_Chair_qoFP · 2024-12-21

**Metareview:**

This paper presents MIRAI, a practical benchmark for predicting future interactions between countries. Overall the reviews are mixed (two 5's and two 6's), and there was an in-depth discussion among the reviewers after the rebuttal. I will mainly quote the reviewers to respect their thoughts. On the positive side, this work is extremely serious, with all the components clearly explained and the code well-documented and reproducible. The authors' effort during the rebuttals demonstrated their desire to produce quality work. However, there are also remaining concerns: (1) The additional RAG-based experiments provide better results than the authors' approach; (2) While the authors claim to have provided additional forecasting results in the Appendix, they are still related to one specific example mentioned in the paper, the relations between China and Australia; (3) The authors claim to have provided an interactive agent demo in the following link, but this does not seem to work: https://drive.google.com/file/d/1kKvYdAYv5hed-sbF_QE1moP-dQbfYoXI/view; (4) Additionally, the repository linked in the paper is also not available: https://anonymous.4open.science/r/ForecastAgent-3419. (We note the anonymous git did work during the review period.) There was an agreement on the lack of relevance of the agent approach to this problem and uncertainty about the impact of the dataset. So while the work is quite thorough, and one therefore can build on the developed benchmark, there are doubts on the need for an agent-based approach and no clear picture on how relevant the benchmark would be for the audience. Overall, I think the paper can be improved and should go through another round of reviewing.

**Additional Comments On Reviewer Discussion:**

Main remaining concerns:
- The additional RAG-based experiments provide better results than the authors' approach.
- While the authors claim to have provided additional forecasting results in the Appendix, they are still related to one specific example mentioned in the paper, the relations between China and Australia.
- There is a lack of relevance of the agent approach to this problem.

---

### Decision · Program_Chairs · 2025-01-22

Reject